# Uncertainty Quantification for Regression Using Proper Scoring Rules

## Abstract

Quantifying uncertainty of machine learning model predictions is essential for reliable decision-making, especially in safety-critical applications. Recently, uncertainty quantification (UQ) theory has advanced significantly, building on a firm basis of learning with proper scoring rules. However, these advances were focused on classification, while extending these ideas to regression remains challenging. In this work, we introduce a unified UQ framework for regression based on proper scoring rules, such as CRPS, logarithmic, squared error, and quadratic scores. We derive closed-form expressions for the resulting uncertainty measures under practical parametric assumptions and show how to estimate them using ensembles of models. In particular, the derived uncertainty measures naturally decompose into aleatoric and epistemic components. The framework recovers popular regression UQ measures based on predictive variance and differential entropy. Our broad evaluation on synthetic and real-world regression datasets provides guidance for selecting reliable UQ measures.

## 1 Introduction

Predictive models trained with machine learning are widely used, yet their predictions are often uncertain. Quantifying this uncertainty is essential, especially in safety-critical applications (Helou et al., 2020; Su et al., 2023). Uncertainty quantification (UQ) provides a way to measure the confidence of a model in its predictions (Hüllermeier & Waegeman, 2021; Gal & Ghahramani, 2016). One of the central goals of UQ is to separate aleatoric uncertainty (AU) associated with irreducible noise inherent to the data-generating process, from epistemic uncertainty (EU), which arises from the limited knowledge of the underlying model (Hüllermeier & Waegeman, 2021). Distinguishing these types can help identify the causes of uncertainty and select the mitigation strategy.

Although UQ of machine learning models predictions is a mature field, it was historically mostly relying on ad hoc heuristic uncertainty measures. Recently, the coherent theoretical body was developed (Wimmer et al., 2023; Kotelevskii et al., 2025; Hofman et al., 2024; Schweighofer et al., 2025), mainly focusing on classification tasks. The core idea of these methods is to consider pointwise risk as a natural measure of predictive uncertainty. Early works (Kotelevskii et al., 2022; Lahlou et al., 2023) applied this idea to specific losses and model families. More recent works extended this approach via proper scoring rules (Gneiting & Raftery, 2007), deriving general uncertainty measures through Bayesian approximations to the different components of the pointwise risk (Kotelevskii et al., 2025; Hofman et al., 2024; Schweighofer et al., 2025).

However, many real-world problems involve the prediction of continuous, unbounded outcomes (Gal & Ghahramani, 2016; Amini et al., 2020; Valdenegro-Toro & Mori, 2022), while aforementioned approaches have not yet been adapted for regression. This paper closes that gap by extending the proper-scoring-rule UQ framework to regression tasks. We consider various appropriate scoring rules developed for regression, such as CRPS which is a leading tool in meteorology Alet et al. (2025); Lang et al. (2025). That allows us to derive a wide family of uncertainty measures that can be readily decomposed into aleatoric and epistemic components. Interestingly, our framework allows to recover some of the established entropy- and variance-based measures for UQ in regression (Depeweg et al., 2018; Bülte et al., 2025).

Our main **contributions** can be summarized as follows.

- A theoretical formulation of regression UQ based on proper scoring rules, with separate quantification of aleatoric and epistemic components; see Section 3
- Closed-form, practically computable approximations for these components under standard regression assumptions for ensemble-based estimators; see Tables 2,3 and Appendix A.
- A broad empirical evaluation on synthetic and real-world datasets, covering selective prediction, out-of-distribution detection, and active learning, culminating in recommendations for selecting uncertainty measures in practice; see Section 5.

## 2 BACKGROUND ON PROPER SCORING RULES

We start by defining the proper scoring rules and introducing the necessary notation. This section is based on the related works on proper scoring rules (Gneiting & Raftery, 2007; Waghmare & Ziegel, 2025), and follows their notation.

We consider supervised learning problems with input $x \in \mathcal{X}$ and output $y \in \mathcal{Y}$. Let $\mathcal{P}$ be a convex class of probability measures on $\mathcal{Y}$. A probabilistic forecast is any probability measure $\widehat{P} \in \mathcal{P}$. For classification, machine learning models usually produce a probabilistic forecast by default: the output is the distribution $\widehat{p} = \widehat{p}(y \mid x)$ over class labels. In regression problems, the standard approach is to proceed with just a point prediction $\widehat{y} = f(x) \in \mathcal{Y}$. However, to efficiently deal with predictive uncertainty, a model may output parameters of the distribution $\widehat{p}(y \mid x)$, e.g., $\widehat{\mu}(x), \widehat{\sigma}(x)$ for a Gaussian distribution.

A scoring rule is an extended real-valued function $S \colon \mathcal{P} \times \mathcal{Y} \to \bar{\mathbb{R}} = [-\infty, +\infty]$ such that $S(\widehat{P}, \cdot)$ is integrable for all $\widehat{P} \in \mathcal{P}$. Thus if the forecast is $\widehat{P}$ and $y$ materializes, the forecaster's penalty is $S(\widehat{P}, y)$. Usually, we are interested in assessing the predictive quality of the forecast $\widehat{P}$ on average over possible values of $y$. To achieve that, we consider the expected score under $P$:

$$S(\widehat{P}, P) = \int S(\widehat{P}, y) \, dP(y) \tag{1}$$

The practically important family of scoring rules are so-called *proper scoring rules* that incentivize the learning of the true data distribution. The scoring rule $S$ is called proper relative to $\mathcal{P}$ if

$$S(P, P) \leq S(\widehat{P}, P) \text{ for all } \widehat{P}, P \in \mathcal{P}. \tag{2}$$

The scoring rule is strictly proper if the equality in (2) holds iff $\widehat{P} = P$. Thus, any (strictly) proper scoring can be used for evaluation of probabilistic forecasts ensuring that its minimum is attained at the true data distribution.

The following definitions regarding proper scoring rules are useful for our future exposition.

**Definition 1.** An expected score between a measure and itself is called the *entropy function*:

$$H(P) = S(P, P) = \int S(P, y) \, dP(y).$$

Entropy determines the smallest possible error that one can achieve which corresponds to the Bayes optimal predictor (the true data distribution).

**Definition 2.** A *divergence function* is the difference between the expected scores of the predicted and true distributions:

$$d(\widehat{P}, P) = S(\widehat{P}, P) - S(P, P) = S(\widehat{P}, P) - H(P),$$

quantifying how much worse it is to predict with distribution $\widehat{P}$ than with the true data distribution $P$.

**Proper scoring rules for regression.** A number of proper scoring rules exist for probability distributions with continuous support (Gneiting & Raftery, 2007). In this work we consider several of the most common proper scoring rules that could be used for regression such as *continuous ranked probability score (CRPS), logarithmic, quadratic and squared error scores* (see their definitions in Table 1). All of these scores are strictly proper except for the squared error score, which is just proper. In what follows, we will show how to derive practical uncertainty estimates for regression based on these scores.

Table 1: Examples of commonly considered proper scoring rules for regression.

| Name | Definition |
|------|-----------|
| Continuous ranked probability score | $\text{CRPS}(\widehat{P}, y) = \int_{\mathbb{R}} \left( F_{\widehat{P}}(t) - \mathbb{I}\{y \leq t\} \right)^2 dt$ |
| Logarithmic score | $\text{LS}(\widehat{P}, y) = -\log \widehat{p}(y)$ |
| Quadratic score | $\text{QS}(\widehat{P}, y) = -2\,\widehat{p}(y) + \int_{\mathbb{R}} \widehat{p}(t)^2 dt$ |
| Squared error score | $\text{SE}(\widehat{P}, y) = \left( y - \mathbb{E}_{Y \sim \widehat{P}}[Y] \right)^2$ |

# 3 REGRESSION UNCERTAINTY MEASURES VIA PROPER SCORING RULES

In this section, we introduce the uncertainty quantification framework for regression and show how one can end up with practical equations for uncertainty measures using proper scoring rules.

## 3.1 POINTWISE RISK AS AN UNCERTAINTY MEASURE

Let $\mathcal{D} = \{(x_i, y_i)\}_{i=1}^n$ be a training set of $n$ independent and identically distributed (i.i.d.) samples from the joint distribution $P(X, Y)$, where $X_i \in \mathcal{X} = \mathbb{R}^d$ and $Y_i \in \mathcal{Y} = \mathbb{R}$. We denote the true conditional distribution of $Y$ at $X = x$ by $P(Y \mid X = x)$. The standard goal of statistical learning is to approximate this conditional distribution with some model $\widehat{P}(Y \mid X = x)$ using the training data. For brevity, we further assume that everything is implicitly conditional on $X = x$, so we write $P = P(Y \mid X = x)$ and $\widehat{P} = \widehat{P}(Y \mid X = x)$. We will denote the corresponding densities by $p$ and $\widehat{p}$, and the corresponding cumulative distribution functions by $F_P$ and $F_{\widehat{P}}$.

The alternative view on the expected score $S(\widehat{P}, P)$ in (1) is to interpret it as an expected error of prediction by model $\widehat{P}$ at a given point. Such an object is usually called *pointwise risk* in statistical learning literature. There are three key risk quantities that we consider in our framework:

- **Total risk.** The expected score between the estimated $\widehat{P}$ and the true $P$ distributions, capturing the overall uncertainty of a prediction:

$$R_{\text{Tot}}(\widehat{P}, P) = \int S(\widehat{P}, y)\, dP(y) = S(\widehat{P}, P).$$

- **Bayes risk.** The risk we would incur if we could predict with the *true* distribution itself:

$$R_{\text{Bayes}}(P) = \int S(P, y)\, dP(y) = S(P, P) = H(P).$$

Since it does not depend on the model, the nature of this error is purely *aleatoric*. It is equal to the entropy of the data distribution and, for any proper score, it equals to the smallest possible expected error of prediction.

- **Excess risk.** It measures how much worse the model is compared to the perfect prediction:

$$R_{\text{Exc}}(\widehat{P}, P) = R_{\text{Tot}}(\widehat{P}, P) - R_{\text{Bayes}}(P) = d(\widehat{P}, P).$$

It is equal to the score's divergence and captures the part of the error not described by the randomness in the data itself. Thus, it is naturally related to *epistemic* uncertainty.

The risks above depend on the true distribution $P$, which is unknown. In the next section, we will consider various approximations of these risks.

## 3.2 APPROXIMATIONS OF THE RISKS

Following Kotelevskii et al. (2025); Hofman et al. (2024); Schweighofer et al. (2025), we consider Bayesian approximations to the risk expressions. We assume that we have a predictive parametric model, with the vector of parameters $\theta$ following a posterior distribution of $p(\theta \mid \mathcal{D})$, giving a *distribution* over predictive distributions $\widehat{P}_\theta$.

While the framework is model-agnostic, in order to obtain tractable closed-form expressions for particular risk approximations we proceed to make practical parametric assumptions about the

Table 2: Bayes-risk approximations under different proper scoring rules. For the Gaussian mixture model, we use the following notation: $\widehat{P}_{\text{ens}} = \frac{1}{M} \sum_{i=1}^{M} \mathcal{N}\left(\mu_i, \sigma_i^2\right)$. For $\widehat{R}_{\text{Bayes}}^3$, different plug-in estimates of the variance $\sigma_*^2$ can be used (see Appendix A.1.2).

| App. | CRPS | Logarithmic | Quadratic | SE |
|---|---|---|---|---|
| $\widehat{R}_{\text{Bayes}}^1$ | $\frac{1}{M\sqrt{\pi}} \sum_{i=1}^{M} \sigma_i$ | $\frac{1}{2M} \sum_{i=1}^{M} \log\left(2\pi e \sigma_i^2\right)$ | $-\frac{1}{2M\sqrt{\pi}} \sum_{i=1}^{M} \frac{1}{\sigma_i}$ | $\frac{1}{M} \sum_{i=1}^{M} \sigma_i^2$ |
| $\widehat{R}_{\text{Bayes}}^2$ | $\frac{1}{2M^2} \sum_{i=1}^{M} \sum_{j=1}^{M} A(\mu_{ij}, \sigma_{ij})$ | $-\int_{\mathbb{R}} \widehat{p}_{\text{ens}}(y) \log \widehat{p}_{\text{ens}}(y) dy$ | $-\int_{\mathbb{R}} \widehat{p}_{\text{ens}}(y)^2 dy$ | $\frac{1}{M} \sum_{i=1}^{M} (\sigma_i^2 + \mu_i^2) - \mu_*^2$ |
| $\widehat{R}_{\text{Bayes}}^3$ | $\sqrt{\sigma_*^2/\pi}$ | $\frac{1}{2} \log\left(2\pi e \sigma_*^2\right)$ | $-\frac{1}{2\sqrt{\pi}\sigma_*}$ | $\sigma_*^2$ |

Table 3: Excess risks $\widehat{R}_{\text{Exc}}^{1,1}$ (Bayesian averaging of both $P$ and $\widehat{P}$) for different proper scoring rules.

| Scoring Rule | $\widehat{R}_{\text{Exc}}^{1,1}$ |
|---|---|
| CRPS | $\frac{1}{M^2} \sum_{i=1}^{M} \sum_{j=1}^{M} \left[ A(\mu_{ij}, \sigma_{ij}) - \frac{\sigma_i + \sigma_j}{\sqrt{\pi}} \right]$ |
| Logarithmic | $\frac{1}{2M^2} \sum_{i=1}^{M} \sum_{j=1}^{M} \left[ \frac{\sigma_j^2 + (\mu_i - \mu_j)^2}{\sigma_i^2} - 1 \right]$ |
| Quadratic | $-\frac{2}{M} \sum_{i=1}^{M} H(\widehat{P}_i) - \frac{2}{M^2} \sum_{i=1}^{M} \sum_{j=1}^{M} \mathcal{N}\left(\mu_i \mid \mu_j, \sigma_i^2 + \sigma_j^2\right)$ |
| SE | $2\widehat{\text{Var}}\left[\mu_i\right] = \frac{2}{M} \sum_{i=1}^{M} (\mu_i - \mu_*)^2$ |

distribution $\widehat{P}_\theta$. For the scope of this work, we consider the Gaussian assumption $\widehat{P}_\theta = \mathcal{N}\left(\mu, \sigma^2\right)$, i.e. that the predictive distribution is defined by a vector of distribution parameters $\theta = (\mu, \sigma^2)$. Other parametric assumptions, e.g. Laplace, could be used to derive different measures. As a Bayesian model $p(\theta \mid \mathcal{D})$ we consider an ensemble of Gaussians with parameters $\theta_i = (\mu_i, \sigma_i^2)$: $\widehat{P}_{\text{ens}} = \frac{1}{M} \sum_{i=1}^{M} \widehat{P}_{\theta_i} = \frac{1}{M} \sum_{i=1}^{M} \mathcal{N}\left(\mu_i, \sigma_i^2\right)$.

From this perspective, we estimate any of the risks above in the following generic ways:

1. **Bayesian averaging of the risk.** Posterior expectation of the risk: $\mathbb{E}_{p(\theta|\mathcal{D})}\left[R(\widehat{P}, \widehat{P}_\theta)\right]$.

2. **Posterior predictive distribution.** First, average the distributions $\widehat{P}_{\text{ens}} = \mathbb{E}_{p(\theta|\mathcal{D})}\left[\widehat{P}_\theta\right]$, and use the plug-in as an estimate of the risk: $R(\widehat{P}, \widehat{P}_{\text{ens}})$.

3. **Gaussian surrogate.** We consider two Gaussian approximations for the posterior $\widehat{P}_*, \bar{P}_*$ (see Appendix A.1.2), which are used as plug-ins for the risk estimation: $R(\widehat{P}, \widehat{P}_* \text{ or } \bar{P}_*)$.

Note that either of these approaches can also be used to derive a specific $\widehat{P}$ estimate as well (Kotelevskii et al., 2025; Hofman et al., 2024; Schweighofer et al., 2025).

**Practical approximation.** Mixing the "how to treat $P$" choices with the "what we feed in as $\widehat{P}$" choices yields (in general) nine different concrete approximation pairs $(\widehat{P}, P)$. We discuss what particular options we choose in A.2 and summarize it in Table 7. Therefore, any risk estimate considered from now on carries two superscripts $\widehat{R}^{i,j}$, stating these choices. Numbers correspond to the generic way to approximate risks we outlined above, e.g., 1 denotes Bayesian averaging of risks. For Bayes risk, we need to build only the approximation of the true distribution $P$, for which we have three options. We illustrate the results for popular regression proper scoring rules for the approximation of Bayes risk in Table 2, and $\widehat{R}_{\text{Exc}}^{1,1}$ in Table 3. We refer to Appendix A for complete derivations and other approximation options [1].

**Generalization over earlier work.** Bülte et al. (2025) recently examined entropy- and variance-based uncertainty measures for regression under a new set of axioms. Those uncertainty measures are

---

[1] We use the following notation: $\mathcal{N}\left(a \mid b, s^2\right) = \frac{1}{\sqrt{2\pi s^2}} e^{-\frac{(a-b)^2}{2s^2}}$, $A(\mu, \sigma) = 2\sigma \phi\left(\frac{\mu}{\sigma}\right) + \mu \left[2\Phi\left(\frac{\mu}{\sigma}\right) - 1\right]$, and $\Phi, \phi$ are CDF and PDF of a standard normal.

defined as (Depeweg et al., 2018):

$$\mathrm{H}(\mathbb{E}_{p(\theta|\mathcal{D})}[\widehat{P}_\theta]) = \mathbb{E}_{p(\theta|\mathcal{D})}[\mathrm{H}(\widehat{P}_\theta)] + \mathbb{E}_{p(\theta|\mathcal{D})}[d_{\mathrm{KL}}(\widehat{P}_\theta, \mathbb{E}_{p(\theta|\mathcal{D})}[\widehat{P}_\theta])], \tag{3}$$

$$\mathrm{Var}(\mathbb{E}_{p(\theta|\mathcal{D})}[\widehat{P}_\theta]) = \mathbb{E}_{p(\theta|\mathcal{D})}[\mathrm{Var}(\widehat{P}_\theta)] + \mathrm{Var}_{p(\theta|\mathcal{D})}[\mathbb{E}(\widehat{P}_\theta)] \tag{4}$$

respectively, where H is the Shannon entropy and $d_{\mathrm{KL}}$ is the Kullback–Leibler divergence. In our scoring-rule view, these measures arise naturally as the *logarithmic* and *SE* special cases (using $\widehat{R}_{\mathrm{Tot}}^{1,2} = \widehat{R}_{\mathrm{Bayes}}^{1} + \widehat{R}_{\mathrm{Exc}}^{1,2}$ as risk approximations), see the derivations in Appendix A.2.2 and A.2.4 and Example 2.1 in Bülte et al. (2025). Thus, our proper scoring rule framework thus comprises entropy- and variance-based measures and introduces additional measures based on CRPS, a score widely used in meteorology for probabilistic forecasts, and the quadratic score. Furthermore, it allows us to easily derive new measures under different parametric assumptions or using alternative scoring rules.

## 4 RELATED WORK

**Axiomatic approaches in classification.** Despite the maturity of uncertainty quantification, the formalization of uncertainty measures appeared only recently. Most of the efforts in this formalization were made in the context of classification, where authors tried to formalize what makes a "good" uncertainty score. Specifically, Wimmer et al. (2023); Sale et al. (2024) propose different axiom sets for classification, but there seems to be no existing measure that satisfies all of them.

**Risk-based uncertainty measures.** The idea of viewing predictive uncertainty as *pointwise risk*, the expected value of a loss (statistical pointwise risk), was first put forward for classification in (Kotelevskii et al., 2022) and both settings in (Lahlou et al., 2023). However, they considered a specific class of models and specific loss functions. Building on this view, in (Schweighofer et al., 2025; Kotelevskii et al., 2025; Hofman et al., 2024) showed that the risk of any proper scoring rule admits a clean decomposition into the *Bayes risk* (aleatoric part) and the *excess risk* (epistemic part), where the former corresponds to the generalized entropy, while the latter is the notion of a Bregman divergence (Bregman, 1967). However, generalizing the ideas of proper scoring rules to regression was not considered.

**Towards regression.** The recent paper Bülte et al. (2025) introduces axioms for regression and assesses entropy and variance for them. However, they do not introduce other measures that can be derived from considering proper scoring rules. Our paper fills this gap by deriving regression-specific scores from proper scoring rules, specifically, the CRPS score.

**Practical deep-learning baselines.** Bayesian dropout (Gal & Ghahramani, 2016), Deep Ensembles (Lakshminarayanan et al., 2017), and the heteroscedastic losses (Kendall & Gal, 2017) remain standard tools of uncertainty quantification in regression in Deep Learning, because they scale well, though they lack a firm axiomatic footing. Deep Evidential Regression (Amini et al., 2020) offers a single-pass alternative that outputs Normal-Inverse-Gamma parameters, giving closed-form aleatoric and epistemic variances, yet recent studies question how reliably DER captures epistemic risk (Juergens et al., 2024).

## 5 EXPERIMENTS

In our experiments, we investigate the empirical behavior of the uncertainty measures introduced in Section 3. We begin by examining how these measures respond to different perturbations of the posterior $p(\theta \mid \mathcal{D})$ and showcase their behavior on a synthetic regression task. We then evaluate their utility on three downstream problems: selective prediction, out-of-distribution detection, and active learning. To further assess the relationship between measures, we evaluate rank correlations between measures and examine when different scores lead to different decisions. Unless stated otherwise, uncertainties are computed by Monte Carlo approximation through heteroscedastic deep ensembles (Lakshminarayanan et al., 2017); training details are provided in Appendix E.1.

## 5.1 CHARACTERIZATION OF UNCERTAINTY MEASURES

We investigate the behavior of the different uncertainty measures by applying shifts to the posterior $p(\theta \mid \mathcal{D})$, where $\theta = (\mu, \sigma^2)$, inspired by the investigation in Bülte et al. (2025, Figure 2). The original posterior is a uniform distribution in the predicted means $\mu \sim \mathcal{U}(-1, 1)$ and a uniform distribution in the predicted variances $\sigma^2 \sim \mathcal{U}(1, 2)$. This closed-form posterior allows us to estimate uncertainty measures directly without relying on sampling methods such as deep ensembles. We consider four different shifts to the posterior: (a) a location shift on the predicted means, (b) a location shift on the predicted variances, (c) a scale shift of the predicted means, and (d) a scale shift of the predicted variances, see Figure 1 (left column).

**Location shift of predicted means.** The distribution of predicted means shifts to $\mu \sim \mathcal{U}(1, 3)$, thus the location of the posterior shifts, but not its scale. The results are shown in Figure 1 (first row). None of the measures considered changes under this shift. This is desirable, as uncertainty measures should not be sensitive to the magnitude of mean predictions.

**Location shift of predicted variances.** The distribution of predicted variances shifts to $\sigma^2 \sim \mathcal{U}(2, 3)$, thus the location of the posterior shifts, but not its scale. The results are shown in Figure 1 (second

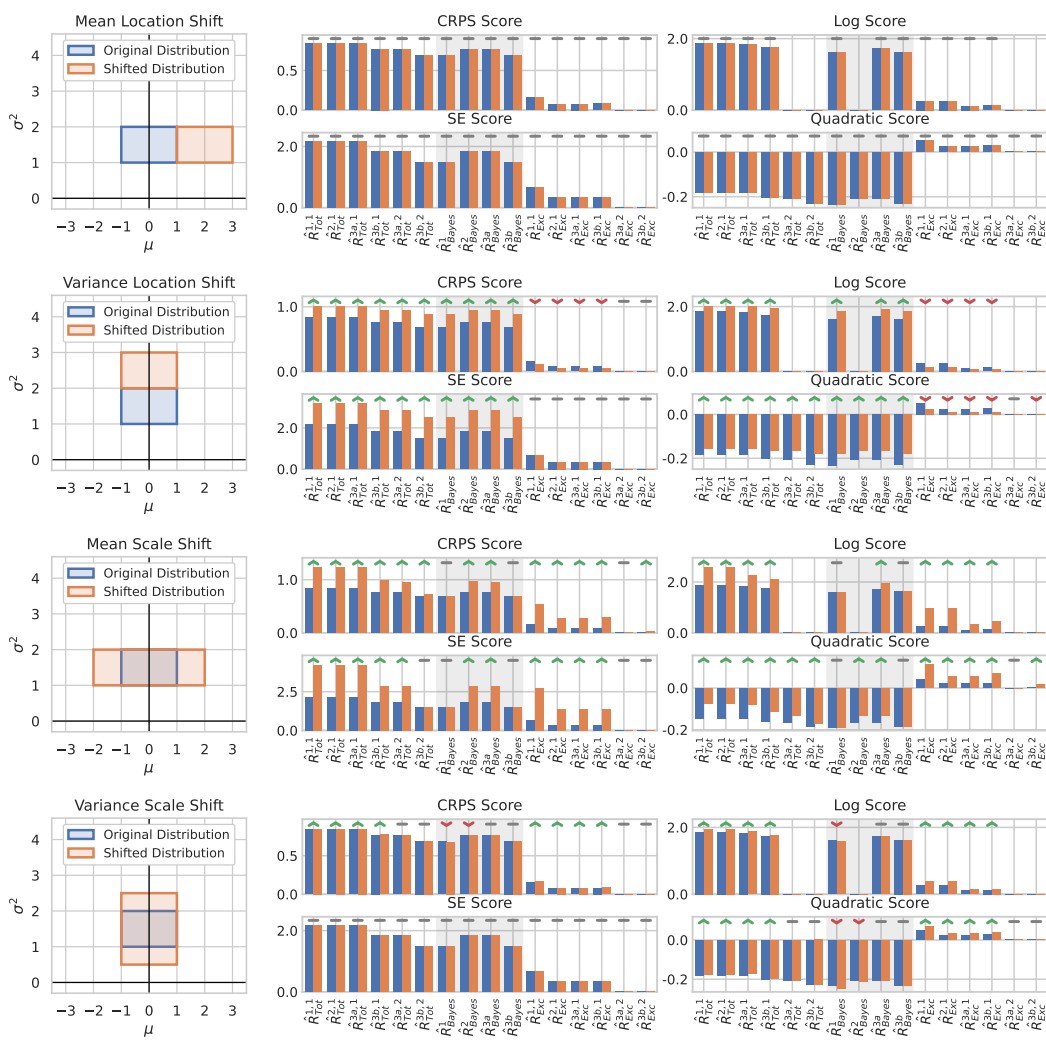

Figure 1: Behavior of uncertainty measures under location and scale shifts of the posterior distribution. Arrows indicate whether a measure increased or decreased due to the shift, gray bars indicate changes $< 1\%$, missing entries that the measure is not computable or constant zero.

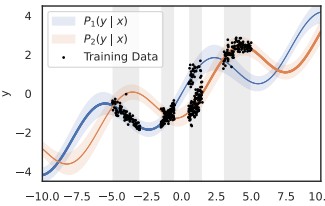 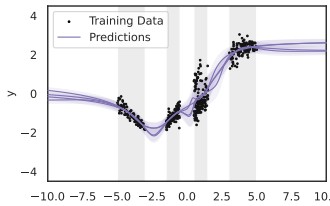 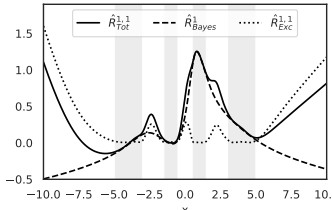

Figure 2: Uncertainty with the logarithmic scoring rule for the synthetic task. **Left:** The ground truth generative distribution is a mixture of two distributions. The thickness of the line indicates the weight of the mixture component. Intervals where training data is sampled are highlighted in gray. **Middle:** Predictive distributions of different models trained on the training data. **Right:** Resulting Total, Bayes and Excess risks.

row). It is expected that all uncertainty measures increase under such a shift. However, we observe that excess risk measures either decrease or stay the same (for $\widehat{R}_{\mathrm{Exc}}^{3a,2}$ under CRPS and Quadratic and for all excess risks under SE score). Here, staying the same is a desirable outcome over decreasing.

**Scale shift of predicted means.** The distribution of predicted means shifts to $\mu \sim \mathcal{U}(-2, 2)$, thus the scale of the posterior shifts, but not its location. The results are shown in Figure 1 (third row). It is expected that all uncertainty measures increase under such a shift. We observe that $\widehat{R}_{\mathrm{Bayes}}^{1}$ and $\widehat{R}_{\mathrm{Bayes}}^{3b}$ under all scores are invariant under such a shift, the same for $\widehat{R}_{\mathrm{Tot}}^{3b,2}$ under SE. This could reduce their viability in downstream tasks. All other measures increase under the scale shift in the predicted means as expected.

**Scale shift of predicted variances.** The distribution of predicted variances shifts to $\sigma^2 \sim \mathcal{U}(0.5, 2.5)$, thus the scale of the posterior shifts, but not its location. The results are shown in Figure 1 (fourth row). It is expected that total and epistemic uncertainty increase under such a shift, but that the aleatoric uncertainty stays the same.

**General observations.** Across all shifts, we observe that $\widehat{R}_{\mathrm{Exc}}^{3a,2}$ and $\widehat{R}_{\mathrm{Exc}}^{3b,2}$ have very low magnitudes, and for the SE score, they are equal to zero. Furthermore, the magnitude of the excess risks is often much lower than the Bayes risks, and they scale most with the mean scale shift, which is expected.

## 5.2 SYNTHETIC DATASETS

Having characterized how the measures react to controlled shifts of the posterior, we now turn to a synthetic data example that verifies these behaviors against common intuition. Specifically, inputs far from the training distribution should have greater epistemic uncertainty, while regions where the data-generating process is inherently ambiguous should yield higher aleatoric uncertainty.

In our 1D regression experiment (see results in Figure 2 and its description in Appendix E.2), the *excess risk* estimate, a proxy for epistemic uncertainty, increases for inputs outside the training data support, where the model cannot reliably infer the underlying dependency. The *Bayes risk* estimate, capturing aleatoric uncertainty, is largest where the mapping from $x$ to $y$ is least determined. In our synthetic example, this occurs in the region where the probabilities of sampling from the two generating curves are similar. Consequently, the *total risk* is high whenever either component is high.

For clarity, we present a Log-score instantiation (which leads to entropy-based measures) in the main text and visualize the corresponding approximations of Bayes, excess, and total risk in Figure 2. Log-score-based uncertainty measures are widely used in practice (Bülte et al., 2025), and our empirical results support this common choice. We also observe the same qualitative behavior for other instantiations, with additional visualizations provided in Appendix E.2.

## 5.3 SELECTIVE PREDICTION

In selective prediction, the model's performance is evaluated on a specific subset that is defined as having low uncertainty. Therefore, the uncertainty measures the ability to indicate whether or not the prediction is correct. We considered multiple datasets for these experiments; see Appendix E.3

Table 4: Results for selective prediction (PRR ↓) for different scoring rules (SR), averaged over datasets. The best result is underlined, all within its standard deviation in bold.

| SR | $\hat{R}_{\text{Tot}}^{1,1}$ | $\hat{R}_{\text{Tot}}^{2,1}$ | $\hat{R}_{\text{Tot}}^{3a,1}$ | $\hat{R}_{\text{Tot}}^{3b,1}$ | $\hat{R}_{\text{Tot}}^{3a,2}$ | $\hat{R}_{\text{Tot}}^{3b,2}$ | $\hat{R}_{\text{Bayes}}^{1}$ | $\hat{R}_{\text{Bayes}}^{2}$ | $\hat{R}_{\text{Bayes}}^{3a}$ | $\hat{R}_{\text{Bayes}}^{3b}$ | $\hat{R}_{\text{Exc}}^{1,1}$ | $\hat{R}_{\text{Exc}}^{2,1}$ | $\hat{R}_{\text{Exc}}^{3a,1}$ | $\hat{R}_{\text{Exc}}^{3b,1}$ | $\hat{R}_{\text{Exc}}^{3a,2}$ | $\hat{R}_{\text{Exc}}^{3b,2}$ |
|---|---|---|---|---|---|---|---|---|---|---|---|---|---|---|---|---|
| CRPS | **0.318** ±.007 | **0.318** ±.007 | **0.319** ±.007 | 0.327 ±.007 | 0.327 ±.007 | 0.352 ±.007 | 0.356 ±.007 | 0.327 ±.007 | 0.327 ±.007 | 0.357 ±.007 | 0.339 ±.006 | 0.339 ±.006 | 0.340 ±.006 | 0.341 ±.006 | 0.504 ±.006 | 0.378 ±.006 |
| Log | **0.323** ±.007 | **0.323** ±.007 | **0.321** ±.007 | 0.327 ±.007 | - | - | 0.356 ±.007 | - | 0.327 ±.007 | 0.357 ±.007 | 0.397 ±.006 | 0.397 ±.006 | 0.400 ±.006 | 0.398 ±.007 | - | - |
| SE | **0.320** ±.007 | **0.320** ±.007 | **0.320** ±.007 | 0.327 ±.007 | 0.327 ±.007 | 0.357 ±.007 | 0.357 ±.007 | 0.327 ±.007 | 0.327 ±.007 | 0.357 ±.007 | **0.325** ±.006 | **0.325** ±.006 | **0.325** ±.006 | **0.325** ±.006 | - | - |
| Quad. | **0.319** ±.007 | **0.319** ±.007 | **0.323** ±.007 | 0.327 ±.007 | 0.329 ±.007 | 0.347 ±.007 | 0.356 ±.007 | 0.326 ±.007 | 0.327 ±.007 | 0.357 ±.007 | 0.514 ±.007 | 0.514 ±.007 | 0.523 ±.006 | 0.511 ±.007 | 0.689 ±.004 | 0.503 ±.007 |

Table 5: Results for out-of-distribution detection (AUROC ↑) for different scoring rules (SR), averaged over datasets. The best result is underlined, all within its standard deviation in bold.

| SR | $\hat{R}_{\text{Tot}}^{1,1}$ | $\hat{R}_{\text{Tot}}^{2,1}$ | $\hat{R}_{\text{Tot}}^{3a,1}$ | $\hat{R}_{\text{Tot}}^{3b,1}$ | $\hat{R}_{\text{Tot}}^{3a,2}$ | $\hat{R}_{\text{Tot}}^{3b,2}$ | $\hat{R}_{\text{Bayes}}^{1}$ | $\hat{R}_{\text{Bayes}}^{2}$ | $\hat{R}_{\text{Bayes}}^{3a}$ | $\hat{R}_{\text{Bayes}}^{3b}$ | $\hat{R}_{\text{Exc}}^{1,1}$ | $\hat{R}_{\text{Exc}}^{2,1}$ | $\hat{R}_{\text{Exc}}^{3a,1}$ | $\hat{R}_{\text{Exc}}^{3b,1}$ | $\hat{R}_{\text{Exc}}^{3a,2}$ | $\hat{R}_{\text{Exc}}^{3b,2}$ |
|---|---|---|---|---|---|---|---|---|---|---|---|---|---|---|---|---|
| CRPS | 0.794 ±.012 | 0.794 ±.012 | 0.794 ±.012 | 0.779 ±.013 | 0.777 ±.012 | 0.727 ±.009 | 0.714 ±.010 | 0.777 ±.012 | 0.777 ±.012 | 0.714 ±.010 | **0.825** ±.010 | **0.825** ±.010 | **0.825** ±.010 | **0.825** ±.010 | 0.795 ±.006 | **0.826** ±.010 |
| Log | 0.802 ±.011 | 0.802 ±.011 | 0.797 ±.012 | 0.785 ±.013 | - | - | 0.713 ±.010 | - | 0.777 ±.012 | 0.714 ±.010 | **0.827** ±.011 | **0.827** ±.011 | **0.826** ±.011 | **0.826** ±.011 | - | - |
| SE | 0.792 ±.012 | 0.792 ±.012 | 0.792 ±.012 | 0.777 ±.012 | 0.777 ±.012 | 0.714 ±.010 | 0.714 ±.010 | 0.777 ±.012 | 0.777 ±.012 | 0.714 ±.010 | **0.820** ±.010 | **0.820** ±.010 | **0.820** ±.010 | **0.820** ±.010 | - | - |
| Quad. | 0.800 ±.012 | 0.800 ±.012 | 0.800 ±.012 | 0.784 ±.013 | 0.777 ±.013 | 0.740 ±.011 | 0.713 ±.010 | 0.777 ±.012 | 0.777 ±.012 | 0.714 ±.010 | **0.816** ±.016 | **0.816** ±.016 | **0.816** ±.016 | **0.817** ±.015 | 0.783 ±.005 | **0.822** ±.012 |

Table 6: Results for active learning (average rank ↓ over datasets and five seeds using different scores as acquisition functions). The best result is underlined, all within its standard deviation in bold.

| SR | $\hat{R}_{\text{Tot}}^{1,1}$ | $\hat{R}_{\text{Tot}}^{2,1}$ | $\hat{R}_{\text{Tot}}^{3a,1}$ | $\hat{R}_{\text{Tot}}^{3b,1}$ | $\hat{R}_{\text{Tot}}^{3a,2}$ | $\hat{R}_{\text{Tot}}^{3b,2}$ | $\hat{R}_{\text{Bayes}}^{1}$ | $\hat{R}_{\text{Bayes}}^{2}$ | $\hat{R}_{\text{Bayes}}^{3a}$ | $\hat{R}_{\text{Bayes}}^{3b}$ | $\hat{R}_{\text{Exc}}^{1,1}$ | $\hat{R}_{\text{Exc}}^{2,1}$ | $\hat{R}_{\text{Exc}}^{3a,1}$ | $\hat{R}_{\text{Exc}}^{3b,1}$ | $\hat{R}_{\text{Exc}}^{3a,2}$ | $\hat{R}_{\text{Exc}}^{3b,2}$ | Random |
|---|---|---|---|---|---|---|---|---|---|---|---|---|---|---|---|---|---|
| CRPS | 16.60 ±3.57 | 16.60 ±3.57 | 16.57 ±4.31 | 17.83 ±3.26 | 17.60 ±4.19 | 18.27 ±2.75 | 19.07 ±4.38 | 16.60 ±2.75 | 17.70 ±3.64 | 19.20 ±3.04 | 17.20 ±3.58 | 17.20 ±3.58 | 15.90 ±3.68 | 15.07 ±4.01 | 15.27 ±3.00 | 15.87 ±3.53 | 20.50 ±5.89 |
| Log | 15.73 ±2.63 | 15.73 ±2.63 | 14.97 ±2.38 | 15.47 ±3.60 | - | - | 19.13 ±2.20 | - | 17.97 ±3.61 | 21.47 ±2.55 | **13.90** ±1.33 | **13.90** ±1.33 | 17.33 ±5.97 | **14.37** ±3.93 | - | - | 20.50 ±5.89 |
| SE | 16.90 ±4.80 | 16.90 ±4.80 | 16.90 ±4.80 | 16.87 ±5.26 | 16.67 ±5.03 | 16.47 ±2.91 | 16.47 ±2.91 | 16.67 ±5.03 | 16.67 ±5.03 | 16.47 ±2.91 | 17.50 ±3.67 | 17.50 ±3.67 | 16.00 ±5.21 | 16.00 ±5.21 | - | - | 20.50 ±5.89 |
| Quad. | 29.23 ±2.06 | 29.23 ±2.06 | 32.57 ±2.34 | 29.47 ±5.12 | 31.70 ±1.93 | 31.27 ±4.23 | 31.07 ±3.07 | 32.47 ±1.10 | 30.33 ±3.81 | 31.83 ±4.17 | 16.10 ±3.37 | 16.10 ±3.37 | **12.60** ±2.19 | 13.43 ±3.93 | 17.00 ±2.34 | 15.90 ±2.46 | 20.50 ±5.89 |

for details. Empirical findings for selective prediction in classification (e.g., Kotelevskii et al., 2025; Schweighofer et al., 2025) suggest that total uncertainty performs well in this task, which we seek to validate for the regression setting as well.

The results are shown in Table 4, which are averages over all the considered datasets (see Table 9 for detailed results). Performances are measured as prediction-reject ratios (PRRs), where a lower PRR indicates more effective rejection of inaccurate predictions using the uncertainty score (details in Appendix E.3). The best performing scores are $\hat{R}_{\text{Tot}}^{1,1}$, $\hat{R}_{\text{Tot}}^{2,1}$ and $\hat{R}_{\text{Tot}}^{3a,1}$ for all considered scoring rules. Furthermore, all excess risks under the SE score, which are equivalent to each other, perform similarly well as the total risks. Noteworthy, excess risks for the quadratic score are performing very poorly compared to all other measures; the same for $\hat{R}_{\text{Exc}}^{3a,2}$ for the CRPS scoring rule. Furthermore, excess risks under the widely considered Log-score perform slightly worse than most other measures. In general, we recommend using $\hat{R}_{\text{Tot}}^{1,1}$ for selective prediction. It leads to the overall best performance under CRPS, and is consistently among the best across scoring rules.

### 5.4 OUT-OF-DISTRIBUTION DETECTION

Out-of-distribution (OOD) detection is widely considered as task for uncertainty estimation. We want to detect OOD inputs, as there is no guarantee that a model trained on some in-distribution (ID) dataset will perform well on it. An uncertainty score often indicates that a new input is OOD, assigning high uncertainty to such inputs. We considered a dataset consisting of a mosaic of four MNIST (Lecun et al., 1998) images, where the target is given as the number formed by those four digits, as an ID dataset. We considered multiple OOD datasets; for details, see Appendix E.4.

This task is evaluated using the AUROC of the uncertainty score to distinguish between ID and OOD data. Results are provided in Table 5, averaged over all considered ID and OOD data pairs. We refer to Table 10 in the Appendix for detailed results per ID/OOD pair. We find that excess risk measures are generally the most suitable for this task, particularly $\hat{R}_{\text{Exc}}^{1,1}$. This aligns with the assumptions on excess risk in prior work (Lahlou et al., 2023; Hofman et al., 2024; Kotelevskii et al., 2025).

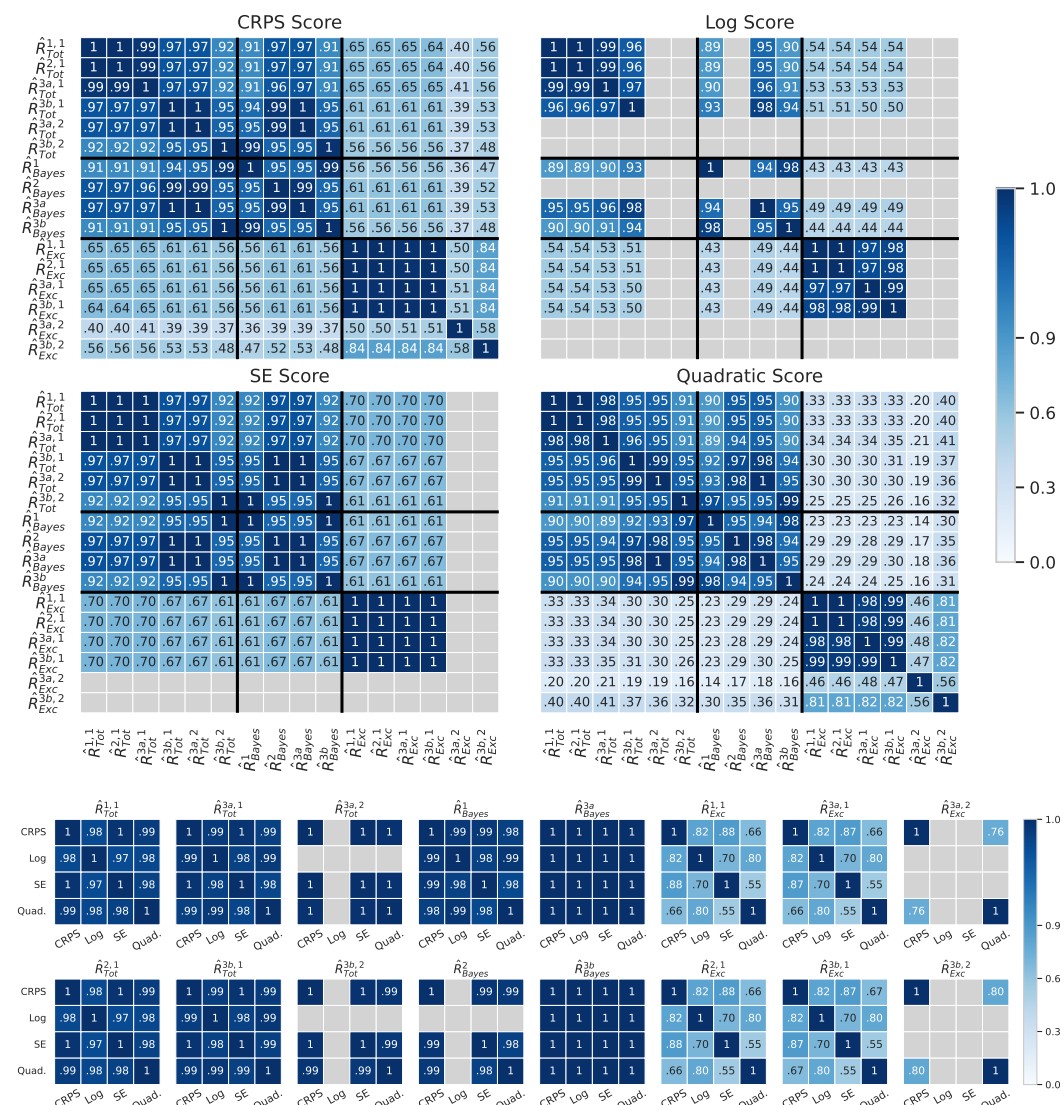

Figure 3: Kendall's $\tau_b$ rank correlation between different risk approximations for different scoring rules (top row) and between different scoring rules for the considered risk approximations (bottom rows). Correlations are averaged over all considered datasets.

## 5.5 ACTIVE LEARNING

Another widely considered real-world task utilizing uncertainty information is active learning. The measure of uncertainty should be used to guide which samples are most uncertain, and thus interesting to obtain labels to add to the training dataset. Generally, excess risks (epistemic uncertainty) are considered to be better guidance than total or Bayes risks (Mukhoti et al., 2023).

The results are provided in Table 6, showing average ranks over the considered datasets; see Appendix E.5 for details. As expected, we observe that excess risks are the best performing measures, but there is no clear best approximation strategy. Most outperform random sampling, yet there are some exceptions: the total and Bayes risks under the quadratic score and $\widehat{R}_{\text{Bayes}}^{3b}$ under the log score.

## 5.6 RANK CORRELATION BETWEEN MEASURES

Finally, we inspect the rank correlation between the measures within our framework. We use the same datasets and models as for selective prediction and calculate the rank correlations induced by

different uncertainty measures. The results shown in Figure 3 consider the rank correlation between different risk approximations per scoring rule, as well as the rank correlations between scoring rules given a risk approximation. The most important findings are, that $\widehat{R}_{\text{Bayes}}^{1}$ is very strongly correlated to $\widehat{R}_{\text{Bayes}}^{3b}$ and the same for $\widehat{R}_{\text{Bayes}}^{2}$ and $\widehat{R}_{\text{Bayes}}^{3a}$. Furthermore, $\widehat{R}_{\text{Tot}}^{1,1}$ and $\widehat{R}_{\text{Tot}}^{2,1}$ are equivalent due to the linearity in the expectation, the same holds for $\widehat{R}_{\text{Exc}}^{1,1}$ and $\widehat{R}_{\text{Exc}}^{2,1}$. Regarding differences between induced rankings for different scoring rules for the same approximations, a striking finding is that for both $\widehat{R}_{\text{Bayes}}^{3a}$ and $\widehat{R}_{\text{Bayes}}^{3b}$, all scoring rules are perfectly correlated, the same for $\widehat{R}_{\text{Tot}}^{3a,2}$ for those scoring rules it is computable for. Furthermore, rank correlations are overall very high among approximations of total and Bayes risks and only substantially vary across excess risks. Although total and Excess risk estimates for the scores are not perfectly uncorrelated, performance on the previous tasks clearly shows the usefulness of utilizing different risks for specific tasks. A recent benchmark of Mucsányi et al. (2024) provides a perspective on this in classification scenarios.

## 6 Discussion

**General findings and failure modes.** Throughout our considered experiments, we found that the particular choice of proper scoring rule and approximation strategy do not have a major impact in that they would lead to much worse results. The exception is $\bar{R}_{\text{Exc}}^{3a,2}$, which we found performs significantly worse throughout all considered experiments. Furthermore, this particular risk approximation was found to have undesirable characteristics across scoring rules, such as invariance under mean scale shift, linking theoretical behavior to downstream performance. We hypothesize, that this is caused by the fact that $\bar{R}_{\text{Exc}}^{3a,2}$ uses two approximation strategies where the disagreement between individual models is averaged out. Additionally, we found that no measure of Bayes risk, regardless of scoring rule or approximation strategy, is best in any of the considered tasks.

**Recommendations.** Our empirical findings in Section 5 show that different risks are desirable as measures in different tasks, leading to the following recommendations. For selective prediction, we find that total risk leads to the best results across particular scoring rules and approximation strategies. For both out-of-distribution detection and active learning, we find that excess risks perform best. While most scoring rules and Bayesian approximation strategies give reasonable results, we recommend $\bar{R}_{\text{Tot}}^{1,1}$ and $\bar{R}_{\text{Exc}}^{1,1}$ for any of the considered scoring rules as a default choice.

## 7 Conclusion

We presented a unified framework for uncertainty quantification in regression based on proper scoring rules. By extending the risk-based decomposition of uncertainty from classification to regression, we derived principled measures of total, aleatoric, and epistemic uncertainty that generalize and subsume existing variance- and entropy-based approaches. Our framework admits closed-form expressions under common assumptions, provides flexible Bayesian and surrogate approximations, and yields practical estimators suitable for deep ensembles.

Through a broad empirical evaluation, we demonstrated that these measures behave consistently under controlled perturbations, align with theoretical desiderata, and provide competitive or superior performance in uncertainty related tasks. Importantly, we highlighted both commonalities and differences across scoring rules, offering guidance for choosing uncertainty measures in practice.

In this work, we demonstrate our general framework under the Gaussian parametric assumption, using Deep Ensembles to compute MC approximations of posterior expectations to estimate the risks. We chose this particular instantiation due to their widespread use in practice. Looking into the possibility to derive closed-form solutions under different scoring rules for different parametric assumptions and using different approaches for Bayesian approximation, together with investigating their empirical properties, is an important direction for future work.

Overall, our work bridges the gap between theoretical considerations and practical deep learning methods, providing a principled and versatile foundation for regression uncertainty quantification.

## REPRODUCIBILITY STATEMENT

We provide the full code to reproduce our experiments as supplementary material and will release it publicly upon acceptance. Furthermore, we added detailed descriptions of all experiments in Appendix E. All experiments were conducted on publicly available datasets or datasets we created ourselves, which will be released alongside the code. We ran experiments with multiple seeds, if applicable, and report summary statistics.

## USAGE OF LARGE LANGUAGE MODELS (LLMS)

LLMs were used as a general-purpose assistive tool during the preparation of this paper. Their usage fell into two categories: (i) for writing assistance, they helped improve clarity and readability of certain passages through language refinement and (ii) for coding assistance, where they provided support with code completion and debugging. LLMs were not used for research ideation, experimental design, theoretical development, or analysis of results. All substantive contributions, including the conception of ideas, methodology, and experiments, were made by the authors.

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

## A    RISK ESTIMATION FOR PROPER SCORING RULES

This appendix develops the general machinery for estimating pointwise risk under proper scoring rules in regression. It formalizes Bayes/total/excess risks and derives tractable approximation for CRPS, logarithmic, quadratic, and squared error scores under a Gaussian assumption.

### A.1    BAYESIAN RISK ESTIMATION

This section specifies Bayesian estimators for the three risk components, showing how to combine posterior averaging, posterior-predictive (mixture) plug-ins, and Gaussian surrogates to obtain computable expressions.

#### A.1.1    GAUSSIAN ENSEMBLE MODEL

We will consider a specific tractable example where our Bayesian model is a finite Gaussian ensemble. This model arises e.g. when we train multiple neural network copies $\left\{\widehat{P}_i\right\}_{i=1}^M$ that predict Gaussian parameters starting from different weights initializations and is sometimes called *deep ensembles* (Lakshminarayanan et al., 2017):

$$\widehat{P}_i = \mathcal{N}(\mu_i, \sigma_i^2), \quad \mu_i \in \mathbb{R}, \sigma_i \in \mathbb{R}_+, i = 1, \ldots, M.$$

We can interpret this as a finite i.i.d. sample drawn from a posterior distribution over parameters of a one-dimensional Gaussian. Our prediction is the posterior predictive, which equals the full Gaussian mixture distribution:

$$\widehat{P}_{\text{ens}} = \frac{1}{M} \sum_{i=1}^M \mathcal{N}\left(\mu_i, \sigma_i^2\right), \quad \widehat{p}_{\text{ens}}(y) = \frac{1}{M} \sum_{i=1}^M \widehat{p}_i(y).$$

#### A.1.2    ENSEMBLE APPROXIMATIONS WITH A SINGLE GAUSSIAN

Previous work on ensembles (Lakshminarayanan et al., 2017) has used the following Gaussian approximation to the full mixture distribution:

$$\widehat{P}_* = \mathcal{N}(\mu_*, \sigma_*^2), \quad \mu_* = \frac{1}{M} \sum_{i=1}^M \mu_i, \quad \sigma_*^2 = \frac{1}{M} \sum_{i=1}^M (\sigma_i^2 + \mu_i^2) - \mu_*^2, \tag{5}$$

where $\mu_*$ and $\sigma_*^2$ are the mean and variance of the full mixture distribution. Another, more simple Gaussian approximation is to set the variance equal to the average variance of the components (Bülte et al., 2025):

$$\bar{P}_* = \mathcal{N}(\mu_*, \bar{\sigma}_*^2), \quad \mu_* = \frac{1}{M} \sum_{i=1}^M \mu_i, \quad \bar{\sigma}_*^2 = \frac{1}{M} \sum_{i=1}^M \sigma_i^2. \tag{6}$$

### A.2    TRACTABLE RISK ESTIMATION USING GAUSSIAN ENSEMBLES

We assume that we only have access to a single Gaussian ensemble model and propose to use in to approximate both $\widehat{P}$ and $P$. Based on the results of the previous section, we arrive at the following four approximations: (1) Bayesian risk averaging, (2) full mixture distribution $\widehat{P}_{\text{ens}}$, and (3) $\widehat{P}_*$. A specific risk estimate can thus be denoted as $\widehat{R}^{i,j}$, $i, j = 1, \ldots, 3$. These cases roughly correspond to the ones for classification that appeared in (Kotelevskii et al., 2025). We summarize them in Table 7, where empty cells correspond to combinations that do not seem particularly useful (approximation for ground truth should be at least as complex as for the prediction). In what follows, we will provide expressions for these estimates for different proper scoring rules.

#### A.2.1    CRPS SCORE

The notation and derivations of various quantities related to CRPS for the ensemble model can be found in the Appendix B.1. In particular, we will use the following notation:

$$A(\mu, \sigma) := 2\sigma\phi\left(\frac{\mu}{\sigma}\right) + \mu\left[2\Phi\left(\frac{\mu}{\sigma}\right) - 1\right], \quad \mu_{ij} := \mu_i - \mu_j, \quad \sigma_{ij} := \sqrt{\sigma_i^2 + \sigma_j^2}.$$

Table 7: Chosen combinations of estimates for excess risk estimation.

| $\downarrow \widehat{P}, \rightarrow P$ | BA | $\widehat{P}_{\text{ens}}$ | $\widehat{P}_*$ |
|---|---|---|---|
| BA | $\widehat{R}^{1,1}$ | - | - |
| $\widehat{P}_{\text{ens}}$ | $\widehat{R}^{2,1}$ | - | - |
| $\widehat{P}_*$ | $\widehat{R}^{3,1}$ | $\widehat{R}^{3,2}$ | - |

**Bayes risk.** Since Bayes risk is equal to the entropy of the true distribution $H(P)$, we only need to choose an approximation scheme for $P$:

- **Bayesian averaging**: $\widehat{R}^1_{\text{Bayes}} = \frac{1}{M} \sum_{i=1}^M H(\widehat{P}_i) = \frac{1}{M\sqrt{\pi}} \sum_{i=1}^M \sigma_i$.

- **Posterior predictive**: $\widehat{R}^2_{\text{Bayes}} = H(\widehat{P}_{\text{ens}}) = \frac{1}{2} \frac{1}{M^2} \sum_{i=1}^M \sum_{j=1}^M A(\mu_{ij}, \sigma_{ij})$.

- **Gaussian approximation**: $\widehat{R}^3_{\text{Bayes}} = H(\widehat{P}_*) = \sqrt{\frac{\sigma_*^2}{\pi}}$.

**Excess risk.** Since the divergence function for CRPS is symmetric, many of $\widehat{R}^{i,j}_{\text{Exc}}$ quantities coincide. Here we present all possible combinations of approximations for the Excess risk in the Gaussian ensemble model.

- **Bayesian averaging for both**. Also known as *expected pairwise Bregman divergence*.

$$\widehat{R}^{1,1}_{\text{Exc}} = \frac{1}{M^2} \sum_{i=1}^M \sum_{j=1}^M d(\widehat{P}_i, \widehat{P}_j) = \frac{1}{M^2} \sum_{i=1}^M \sum_{j=1}^M \left[ A(\mu_{ij}, \sigma_{ij}) - \frac{\sigma_i + \sigma_j}{\sqrt{\pi}} \right] \tag{7}$$

$$= \frac{1}{M^2} \sum_{i=1}^M \sum_{j=1}^M A(\mu_{ij}, \sigma_{ij}) - \frac{2}{M} \sum_{i=1}^M \frac{\sigma_i}{\sqrt{\pi}}. \tag{8}$$

- **Gaussian mixture and Bayesian averaging**. This approximation is also referred to as *Bregman information* (and *reverse Bregman information* for non-symmetric divergences).

$$\widehat{R}^{2,1}_{\text{Exc}} = \widehat{R}^{1,2}_{\text{Exc}} = \frac{1}{M} \sum_{i=1}^M d(\widehat{P}_i, \widehat{P}_{\text{ens}}) = \frac{1}{M} \sum_{i=1}^M \left[ \text{CRPS}(\widehat{P}_i, \widehat{P}_{\text{ens}}) - H(\widehat{P}_{\text{ens}}) \right] \tag{9}$$

$$= \frac{1}{2} \frac{1}{M^2} \sum_{i=1}^M \sum_{j=1}^M A(\mu_{ij}, \sigma_{ij}) - \frac{1}{M} \sum_{i=1}^M \frac{\sigma_i}{\sqrt{\pi}}. \tag{10}$$

- **Moment-matched Gaussian approximation and Bayesian averaging**:

$$\widehat{R}^{3,1}_{\text{Exc}} = \widehat{R}^{1,3}_{\text{Exc}} = \frac{1}{M} \sum_{j=1}^M d(\widehat{P}_*, \widehat{P}_j) = \frac{1}{M} \sum_{j=1}^M \left[ A(\mu_{*j}, \sigma_{*j}) - \frac{\sigma_* + \sigma_j}{\sqrt{\pi}} \right] \tag{11}$$

$$= \frac{1}{M} \sum_{j=1}^M A(\mu_{*j}, \sigma_{*j}) - \frac{\sigma_*}{\sqrt{\pi}} - \frac{\frac{1}{M} \sum_{j=1}^M \sigma_j}{\sqrt{\pi}}. \tag{12}$$

- **Moment-matched Gaussian approximation and mixture**:

$$\widehat{R}^{3,2}_{\text{Exc}} = \widehat{R}^{2,3}_{\text{Exc}} = d(\widehat{P}_*, \widehat{P}_{\text{ens}}) = \text{CRPS}(\widehat{P}_*, \widehat{P}_{\text{ens}}) - H(\widehat{P}_{\text{ens}}) \tag{13}$$

$$= \frac{1}{M} \sum_{j=1}^M A(\mu_{*j}, \sigma_{*j}) - \frac{\sigma_*}{\sqrt{\pi}} - H(\widehat{P}_{\text{ens}}). \tag{14}$$

We notice a peculiar relation: $2\widehat{R}^{2,1}_{\text{Exc}} = \widehat{R}^{1,1}_{\text{Exc}}$. Similar results were obtained for a score with symmetric divergence in (Kotelevskii et al., 2025) in the case of classification.

### A.2.2 LOGARITHMIC SCORE

The logarithmic score is a proper scoring rule for both discrete and continuous distributions, which is defined as follows:

$$\text{LS}(\widehat{P}, y) = -\log \widehat{p}(y), \tag{15}$$

where $\widehat{p}(y)$ is the probability density function of the distribution $\widehat{P}$ evaluated at $y$. This is an example of a *local scoring rule*: it depends only on the density value at the point $y$ and does not take into account the entire distribution.

The entropy function for the logarithmic score is the negative Shannon entropy:

$$H(P) = \text{LS}(P, P) = \int_{\mathbb{R}} \text{LS}(P, y)\, p(y)\, dy = -\int_{\mathbb{R}} p(y) \log p(y)\, dy. \tag{16}$$

The divergence function of the logarithmic score is the well-known Kullback-Leibler divergence:

$$d(\widehat{P}, P) = \int_{\mathbb{R}} \left[ -\log \widehat{p}(y) \right] p(y)\, dy - \int_{\mathbb{R}} \left[ -\log p(y) \right] p(y)\, dy = \int_{\mathbb{R}} p(y) \log \frac{p(y)}{\widehat{p}(y)}\, dy = D_{\text{KL}}(P \,\|\, \widehat{P}). \tag{17}$$

KL-divergence between two Gaussian mixtures can not be expressed in a closed form, but it can be approximated with MC methods or bounded, see (Hershey & Olsen, 2007; Durrieu et al., 2012). In particular, the work (Durrieu et al., 2012) provides lower and upper bounds for the KL divergence between two Gaussian mixtures using variational approximations.

Detailed derivations of the quantities presented in the following are deferred to Section B.2.

**Bayes risk.** For the Gaussian ensemble model, we can derive the following approximations of the Bayes risk for the log score:

- $\widehat{R}^1_{\text{Bayes}} = \frac{1}{M} \sum_{i=1}^{M} H(\widehat{P}_i) = \frac{1}{2M} \sum_{i=1}^{M} \log \left( 2\pi e \sigma_i^2 \right).$

- $\widehat{R}^2_{\text{Bayes}} = H(\widehat{P}_{\text{ens}}) = -\int_{\mathbb{R}} \log \left( \frac{1}{M} \sum_{i=1}^{M} \widehat{p}_i(y) \right) \cdot \left( \frac{1}{M} \sum_{i=1}^{M} \widehat{p}_i(y) \right) dy$ – this expression is known to have no closed form. Using Jensen's inequality, we can derive a lower bound:

$$\widehat{R}^2_{\text{Bayes}} \geq \frac{1}{M} \sum_{i=1}^{M} H(\widehat{P}_i) = \widehat{R}^1_{\text{Bayes}}.$$

- $\widehat{R}^3_{\text{Bayes}} = H(\widehat{P}_*) = \frac{1}{2} \log \left( 2\pi e \sigma_*^2 \right).$

**Excess risk.** In the case of the Gaussian ensemble, we can use the following approximations:

- **Bayesian averaging for both**:

$$\widehat{R}^{1,1}_{\text{Exc}} = \frac{1}{M^2} \sum_{i=1}^{M} \sum_{j=1}^{M} d(\widehat{P}_i, \widehat{P}_j) = \frac{1}{2M^2} \sum_{i=1}^{M} \sum_{j=1}^{M} \left[ \frac{\sigma_j^2 + (\mu_i - \mu_j)^2}{\sigma_i^2} - 1 \right]. \tag{18}$$

- **Gaussian mixture and Bayesian averaging**:

$$\widehat{R}^{2,1}_{\text{Exc}} = \frac{1}{M} \sum_{i=1}^{M} d(\widehat{P}_{\text{ens}}, \widehat{P}_i) = \frac{1}{M} \sum_{i=1}^{M} D_{\text{KL}}(\widehat{P}_i \,\|\, \widehat{P}_{\text{ens}}), \tag{19}$$

$$\widehat{R}^{1,2}_{\text{Exc}} = \frac{1}{M} \sum_{i=1}^{M} d(\widehat{P}_i, \widehat{P}_{\text{ens}}) = \frac{1}{M} \sum_{i=1}^{M} D_{\text{KL}}(\widehat{P}_{\text{ens}} \,\|\, \widehat{P}_i). \tag{20}$$

There is no nice analytical form for those since $D_{\text{KL}}$ contains the log of the mixture density. The expression for $\widehat{R}^{2,1}_{\text{Exc}}$ was used for discrete distributions in (Lakshminarayanan et al., 2017) as "disagreement".

- **Moment-matched Gaussian approximation and Bayesian averaging**:

$$\widehat{R}_{\text{Exc}}^{3,1} = \frac{1}{M} \sum_{i=1}^{M} d(\widehat{P}_*, \widehat{P}_i) = \frac{1}{2} \left[ \log\left(\sigma_*^2\right) - \frac{1}{M} \sum_{i=1}^{M} \log\left(\sigma_i^2\right) \right]. \tag{21}$$

Another, more simple approximation of the mixture is $\bar{P}_* = \mathcal{N}(\mu_*, \bar{\sigma}_*^2)$, where $\bar{\sigma}_*^2 = \frac{1}{M} \sum_{i=1}^{M} \sigma_i^2$. With this, we recover the approximation from (Bülte et al., 2025):

$$\bar{R}_{\text{Exc}}^{3,1} = \frac{1}{2} \left[ \log\left(\bar{\sigma}_*^2\right) - \frac{1}{M} \sum_{i=1}^{M} \log\left(\sigma_i^2\right) + \frac{1}{\bar{\sigma}_*^2 M} \sum_{i=1}^{M} (\mu_i - \mu_*)^2 \right].$$

- **Moment-matched Gaussian approximation and mixture**:

$$\widehat{R}_{\text{Exc}}^{3,2} = D_{\text{KL}}(\widehat{P}_{\text{ens}} \,\|\, \widehat{P}_*). \tag{22}$$

### A.2.3 QUADRATIC SCORE

In our results below, we will use the following notation:

$$\mathcal{N}\left(a \mid b, s^2\right) = \frac{1}{\sqrt{2\pi s^2}} \exp\left(-\frac{(a-b)^2}{2s^2}\right) = \frac{1}{\sqrt{\sigma_i^2 + \sigma_j^2}} \, \phi\left(\frac{\mu_i - \mu_j}{\sqrt{\sigma_i^2 + \sigma_j^2}}\right),$$

see Appendix B.3 for more details and derivations of the below quantities.

**Bayes risk.** For the Gaussian ensemble model, we can derive the following approximations of the Bayes risk for the quadratic score:

- $\widehat{R}_{\text{Bayes}}^1 = \frac{1}{M} \sum_{i=1}^{M} H(\widehat{P}_i) = -\frac{1}{2M\sqrt{\pi}} \sum_{i=1}^{M} \frac{1}{\sigma_i}$.

- $\widehat{R}_{\text{Bayes}}^2 = H(\widehat{P}_{\text{ens}}) = -\int_{\mathbb{R}} \left(\frac{1}{M} \sum_{i=1}^{M} \widehat{p}_i(y)\right)^2 dy = \frac{1}{2M} \sum_{i=1}^{M} H(\widehat{P}_i) + \frac{1}{2M^2} \sum_{i=1}^{M} \sum_{j=1}^{M} \text{QS}(\widehat{P}_i, \widehat{P}_j)$.
  Alternatively, we can express the entropy of the quadratic score for a Gaussian mixture as follows: $\widehat{R}_{\text{Bayes}}^2 = -\frac{1}{M^2} \sum_{i=1}^{M} \sum_{j=1}^{M} \mathcal{N}\left(\mu_i \mid \mu_j, \sigma_i^2 + \sigma_j^2\right)$.

- $\widehat{R}_{\text{Bayes}}^3 = H(\widehat{P}_*) = -\frac{1}{2\sqrt{\pi \sigma_*^2}}$.

**Excess risk.** The divergence is symmetric for the quadratic score, so the situation is analogous to the case of CRPS. We have two different approximations of Excess risk:

- **Expected Pairwise Bregman Divergence (EPBD)**: if we employ Bayesian approach for estimation both $P$ and $\widehat{P}$, we get the following approximation: $\widehat{R}_{\text{Exc}}^{1,1} = \frac{1}{M^2} \sum_{i=1}^{M} \sum_{j=1}^{M} d(\widehat{P}_i, \widehat{P}_j)$, where closed form of $d(\widehat{P}_i, \widehat{P}_j)$ is given by (see equation (71)):

$$d(\widehat{P}_i, \widehat{P}_j) = \frac{1}{2\sqrt{\pi}\sigma_i} + \frac{1}{2\sqrt{\pi}\sigma_j} - 2\mathcal{N}\left(\mu_i \mid \mu_j, \sigma_i^2 + \sigma_j^2\right).$$

We combine these results to get the final expression:

$$\widehat{R}_{\text{Exc}}^{1,1} = \underbrace{\frac{1}{M\sqrt{\pi}} \sum_{i=1}^{M} \frac{1}{\sigma_i}}_{-2\widehat{R}_{\text{Bayes}}^1} - \frac{2}{M^2} \sum_{i=1}^{M} \sum_{j=1}^{M} \mathcal{N}\left(\mu_i \mid \mu_j, \sigma_i^2 + \sigma_j^2\right).$$

Notice that the first part equal $-2\widehat{R}_{\text{Bayes}}^1$.

- **Bregman Information (BI)**: here we combine Bayesian averaging and the mixture distribution. Using equation (75) for the divergence between the mixture and its component we get:

$$
\widehat{R}_{\text{Exc}}^{2,1} = \widehat{R}_{\text{Exc}}^{1,2} = \frac{1}{M} \sum_{i=1}^{M} d(\widehat{P}_i, \widehat{P}_{\text{ens}})
$$

$$
= \frac{1}{M} \sum_{i=1}^{M} \left[ -\frac{2}{M} \left( \sum_{j=1}^{M} \mathcal{N}(\mu_i \mid \mu_j, \sigma_i^2 + \sigma_j^2) \right) - H(\widehat{P}_i) - H(\widehat{P}_{\text{ens}}) \right]
$$

$$
= \underbrace{\frac{1}{2M\sqrt{\pi}} \sum_{i=1}^{M} \frac{1}{\sigma_i}}_{-\widehat{R}_{\text{Bayes}}^{1}} - \frac{2}{M^2} \sum_{i=1}^{M} \sum_{j=1}^{M} \mathcal{N}(\mu_i \mid \mu_j, \sigma_i^2 + \sigma_j^2) - \widehat{R}_{\text{Bayes}}^2 = \frac{1}{2} \widehat{R}_{\text{Exc}}^{1,1}.
$$

- **Moment-matched Gaussian approximation and Bayesian averaging**:

$$
\widehat{R}_{\text{Exc}}^{3,1} = \widehat{R}_{\text{Exc}}^{1,3} = \frac{1}{M} \sum_{i=1}^{M} d(\widehat{P}_*, \widehat{P}_i)
$$

$$
= \frac{1}{M} \sum_{i=1}^{M} \left[ -H(\widehat{P}_i) - H(\widehat{P}_*) - 2\mathcal{N}(\mu_* \mid \mu_i, \sigma_*^2 + \sigma_i^2) \right]
$$

$$
= \underbrace{\frac{1}{2M\sqrt{\pi}} \sum_{i=1}^{M} \frac{1}{\sigma_i}}_{-\widehat{R}_{\text{Bayes}}^{1}} - \frac{2}{M} \sum_{i=1}^{M} \mathcal{N}(\mu_* \mid \mu_i, \sigma_*^2 + \sigma_i^2) + \underbrace{\frac{1}{2\sqrt{\pi \sigma_*^2}}}_{-\widehat{R}_{\text{Bayes}}^{3}}.
$$

- **Moment-matched Gaussian approximation and mixture**:

$$
\widehat{R}_{\text{Exc}}^{3,2} = \widehat{R}_{\text{Exc}}^{2,3} = d(\widehat{P}_*, \widehat{P}_{\text{ens}}) = -\frac{2}{M} \left( \sum_{j=1}^{M} \mathcal{N}(\mu_* \mid \mu_j, \sigma_*^2 + \sigma_j^2) \right) - \underbrace{H(\widehat{P}_*)}_{\widehat{R}_{\text{Bayes}}^{3}} - \underbrace{H(\widehat{P}_{\text{ens}})}_{\widehat{R}_{\text{Bayes}}^{2}}.
$$

### A.2.4 SE SCORE

We can use the same three approaches to estimate the risk components for the SE score as we did for other proper scoring rules. Due to the symmetry of the divergence, the central label and central prediction coincide again. They are equal to the full mixture distribution, which we again denote as $\widehat{P}_{\text{ens}}$. The mean and variance of this distribution are denoted as $\mu_*$ and $\sigma_*^2$. We have previously introduced them to construct our Gaussian approximation in equation (5). Here they are again for clarity:

$$
\mathbb{E}[\widehat{P}_{\text{ens}}] = \mu_* = \frac{1}{M} \sum_{i=1}^{M} \mu_i, \ \text{Var}[\widehat{P}_{\text{ens}}] = \sigma_*^2 = \frac{1}{M} \sum_{i=1}^{M} (\sigma_i^2 + \mu_i^2) - \mu_*^2.
$$

For a more detailed treatment of the SE score, please refer to Section C.

**Bayes risk.**

- **Bayesian averaging**: $\widehat{R}_{\text{Bayes}}^1 = \frac{1}{M} \sum_{i=1}^{M} H_{\text{SE}}(\widehat{P}_i) = \frac{1}{M} \sum_{i=1}^{M} \text{Var}_{Y \sim \widehat{P}_i}[Y] = \frac{1}{M} \sum_{i=1}^{M} \sigma_i^2$.
- **Posterior predictive**: $\widehat{R}_{\text{Bayes}}^2 = H_{\text{SE}}(\widehat{P}_{\text{ens}}) = \text{Var}_{Y \sim \widehat{P}_{\text{ens}}}[Y] = \sigma_*^2 = \frac{1}{M} \sum_{i=1}^{M} (\sigma_i^2 + \mu_i^2) - \mu_*^2$.
- **Gaussian approximation**: $\widehat{R}_{\text{Bayes}}^3 = H_{\text{SE}}(\widehat{P}_*) = \text{Var}_{Y \sim \widehat{P}_*}[Y] = \sigma_*^2 = \widehat{R}_{\text{Bayes}}^2$.

**Excess risk.** For the symmetric divergence of SE we get:

- **Bayesian averaging**: $\widehat{R}_{\text{Exc}}^{1,1} = \frac{1}{M^2} \sum_{i=1}^{M} \sum_{j=1}^{M} d_{\text{SE}}(\widehat{P}_i, \widehat{P}_j) = \frac{1}{M^2} \sum_{i=1}^{M} \sum_{j=1}^{M} (\mu_j - \mu_i)^2 = 2\widehat{\text{Var}}\,[\mu_i]$.

- **Posterior predictive and Bayesian averaging**: $\widehat{R}_{\text{Exc}}^{2,1} = \widehat{R}_{\text{Exc}}^{1,2} = \frac{1}{M} \sum_{i=1}^{M} d_{\text{SE}}(\widehat{P}_i, \widehat{P}_{\text{ens}}) = \frac{1}{M} \sum_{i=1}^{M} (\mathbb{E}_{X \sim \widehat{P}_i}[X] - \mathbb{E}_{Y \sim \widehat{P}_{\text{ens}}}[Y])^2 = = \frac{1}{M} \sum_{i=1}^{M} (\mu_i - \mu_*)^2 = \widehat{\text{Var}}\,[\mu_i]$.

- **Moment-matched Gaussian approximation and Bayesian averaging**: since SE score only looks at the mean of the distribution, using this Gaussian approximation will give the same result as the previous section.

$$\widehat{R}_{\text{Exc}}^{3,1} = \widehat{R}_{\text{Exc}}^{1,3} = \frac{1}{M} \sum_{i=1}^{M} (\mu_i - \mu_*)^2 = \widehat{\text{Var}}\,[\mu_i].$$

- **Moment-matched Gaussian approximation and posterior predictive**: the means of these distributions coincide so we get a zero.

$$\widehat{R}_{\text{Exc}}^{3,2} = \widehat{R}_{\text{Exc}}^{2,3} = 0.$$

## B  SCORE COMPUTATION AND TOOLS

### B.1  CRPS

Continuous ranked probability score has multiple equivalent representations:

$$\text{CRPS}(P, y) = \int_{\mathbb{R}} \big(F_P(t) - \mathbb{I}\{y \leq t\}\big)^2 dt. \tag{23}$$

$$\text{CRPS}(P, y) = \mathbb{E}_{X \sim P}|X - y| - \frac{1}{2}\mathbb{E}_{X, X' \sim P}|X - X'|. \tag{24}$$

$$\text{CRPS}(P, y) = \int |x - y|\, dP(x) - \frac{1}{2} \int \int |x - x'|\, dP(x)\, dP(x'). \tag{25}$$

Expected CRPS between two distributions $P$ and $Q$ can be expressed as:

$$\text{CRPS}(P, Q) = \mathbb{E}_{X \sim P, Y \sim Q}|X - Y| - \frac{1}{2}\mathbb{E}_{X, X' \sim P}|X - X'|. \tag{26}$$

#### B.1.1  ENTROPY FUNCTION OF CRPS

$$H(P) = \int \text{CRPS}(P, y)\, dP(y) = \int \left[ \mathbb{E}_{X \sim P}|X - y| - \frac{1}{2}\mathbb{E}_{X, X' \sim P}|X - X'| \right] dP(y)$$

$$= \int \mathbb{E}_{X \sim P}|X - y|\, dP(y) - \frac{1}{2}\mathbb{E}_{X, X' \sim P}|X - X'| = \frac{1}{2}\mathbb{E}_{X, X' \sim P}|X - X'|.$$

#### B.1.2  DIVERGENCE FUNCTION OF CRPS

$$d(P, Q) = \int \text{CRPS}(P, y)\, dQ(y) - H(Q)$$

$$= \int \left[ \mathbb{E}_{X \sim P}|X - y| - \frac{1}{2}\mathbb{E}_{X, X' \sim P}|X - X'| \right] dQ(y) - H(Q)$$

$$= \mathbb{E}_{X \sim P, Y \sim Q}|X - Y| - \frac{1}{2}\mathbb{E}_{X, X' \sim P}|X - X'| - \frac{1}{2}\mathbb{E}_{Y, Y' \sim Q}|Y - Y'|$$

$$= \mathbb{E}_{X \sim P, Y \sim Q}|X - Y| - H(P) - H(Q).$$

Another expression for the divergence function of CRPS is given by:

$$d(P, Q) = \int_{\mathbb{R}} \big(F_P(y) - F_Q(y)\big)^2 dy. \tag{27}$$

### B.1.3 CRPS FOR GAUSSIANS

**Notation.** Some commonly used notation:

$$\phi(t) = \frac{1}{\sqrt{2\pi}} e^{-\frac{t^2}{2}} \text{ is the standard normal density function.} \tag{28}$$

$$\Phi(z) = \int_{-\infty}^{z} \frac{1}{\sqrt{2\pi}} e^{-\frac{z^2}{2}} \text{ is the standard normal cumulative distribution function.} \tag{29}$$

We follow (Villez, 2017) and also introduce the following notation:

$$\mu_{ij} := \mu_i - \mu_j, \tag{30}$$

$$\sigma_{ij} := \sqrt{\sigma_i^2 + \sigma_j^2}, \tag{31}$$

$$A(\mu, \sigma) := 2\sigma\phi\left(\frac{\mu}{\sigma}\right) + \mu\left[2\Phi\left(\frac{\mu}{\sigma}\right) - 1\right]. \tag{32}$$

We note that for $X \sim \mathcal{N}(\mu, \sigma)$ it can be shown that $\mathbb{E}\left[|X|\right] = A(\mu, \sigma)$.

**CRPS for a Gaussian.**

$$\text{CRPS}\big(\mathcal{N}(\mu, \sigma), y\big) = \sigma\left(2\phi(z) + z\big(2\Phi(z) - 1\big) - \frac{1}{\sqrt{\pi}}\right), \tag{33}$$

where $z = \frac{y-\mu}{\sigma}$, $\phi(z)$ is PDF of the standard normal distribution, and $\Phi(z)$ is CDF of the standard normal distribution

*Proof.*

$$\text{CRPS}(\mathcal{N}(\mu, \sigma), y) = \int_{-\infty}^{+\infty} (F_{\mathcal{N}}(t) - \mathbb{I}\{y \le t\})^2 dt$$

$$= \int_{-\infty}^{y} (F_{\mathcal{N}}(t) - \mathbb{I}\{y \le t\})^2 dt + \int_{y}^{+\infty} (F_{\mathcal{N}}(t) - \mathbb{I}\{y \le t\})^2 dt$$

$$= \int_{-\infty}^{y} (F_{\mathcal{N}}(t) - 0)^2 dt + \int_{y}^{+\infty} (F_{\mathcal{N}}(t) - 1)^2 dt$$

$$= \int_{-\infty}^{y} F_{\mathcal{N}}^2(t) dt + \int_{y}^{+\infty} \big(1 - F_{\mathcal{N}}(t)\big)^2 dt = I_1 + I_2.$$

First we compute

$$I_1 = \int_{-\infty}^{y} F_{\mathcal{N}}^2(t) dt = \int_{-\infty}^{y} \Phi^2\left(\frac{t-\mu}{\sigma}\right) dt = \sigma \int_{-\infty}^{\frac{y-\mu}{\sigma}} \Phi^2(z) dz = \sigma \int_{-\infty}^{z_y} \Phi^2(z) dz,$$

where we used the substitution $z = \frac{t-\mu}{\sigma}$ and define $z_y = \frac{y-\mu}{\sigma}$. For the second part, we get

$$I_2 = \int_{y}^{+\infty} \big(1 - F_{\mathcal{N}}(t)\big)^2 dt = \int_{y}^{+\infty} \left(1 - \Phi\left(\frac{t-\mu}{\sigma}\right)\right)^2 dt$$

$$= \sigma \int_{\frac{y-\mu}{\sigma}}^{+\infty} \big(1 - \Phi(z)\big)^2 dz = \sigma \int_{z_y}^{+\infty} \big(1 - \Phi(z)\big)^2 dz.$$

Let us calculate the first part:

$$I_1 = \sigma \int_{-\infty}^{z_y} \Phi^2(z) dz = \begin{bmatrix} u = \Phi(z)^2 & dv = dz \\ du = 2\Phi(z)\phi(z) dz & v = z \end{bmatrix} = \sigma \left[ z\Phi^2(z)|_{-\infty}^{z_y} - \int_{-\infty}^{z_y} 2z\Phi(z)\phi(z) dz \right]$$

$$= \sigma \left[ z_y\Phi^2(z_y) - 2\int_{-\infty}^{z_y} z\Phi(z)\phi(z) dz \right] = \sigma \left[ z_y\Phi^2(z_y) + 2\int_{-\infty}^{z_y} z\Phi(z) \, d\phi(z) \right],$$

where we have used the fact that $\big(\phi(z)\big)'_z = -z\phi(z)$.

Next,

$$I_2 = \sigma \int_{z_y}^{+\infty} \left(1 - \Phi(z)\right)^2 dz = \begin{bmatrix} u = \left(1 - \Phi(z)\right)^2 & dv = dz \\ du = 2\left(\Phi(z) - 1\right)\phi(z)dz & v = z \end{bmatrix}$$

$$= \sigma \left[ z\left(1 - \Phi(z)\right)^2 \big|_{z_y}^{+\infty} - 2 \int_{z_y}^{+\infty} z\left(\Phi(z) - 1\right)\phi(z)dz \right]$$

$$= \sigma \left[ -z_y\left(1 - \Phi(z_y)\right)^2 + 2 \int_{z_y}^{+\infty} \left(\Phi(z) - 1\right)d\phi(z) \right]. \tag{34}$$

Finally,

$$\mathrm{CRPS}(\mathcal{N}(\mu,\sigma),y) = I_1 + I_2 = \sigma \left[ z_y\left(2\Phi(z_y) - 1\right) + 2\int_{-\infty}^{+\infty} \Phi(z)\,d\phi(z) - 2\int_{z_y}^{+\infty} d\phi(z) \right]$$

$$= \sigma \left[ z_y\left(2\Phi(z_y) - 1\right) + 2I_3 + 2\phi(z_y) \right]$$

$$= \sigma \left[ z_y\left(2\Phi(z_y) - 1\right) - \frac{1}{\sqrt{\pi}} + 2\phi(z_y) \right]. \tag{35}$$

We used the following identity:

$$I_3 = \int_{-\infty}^{+\infty} \Phi(z)\,d\phi(z) = \Phi(z)\phi(z)\big|_{-\infty}^{+\infty} - \int_{-\infty}^{+\infty} \phi^2(z)dz = -\frac{1}{2\sqrt{\pi}}.$$

$\square$

**Entropy of CRPS for a Gaussian.** If we use the alternative representation of CRPS, we can express it as the expected absolute value of a centered Gaussian:

$$X, X' \sim \mathcal{N}(\mu, \sigma^2) \implies X - X' \sim \mathcal{N}(0, 2\sigma^2)$$

$$\implies \mathbb{E}_{X,X'\sim P}\left|X - X'\right| = A(0, \sqrt{2}\sigma) = \sqrt{2}\sigma\sqrt{\frac{2}{\pi}} = \frac{2\sigma}{\sqrt{\pi}}. \tag{36}$$

$$H(\mathcal{N}(\mu, \sigma^2)) = \frac{1}{2}\mathbb{E}_{X,X'\sim P}\left|X - X'\right| = \frac{1}{2}\frac{2\sigma}{\sqrt{\pi}} = \frac{\sigma}{\sqrt{\pi}}. \tag{37}$$

**Divergence function of CRPS for two Gaussians.** Let $P = \mathcal{N}(\mu_i, \sigma_i^2)$ and $Q = \mathcal{N}(\mu_j, \sigma_j^2)$ be two Gaussian distributions. Recall that the divergence function of CRPS is given by:

$$d(P, Q) = \mathbb{E}_{X\sim P, Y\sim Q}\left|X - Y\right| - \frac{1}{2}\mathbb{E}_{X,X'\sim P}\left|X - X'\right| - \frac{1}{2}\mathbb{E}_{Y,Y'\sim Q}\left|Y - Y'\right|. \tag{38}$$

Since $X$ and $Y$ are independent, $X - Y \sim \mathcal{N}(\mu_i - \mu_j, \sigma_i^2 + \sigma_j^2)$. Using our notation, we get:

$$\mathbb{E}_{X\sim P, Y\sim Q}\left|X - Y\right| = A(\mu_{ij}, \sigma_{ij}). \tag{39}$$

We also note that $H(P) = \frac{\sigma_i}{\sqrt{\pi}}$ and $H(Q) = \frac{\sigma_j}{\sqrt{\pi}}$. Combining all these results, we can express the divergence function as follows:

$$d(P, Q) = A(\mu_{ij}, \sigma_{ij}) - \frac{\sigma_i + \sigma_j}{\sqrt{\pi}}. \tag{40}$$

**Expected CRPS between two Gaussians.**

$$\mathrm{CRPS}(P, Q) = \mathbb{E}_{X\sim P, Y\sim Q}\left|X - Y\right| - \frac{1}{2}\mathbb{E}_{X,X'\sim P}\left|X - X'\right|. \tag{41}$$

Let $P = \mathcal{N}(\mu_i, \sigma_i^2)$ and $Q = \mathcal{N}(\mu_j, \sigma_j^2)$ be two Gaussian distributions. Then the expected CRPS between these two distributions can be expressed as:

$$\mathrm{CRPS}\left(\mathcal{N}(\mu_i, \sigma_i), \mathcal{N}(\mu_j, \sigma_j)\right) = A(\mu_{ij}, \sigma_{ij}) - \frac{\sigma_i}{\sqrt{\pi}}. \tag{42}$$

**Entropy of CRPS for Gaussian mixture.**

$$H(\widehat{P}_{\text{ens}}) = \frac{1}{2}\mathbb{E}_{X,X'\sim\widehat{P}_{\text{ens}}}|X-X'| = \frac{1}{2}\frac{1}{M^2}\sum_{i=1}^{M}\sum_{j=1}^{M}\mathbb{E}_{X\sim\widehat{P}_i,X'\sim\widehat{P}_j}|X-X'| = \frac{1}{2}\frac{1}{M^2}\sum_{i=1}^{M}\sum_{j=1}^{M}A(\mu_{ij},\sigma_{ij}).$$

(43)

**Expected CRPS between a Gaussian and a Gaussian mixture.**

$$\text{CRPS}(\widehat{P}_i,\widehat{P}_{\text{ens}}) = \mathbb{E}_{X\sim\widehat{P}_i,Y\sim\widehat{P}_{\text{ens}}}|X-Y| - \frac{1}{2}\mathbb{E}_{X,X'\sim\widehat{P}_i}|X-X'| = \frac{1}{M}\sum_{j=1}^{M}A(\mu_{ij},\sigma_{ij}) - \frac{\sigma_i}{\sqrt{\pi}}.$$

(44)

**Divergence between a Gaussian and a Gaussian mixture.**

$$d(\widehat{P}_i,\widehat{P}_{\text{ens}}) = \text{CRPS}(\widehat{P}_i,\widehat{P}_{\text{ens}}) - H(\widehat{P}_{\text{ens}}) = \frac{1}{M}\sum_{j=1}^{M}A(\mu_{ij},\sigma_{ij}) - \frac{\sigma_i}{\sqrt{\pi}} - \frac{1}{2}\frac{1}{M^2}\sum_{l=1}^{M}\sum_{j=1}^{M}A(\mu_{lj},\sigma_{lj}).$$

(45)

### B.2 LOG SCORE

#### B.2.1 ENTROPY OF LOG SCORE

Log score is associated with Shannon entropy:

$$H(P) = \mathbb{E}_{Y\sim P}\left[-\log p(Y)\right] = -\int_{\mathbb{R}} p(y)\log p(y)\,dy.$$

(46)

#### B.2.2 DIVERGENCE OF LOG SCORE

Divergence function is the Kullback-Leibler divergence:

$$d(\widehat{P},P) = D_{\text{KL}}(P \,\|\, \widehat{P}) = \int_{\mathbb{R}} p(y)\log\frac{p(y)}{\widehat{p}(y)}\,dy.$$

(47)

#### B.2.3 LOG SCORE FOR GAUSSIANS

**Log score entropy for a Gaussian.** Logarithm of the Gaussian density is a quadratic function, so we can easily express the entropy using the expression for Gaussian:

$$Y\sim P = \mathcal{N}(\mu,\sigma) \implies p(y) = \frac{1}{\sqrt{2\pi}\sigma}e^{-\frac{(y-\mu)^2}{2\sigma^2}} \implies \text{LS}(P,y) = -\log p(y) = \frac{1}{2}\log\left(2\pi\sigma^2\right) + \frac{(y-\mu)^2}{2\sigma^2}.$$

(48)

$$H\left(\mathcal{N}(\mu,\sigma)\right) = \mathbb{E}_{Y\sim\mathcal{N}(\mu,\sigma)}\left[-\log p(Y)\right] = \mathbb{E}_{Y\sim\mathcal{N}(\mu,\sigma)}\left[\frac{1}{2}\log\left(2\pi\sigma^2\right) + \frac{(y-\mu)^2}{2\sigma^2}\right]$$

$$= \frac{1}{2}\log\left(2\pi\sigma^2\right) + \frac{1}{2\sigma^2}\mathbb{E}_{Y\sim\mathcal{N}(\mu,\sigma)}\left[(y-\mu)^2\right] = \frac{1}{2}\log\left(2\pi\sigma^2\right) + \frac{1}{2} = \frac{1}{2}\log\left(2\pi e\sigma^2\right).$$

**Log score divergence for two Gaussians.** Here, the derivation is very similar, using the variance/second moment formulas.

$$P = \mathcal{N}(\mu_p,\sigma_p^2),\ Q = \mathcal{N}(\mu_q,\sigma_q^2) \implies p(y) = \frac{1}{\sqrt{2\pi}\sigma_p}e^{-\frac{(y-\mu_p)^2}{2\sigma_p^2}},\ q(y) = \frac{1}{\sqrt{2\pi}\sigma_q}e^{-\frac{(y-\mu_q)^2}{2\sigma_q^2}}.$$

(49)

$$\log p(y) = -\frac{1}{2}\log(2\pi\sigma_p^2) - \frac{(y-\mu_p)^2}{2\sigma_p^2},\ \log q(y) = -\frac{1}{2}\log(2\pi\sigma_q^2) - \frac{(y-\mu_q)^2}{2\sigma_q^2}.$$

(50)

$$d(P, Q) = \mathbb{E}_{Y \sim Q} \left[ -\log p(Y) \right] - H(Q) = D_{\mathrm{KL}}(Q \,\|\, P)$$

$$= \mathbb{E}_{Y \sim Q} \left[ \log q(Y) - \log p(Y) \right] = \mathbb{E}_{Y \sim Q} \left[ \frac{1}{2} \log(2\pi\sigma_p^2) + \frac{(y - \mu_p)^2}{2\sigma_p^2} - \frac{1}{2} \log(2\pi\sigma_q^2) - \frac{(y - \mu_q)^2}{2\sigma_q^2} \right]$$

$$= \frac{1}{2} \left[ \log \frac{\sigma_p^2}{\sigma_q^2} - 1 + \frac{\sigma_q^2 + (\mu_p - \mu_q)^2}{\sigma_p^2} \right].$$

**Expected Log score between two Gaussians.**

$$\mathrm{LS}(P, Q) = d(P, Q) + H(Q) = \frac{1}{2} \left[ \log \frac{\sigma_p^2}{\sigma_q^2} - 1 + \frac{\sigma_q^2 + (\mu_p - \mu_q)^2}{\sigma_p^2} \right] + \frac{1}{2} \log \left( 2\pi e \sigma_q^2 \right)$$

$$= \frac{1}{2} \left[ \log \left( 2\pi \sigma_p^2 \right) + \frac{\sigma_q^2 + (\mu_p - \mu_q)^2}{\sigma_p^2} \right]. \tag{51}$$

**Pairwise divergences of a Gaussian ensemble.**

$$\widehat{R}_{\mathrm{Exc}}^{1,1} = \frac{1}{M^2} \sum_{i=1}^{M} \sum_{j=1}^{M} d(\widehat{P}_i, \widehat{P}_j) = \frac{1}{M^2} \sum_{i=1}^{M} \sum_{j=1}^{M} D_{\mathrm{KL}}(\widehat{P}_j, \widehat{P}_i)$$

$$= \frac{1}{M^2} \sum_{i=1}^{M} \sum_{j=1}^{M} \frac{1}{2} \left[ \log \frac{\sigma_i^2}{\sigma_j^2} - 1 + \frac{\sigma_j^2 + (\mu_i - \mu_j)^2}{\sigma_i^2} \right]$$

$$= \frac{1}{2M^2} \sum_{i=1}^{M} \sum_{j=1}^{M} \left[ \frac{\sigma_j^2 + (\mu_i - \mu_j)^2}{\sigma_i^2} - 1 \right], \tag{52}$$

because sum of all pairwise differences of logarithms is 0.

**Divergence between a Gaussian mixture and one of its components.** For the log score the divergence function is KL-divergence, it is not symmetric. This provides two possible ways to approximate the excess risk: assume the true distribution to be the full mixture and average over the components or the other way around. In the notation of (Kotelevskii et al., 2025) they correspond to Bregman Information (BI) and Reverse Bregman Information (RBI).

$$d(\widehat{P}_{\mathrm{ens}}, \widehat{P}_i) = D_{\mathrm{KL}}(\widehat{P}_i \,\|\, \widehat{P}_{\mathrm{ens}})$$

$$= \mathrm{LS}(\widehat{P}_{\mathrm{ens}}, \widehat{P}_i) - H(\widehat{P}_i) = -\int_{\mathbb{R}} \widehat{p}_i(y) \log \left( \widehat{p}_{\mathrm{ens}}(y) \right) dy - \frac{1}{2} \log \left( 2\pi e \sigma_i^2 \right).$$

$$d(\widehat{P}_i, \widehat{P}_{\mathrm{ens}}) = D_{\mathrm{KL}}(\widehat{P}_{\mathrm{ens}} \,\|\, \widehat{P}_i) = \mathrm{LS}(\widehat{P}_i, \widehat{P}_{\mathrm{ens}}) - H(\widehat{P}_{\mathrm{ens}})$$

$$= -\int_{\mathbb{R}} \widehat{p}_{\mathrm{ens}}(y) \log \widehat{p}_i(y) \, dy - H(\widehat{P}_{\mathrm{ens}}) = \frac{1}{M} \sum_{j=1}^{M} \mathrm{LS}(\widehat{P}_i, \widehat{P}_j) - H(\widehat{P}_{\mathrm{ens}})$$

$$= \frac{1}{M} \sum_{j=1}^{M} \mathrm{LS}(\widehat{P}_i, \widehat{P}_j) - H(\widehat{P}_{\mathrm{ens}}).$$

There is no more explicit analytical formula due to appearance of logarithm of the mixture density, which is a sum of the component densities. In the discrete case there are expressions involving the *LogSumExp* function.

**Divergence between a Gaussian ensemble and its Gaussian approximation.**

$$\widehat{P}_* = \mathcal{N}(\mu_*, \sigma_*), \mu_* = \frac{1}{M} \sum_{i=1}^{M} \mu_i, \sigma_*^2 = \frac{1}{M} \sum_{i=1}^{M} (\sigma_i^2 + \mu_i^2) - \mu_*^2.$$

$$\widehat{R}_{\text{Exc}}^{3a,1} = \frac{1}{M} \sum_{i=1}^{M} d(\widehat{P}_*, \widehat{P}_i) = \frac{1}{M} \sum_{i=1}^{M} \frac{1}{2} \left[ \log \frac{\sigma_*^2}{\sigma_i^2} - 1 + \frac{\sigma_i^2 + (\mu_* - \mu_i)^2}{\sigma_*^2} \right]$$

$$= \frac{1}{2} \left[ \log\left(\sigma_*^2\right) - \frac{1}{M} \sum_{i=1}^{M} \log\left(\sigma_i^2\right) - 1 + \frac{1}{\sigma_*^2} \frac{1}{M} \sum_{i=1}^{M} \sigma_i^2 + (\mu_* - \mu_i)^2 \xrightarrow{\quad \sigma_*^2 \quad} \right]$$

$$= \frac{1}{2} \left[ \log\left(\sigma_*^2\right) - \frac{1}{M} \sum_{i=1}^{M} \log\left(\sigma_i^2\right) \right].$$

**Divergence between a Gaussian ensemble and mean Gaussian approximation.** $\bar{P}_* = \mathcal{N}(\mu_*, \bar{\sigma}_*^2)$, where $\mu_* = \frac{1}{M} \sum_{i=1}^{M} \mu_i$ and $\bar{\sigma}_*^2 = \frac{1}{M} \sum_{i=1}^{M} \sigma_i^2$.

$$\widehat{R}_{\text{Exc}}^{3b,1} = \frac{1}{M} \sum_{i=1}^{M} d(\widehat{P}, \widehat{P}_i) = \frac{1}{M} \sum_{i=1}^{M} \frac{1}{2} \left[ \log \frac{\bar{\sigma}_*^2}{\sigma_i^2} - 1 + \frac{\sigma_i^2 + (\mu_* - \mu_i)^2}{\bar{\sigma}_*^2} \right]$$

$$= \frac{1}{2} \left[ \log\left(\bar{\sigma}_*^2\right) - \frac{1}{M} \sum_{i=1}^{M} \log\left(\sigma_i^2\right) - 1 + \frac{1}{\bar{\sigma}_*^2 M} \sum_{i=1}^{M} \sigma_i^2 \xrightarrow{\quad 1 \quad} + \frac{1}{\bar{\sigma}_*^2 M} \sum_{i=1}^{M} (\mu_* - \mu_i)^2 \right]$$

$$= \frac{1}{2} \left[ \log\left(\bar{\sigma}_*^2\right) - \frac{1}{M} \sum_{i=1}^{M} \log\left(\sigma_i^2\right) + \frac{1}{\bar{\sigma}_*^2 M} \sum_{i=1}^{M} (\mu_* - \mu_i)^2 \right].$$

### B.3 QUADRATIC SCORE

Quadratic score is a generalization of the Brier score for continuous outcomes:

$$\text{QS}(\widehat{P}, y) = -2\widehat{p}(y) + \int_{\mathbb{R}} \widehat{p}(t)^2 dt. \tag{53}$$

It can also be viewed as a way to make the value of the probability density at the outcome $y$ into a proper scoring rule. Simply taking $S(\widehat{P}, y) = -\widehat{p}(y)$ is not a proper scoring rule.

#### B.3.1 QUADRATIC SCORE ENTROPY

$$H(P) = \mathbb{E}_{Y \sim P}\left[\text{QS}(P, Y)\right] = -2\mathbb{E}_{Y \sim P}\left[p(Y)\right] + \int_{\mathbb{R}} p(t)^2 dt$$

$$= -2 \int_{\mathbb{R}} p(y)p(y)dy + \int_{\mathbb{R}} p(t)^2 dt = -\int_{\mathbb{R}} p(y)^2 dy. \tag{54}$$

#### B.3.2 QUADRATIC SCORE ENTROPY FOR A GAUSSIAN

$$H\left(\mathcal{N}(\mu, \sigma)\right) = -\int_{\mathbb{R}} p(y)^2 dy = -\int_{\mathbb{R}} \left( \frac{1}{\sqrt{2\pi}\sigma} e^{-\frac{(t-\mu)^2}{2\sigma^2}} \right)^2 dt = -\frac{1}{2\pi\sigma^2} \int_{\mathbb{R}} e^{-\frac{(t-\mu)^2}{\sigma^2}} dt$$

$$= \begin{bmatrix} u = \frac{t-\mu}{\sigma} \\ du = \frac{f}{\sigma} dt \end{bmatrix} = -\frac{1}{2\pi\sigma^2} \sigma \int_{\mathbb{R}} e^{-u^2} du = \frac{\sqrt{\pi}}{2\pi\sigma} = -\frac{1}{2\sqrt{\pi}\sigma}.$$

#### B.3.3 EXPECTED QUADRATIC SCORE FOR A GAUSSIAN PREDICTION AND GAUSSIAN OUTCOME

$$\mathbb{E}_{Y \sim Q}\left[\text{QS}(P, Y)\right] = -2\mathbb{E}_{Y \sim Q}\left[p(Y)\right] + \int_{\mathbb{R}} p(t)^2 dt = -2 \int_{\mathbb{R}} p(y)q(y)dy - H(P). \tag{55}$$

We will focus first on computing the integral. Since $P$ and $Q$ are Gaussian distributions, we can express the integral as follows:

$$\int_{\mathbb{R}} p(t)q(t)dt = \frac{1}{2\pi\sigma_P\sigma_Q}\int_{\mathbb{R}} e^{-\frac{(t-\mu_P)^2}{2\sigma_P^2}-\frac{(t-\mu_Q)^2}{2\sigma_Q^2}}dt = \frac{1}{2\pi\sigma_P\sigma_Q}\int_{\mathbb{R}} e^{-\frac{1}{2}\left(\frac{(t-\mu_P)^2}{\sigma_P^2}+\frac{(t-\mu_Q)^2}{\sigma_Q^2}\right)}dt. \tag{56}$$

Let us simplify the exponent:

$$\frac{(t-\mu_P)^2}{\sigma_P^2}+\frac{(t-\mu_Q)^2}{\sigma_Q^2} = \frac{t^2}{\sigma_P^2}-2\frac{t\mu_P}{\sigma_P^2}+\frac{\mu_P}{\sigma_P^2}+\frac{t^2}{\sigma_Q^2}-2\frac{t\mu_Q}{\sigma_Q^2}+\frac{\mu_Q}{\sigma_Q^2}$$

$$= \left(\frac{1}{\sigma_P^2}+\frac{1}{\sigma_Q^2}\right)t^2-2\left(\frac{\mu_P}{\sigma_P^2}+\frac{\mu_Q}{\sigma_Q^2}\right)t+\left(\frac{\mu_P^2}{\sigma_P^2}+\frac{\mu_Q^2}{\sigma_Q^2}\right) = At^2-2Bt+C = A\left(t-\frac{B}{A}\right)^2-\frac{B^2}{A}+C, \tag{57}$$

where we denote:

$$A = \frac{1}{\sigma_P^2}+\frac{1}{\sigma_Q^2}, \ B = \frac{\mu_P}{\sigma_P^2}+\frac{\mu_Q}{\sigma_Q^2}, \ C = \frac{\mu_P^2}{\sigma_P^2}+\frac{\mu_Q^2}{\sigma_Q^2}. \tag{58}$$

Now we have:

$$\int_{\mathbb{R}} e^{-\frac{1}{2}\left(At^2-2Bt+C\right)}dt = e^{-\frac{1}{2}\left(C-\frac{B^2}{A}\right)}\int_{\mathbb{R}} e^{-\frac{A}{2}\left(t-\frac{B}{A}\right)^2}dt. \tag{59}$$

The leftover integral is a Gaussian integral, which can be computed as follows:

$$\int_{\mathbb{R}} e^{-\frac{A}{2}\left(t-\frac{B}{A}\right)^2}dt = \sqrt{\frac{2\pi}{A}} = \sqrt{2\pi}\frac{\sigma_P\sigma_Q}{\sqrt{\sigma_P^2+\sigma_Q^2}}. \tag{60}$$

Now, we simplify the exponent of the coefficient in front of that integral:

$$A = \frac{1}{\sigma_P^2}+\frac{1}{\sigma_Q^2} = \frac{\sigma_Q^2+\sigma_P^2}{\sigma_P^2\sigma_Q^2}, \ B^2 = \left(\frac{\mu_P}{\sigma_P^2}+\frac{\mu_Q}{\sigma_Q^2}\right)^2 = \frac{\left(\mu_P\sigma_Q^2+\mu_Q\sigma_P^2\right)^2}{\sigma_P^4\sigma_Q^4}, \ C = \frac{\mu_P^2\sigma_Q^2+\mu_Q^2\sigma_P^2}{\sigma_P^2\sigma_Q^2}. \tag{61}$$

$$C-\frac{B^2}{A} = \frac{\mu_P^2\sigma_Q^2+\mu_Q^2\sigma_P^2}{\sigma_P^2\sigma_Q^2}-\frac{\left(\mu_P\sigma_Q^2+\mu_Q\sigma_P^2\right)^2}{\sigma_P^2\sigma_Q^2\left(\sigma_Q^2+\sigma_P^2\right)} = \frac{\left(\mu_P^2\sigma_Q^2+\mu_Q^2\sigma_P^2\right)\left(\sigma_Q^2+\sigma_P^2\right)-\left(\mu_P\sigma_Q^2+\mu_Q\sigma_P^2\right)^2}{\sigma_P^2\sigma_Q^2\left(\sigma_Q^2+\sigma_P^2\right)}. \tag{62}$$

$$\left(\mu_P^2\sigma_Q^2+\mu_Q^2\sigma_P^2\right)\left(\sigma_Q^2+\sigma_P^2\right)-\left(\mu_P\sigma_Q^2+\mu_Q\sigma_P^2\right)^2$$

$$= \mu_P^2\sigma_Q^4+\mu_Q^2\sigma_P^4+\mu_P^2\sigma_P^2\sigma_Q^2+\mu_Q^2\sigma_P^2\sigma_Q^2-\left(\mu_P^2\sigma_Q^4+2\mu_P\mu_Q\sigma_P^2\sigma_Q^2+\mu_Q^2\sigma_P^4\right)$$

$$= \sigma_P^2\sigma_Q^2\left(\mu_P-\mu_Q\right)^2. \tag{63}$$

Thus, we have:

$$C-\frac{B^2}{A} = \frac{\sigma_P^2\sigma_Q^2\left(\mu_P-\mu_Q\right)^2}{\sigma_P^2\sigma_Q^2\left(\sigma_Q^2+\sigma_P^2\right)} = \frac{\left(\mu_P-\mu_Q\right)^2}{\sigma_Q^2+\sigma_P^2}. \tag{64}$$

$$\int_{\mathbb{R}} p(t)\,q(t)\,dt = \frac{1}{2\pi\sigma_P\sigma_Q}\int_{\mathbb{R}} e^{-\frac{(t-\mu_P)^2}{2\sigma_P^2}-\frac{(t-\mu_Q)^2}{2\sigma_Q^2}}dt = \frac{1}{2\pi\sigma_P\sigma_Q}\sqrt{2\pi}\frac{\sigma_P\sigma_Q}{\sqrt{\sigma_P^2+\sigma_Q^2}}e^{-\frac{1}{2}\frac{\left(\mu_P-\mu_Q\right)^2}{\sigma_P^2+\sigma_Q^2}}$$

$$= \frac{1}{\sqrt{2\pi(\sigma_P^2+\sigma_Q^2)}}e^{-\frac{\left(\mu_P-\mu_Q\right)^2}{2\left(\sigma_P^2+\sigma_Q^2\right)}}.$$

We can notice that this is the density of a Gaussian distribution with mean $\mu_P$ and variance $\sigma_P^2 + \sigma_Q^2$ evaluated at $\mu_Q$. We will also use a well-known notation for this value: $\mathcal{N}(\mu_P \mid \mu_Q, \sigma_P^2 + \sigma_Q^2)$. Here

$$\mathcal{N}(a \mid b, s^2) = \frac{1}{\sqrt{2\pi s^2}} e^{-\frac{(a-b)^2}{2s^2}}, \tag{65}$$

and

$$\mathrm{QS}(P, Q) = -2\mathcal{N}\left(\mu_P \mid \mu_Q, \sigma_P^2 + \sigma_Q^2\right) + \frac{1}{2\sqrt{\pi}\sigma_P}. \tag{66}$$

Additionally, we present another useful identity involving the expected quadratic score for a Gaussian prediction and Gaussian outcome:

$$\int_{\mathbb{R}} p(t)\, q(t)\, dt = -\frac{\mathrm{QS}(P, Q) + H(P)}{2}. \tag{67}$$

### B.3.4 QUADRATIC SCORE DIVERGENCE FOR TWO GAUSSIANS

$$d(P, Q) = \mathbb{E}_{Y \sim Q}\left[\mathrm{QS}(P, Y)\right] - H(Q) = -2\mathbb{E}_{Y \sim Q}\left[p(Y)\right] - H(P) - H(Q). \tag{68}$$

We can already compute these terms using the results from the previous sections. We have the following:

$$-2\mathbb{E}_{Y \sim Q}\left[p(Y)\right] = -2\mathcal{N}\left(\mu_P \mid \mu_Q, \sigma_P^2 + \sigma_Q^2\right) = -\frac{2}{\sqrt{2\pi(\sigma_P^2 + \sigma_Q^2)}} e^{-\frac{\left(\mu_P - \mu_Q\right)^2}{2\left(\sigma_P^2 + \sigma_Q^2\right)}}. \tag{69}$$

$$H(P) = -\frac{1}{2\sqrt{\pi}\sigma_P}, \quad H(Q) = -\frac{1}{2\sqrt{\pi}\sigma_Q}. \tag{70}$$

Thus, we can express the quadratic score divergence for two Gaussians as follows:

$$d(P, Q) = -2\mathcal{N}\left(\mu_P \mid \mu_Q, \sigma_P^2 + \sigma_Q^2\right) + \frac{1}{2\sqrt{\pi}\sigma_P} + \frac{1}{2\sqrt{\pi}\sigma_Q}. \tag{71}$$

Another convenient expression for the quadratic score divergence is:

$$d(P, Q) = \mathbb{E}_{Y \sim Q}\left[\mathrm{QS}(P, Y)\right] - H(Q) = -2\int_{\mathbb{R}} p(t)\, q(t)\, dt + \int_{\mathbb{R}} p(t)^2\, dt + \int_{\mathbb{R}} q(t)^2\, dt = \int_{\mathbb{R}}\left(p(t) - q(t)\right)^2 dt.$$

This divergence is symmetric: $d(P, Q) = d(Q, P)$. It is similar to CRPS score divergence, but this time we use densities instead of cumulative distribution functions.

### B.3.5 ENTROPY OF QUADRATIC SCORE FOR A GAUSSIAN MIXTURE

$$H(\widehat{P}_{\mathrm{ens}}) = -\int_{\mathbb{R}}\left(\frac{1}{M}\sum_{i=1}^{M}\widehat{p}_i(y)\right)^2 dy = -\frac{1}{M^2}\sum_{i=1}^{M}\sum_{j=1}^{M}\mathcal{N}\left(\mu_i \mid \mu_j, \sigma_i^2 + \sigma_j^2\right). \tag{72}$$

Alternatively, we can express the entropy of quadratic score for a Gaussian mixture as follows:

$$H(\widehat{P}_{\mathrm{ens}}) = \frac{1}{M^2}\left[\sum_{i=1}^{M} H(\widehat{P}_i) - 2\sum_{j<i}\mathcal{N}\left(\mu_i \mid \mu_j, \sigma_i^2 + \sigma_j^2\right)\right] \tag{73}$$

$$= -\frac{1}{M^2}\left[\frac{1}{2\sqrt{\pi}}\sum_{i=1}^{M}\frac{1}{\sigma_i} + 2\sum_{j<i}\mathcal{N}\left(\mu_i \mid \mu_j, \sigma_i^2 + \sigma_j^2\right)\right]. \tag{74}$$

Next we express the entropy of quadratic score for a Gaussian mixture in terms of the expected quadratic score and entropy of its components:

$$H(\widehat{P}_{\mathrm{ens}}) = -\frac{1}{M^2}\sum_{i=1}^{M}\sum_{j=1}^{M}\int_{\mathbb{R}}\widehat{p}_i(y)\,\widehat{p}_j(y)\, dy = \frac{1}{M^2}\sum_{i=1}^{M}\sum_{j=1}^{M}\frac{\mathrm{QS}(\widehat{P}_i, \widehat{P}_j) + H(\widehat{P}_i)}{2}$$

$$= \frac{1}{2M}\sum_{i=1}^{M} H(\widehat{P}_i) + \frac{1}{2M^2}\sum_{i=1}^{M}\sum_{j=1}^{M}\mathrm{QS}(\widehat{P}_i, \widehat{P}_j) = \frac{M+1}{2M^2}\sum_{i=1}^{M} H(\widehat{P}_i) + \frac{1}{2M^2}\sum_{i\neq j}\mathrm{QS}(\widehat{P}_i, \widehat{P}_j).$$

### B.3.6 Quadratic Score Divergence between a Gaussian and a Gaussian Mixture

$$d(\widehat{P}_i, \widehat{P}_{\mathrm{ens}}) = \mathbb{E}_{Y \sim \widehat{P}_{\mathrm{ens}}} \left[ \mathrm{QS}(\widehat{P}_i, Y) \right] - H(\widehat{P}_{\mathrm{ens}}) = \frac{1}{M} \sum_{j=1}^{M} \mathbb{E}_{Y \sim \widehat{P}_j} \left[ \mathrm{QS}(\widehat{P}_i, Y) \right] - H(\widehat{P}_{\mathrm{ens}})$$

$$= \frac{1}{M} \sum_{j=1}^{M} \mathrm{QS}(\widehat{P}_i, \widehat{P}_j) - H(\widehat{P}_{\mathrm{ens}}) = -\frac{2}{M} \sum_{j=1}^{M} \mathcal{N}\left( \mu_i \mid \mu_j, \sigma_i^2 + \sigma_j^2 \right) - H(\widehat{P}_i) - H(\widehat{P}_{\mathrm{ens}}).$$

$$(75)$$

## C    Proper Scoring Rules from Consistent Scoring Functions

A closely related concept to proper scoring rule is the *consistent scoring function*. In some situations a probabilistic forecast is not possible or not needed (e.g., a point prediction is required by some regulation); for this setup a similar theory was developed (Savage, 1971; Gneiting, 2009). There is one important difference when evaluating point forecasts: since we still follow the random data model (the true outcome $y$ is stochastic, comes from distribution $P$), we need to select a functional $T(P)$ of the true data distribution that we want to estimate. You can refer to (Gneiting, 2009) for a detailed discussion, and in the next subsection we provide a brief overview of the main definitions and results.

### C.1    Introduction

**Definition 3** (Definition 2.1 from (Gneiting, 2009))**.** Scoring function $S$ is consistent for the functional $T$ relative to the class of distributions $\mathcal{P}$ if

$$\mathbb{E}_{Y \sim P} \left[ S(t, Y) \right] \leq \mathbb{E}_{Y \sim P} \left[ S(x, Y) \right], \tag{76}$$

for all distributions $P \in \mathcal{P}$, all $t \in T(P)$ and all $x \in \mathrm{dom}(Y)$.

One of the most famous results in this area is the following.

**Theorem 4** (Savage (Savage, 1971), 1971)**.** *Every scoring function that is consistent for the mean functional $T(P) = \mathbb{E}_P[Y]$ admits a representation as a Bregman divergence:*

$$S(x, y) = \varphi(y) - \varphi(x) - \varphi'(x)(y - x), \tag{77}$$

*where $\varphi$ is a convex function, and $\varphi'$ is its subgradient. Such $S(x, y)$ are also called Bregman functions.*

For example, the quadratic score $S(x, y) = (y - x)^2$ is a Bregman function with $\varphi(t) = t^2$.

**Theorem 5** (Theorem 2.2 from (Gneiting, 2009))**.** *Score $S$ is consistent for $T$ if and only if any $t \in T(P)$ is an optimal point prediction: $t \in y_{Bayes}$.*

Optimal point prediction in this setting is the Bayes act $\widehat{y}_{\mathrm{Bayes}}$ and can be expressed as follows:

$$\widehat{y}_{\mathrm{Bayes}} = \arg \min_x \mathbb{E}_{Y \sim P} \left[ S(x, Y) \right]. \tag{78}$$

In the case of the mean functional, $y_{\mathrm{Bayes}} = T(P) = \mathbb{E}_{Y \sim P}[Y]$ which also follows from the representation of the consistent score as a Bregman function. Now we are ready to introduce the main tool. If we have a scoring function that is consistent for some functional, then we can use it to construct a (rather simple) proper scoring rule:

**Theorem 6** (Theorem 2.3 from (Gneiting, 2009))**.** *If $S_T$ is consistent for the functional $T(\cdot)$, then the following scoring rule is proper:*

$$S(\widehat{P}, y) = S_T \big( T(\widehat{P}), y \big).$$

## C.2 SE SCORE FROM QUADRATIC SCORING FUNCTION

Based on the quadratic scoring function, which is consistent for the mean, we introduce the following proper scoring rule for continuous distributions:

$$\text{SE}(\widehat{P}, y) = \big(y - \mathbb{E}_{Y \sim \widehat{P}}[Y]\big)^2,$$

and call it the SE score. This score is proper, but not strictly proper, since any distribution with the same mean will have the same score.

## C.3 RISK COMPONENTS FOR SE SCORE

The risk components for this score are as follows:

- **Total risk**: $R_{\text{Tot}}(\widehat{P}, P) = \mathbb{E}_{Y \sim P}\big[\text{SE}(\widehat{P}, Y)\big] = \mathbb{E}_{Y \sim P}\big[(Y - \mathbb{E}_{X \sim \widehat{P}}[X])^2\big]$, where

$$
\begin{aligned}
\mathbb{E}_{Y \sim P}\big[(Y - \mathbb{E}_{X \sim \widehat{P}}[X])^2\big] &= \mathbb{E}_{Y \sim P}\big[Y^2\big] - 2\mathbb{E}_{Y \sim P}[Y] \cdot \mathbb{E}_{X \sim \widehat{P}}[X] + (\mathbb{E}_{X \sim \widehat{P}}[X])^2 \\
&= \big(\mathbb{E}_{X \sim \widehat{P}}[X] - \mathbb{E}_{Y \sim P}[Y]\big)^2 + \mathbb{E}_{Y \sim P}\big[Y^2\big] - (\mathbb{E}_{Y \sim P}[Y])^2 \\
&= \big(\mathbb{E}_{X \sim \widehat{P}}[X] - \mathbb{E}_{Y \sim P}[Y]\big)^2 + \text{Var}_{Y \sim P}[Y].
\end{aligned}
$$

$$R_{\text{Tot}}(\widehat{P}, P) = \big(\mathbb{E}_{X \sim \widehat{P}}[X] - \mathbb{E}_{Y \sim P}[Y]\big)^2 + \text{Var}_{Y \sim P}[Y].$$

- **Bayes risk**: $R_{\text{Bayes}}(P) = H_{\text{SE}}(P) = \mathbb{E}_{Y \sim P}\big[\text{SE}(P, Y)\big] = \text{Var}_{Y \sim P}[Y]$.
- **Excess risk**: $R_{\text{Exc}}(\widehat{P}, P) = d_{\text{SE}}(\widehat{P}, P) = R_{\text{Tot}}(\widehat{P}, P) - R_{\text{Bayes}}(P) = \big(\mathbb{E}_{X \sim \widehat{P}}[X] - \mathbb{E}_{Y \sim P}[Y]\big)^2$.

In other words, the SE score corresponds to the following entropy and divergence functions:

$$H_{\text{SE}}(P) = \text{Var}_{Y \sim P}[Y], \ d_{\text{SE}}(\widehat{P}, P) = \big(\mathbb{E}_{X \sim \widehat{P}}[X] - \mathbb{E}_{Y \sim P}[Y]\big)^2.$$

# D CHARACTERIZATION OF PROPER SCORING RULES AND GENERAL FORM OF RISK ESTIMATES

The following theorem provides a general form of a proper scoring rule using the notion of a convex function on the space of probability measures:

**Definition 7.** A scoring rule $S$ is *regular* if $H(P) = S(P, P)$ is finite and $S(P, Q) > -\infty, \forall P, Q \in \mathcal{P}$.

**Theorem 8** (Theorem 1 from (Gneiting & Raftery, 2007), notation from Theorem 12 in (Waghmare & Ziegel, 2025)). *A regular scoring rule is proper if and only if there exists a concave function $H \colon \mathcal{P} \to \mathbb{R}$ such that*

$$S(P, y) = H(P) + \langle h_P, \delta_y - P \rangle = H(P) + h_P(y) - \int h_P(t) \, dP(t) \tag{79}$$

*for every $P \in \mathcal{P}$ and $y \in \mathcal{Y}$, where $h_P$ is a supergradient of $H$ at $P$ and $\langle h_P, Q \rangle = \int h_P(t) \, dQ(t)$.*

From this theorem we can obtain the following general expression of the divergence for a proper scoring rule:

$$d(P, Q) = H(P) - H(Q) - \langle h_P, P - Q \rangle. \tag{80}$$

The concave function of Theorem 8 coincides with the corresponding score entropy $H$. In case of our ensemble setting, this leads to the following relation between the risk estimates:

$$\widehat{R}_{\text{Bayes}}^1 = \frac{1}{M} \sum_{i=1}^{M} H(\widehat{P}_i) \leq H\left(\frac{1}{M} \sum_{i=1}^{M} \widehat{P}_i\right) \tag{81}$$

$$= H(\widehat{P}_{\text{ens}}) = \widehat{R}_{\text{Bayes}}^2.$$

Table 8: Representations of some proper scoring rules

| Score | Entropy $H(P)$ | Supergradient map $h_P(y)$ |
|---|---|---|
| CRPS | $\frac{1}{2}\mathbb{E}_{X,X'\sim P}\|X - X'\|$ | $\mathbb{E}_{X\sim P}\|X - y\|$ |
| LS | $-\int_{\mathbb{R}} p(t)\log p(t)dt$ | $-\log p(y)$ |
| QS | $-\int_{\mathbb{R}} p(t)^2 dt$ | $-2p(y)$ |
| SE | $\text{Var}_{Y\sim P}[Y]$ | $y^2 - 2y\mathbb{E}_{X\sim P}[X]$ |

For the Logarithmic score, we know that:

$$\text{LS:} \ \widehat{R}^3_{\text{Bayes}} = H(\widehat{P}_*) \geq H(\widehat{P}_{\text{ens}}) = \widehat{R}^2_{\text{Bayes}}, \tag{82}$$

since among the distributions with the same variance (in this case, both have variance $\sigma_*^2$), the Gaussian distribution has the highest (Shannon) entropy. Combining these results, we get:

$$\text{LS:} \ \widehat{R}^1_{\text{Bayes}} \leq \widehat{R}^2_{\text{Bayes}} \leq \widehat{R}^3_{\text{Bayes}}. \tag{83}$$

Whether this result holds for other scores or in general remains open. We think that it depends on the relationship between the particular score's entropy and the Gaussian distribution that we used for our ensemble estimate.

**CRPS.**

$$\widehat{R}^3_{\text{Bayes}} = \sqrt{\frac{\sigma_*^2}{\pi}} = \frac{1}{\sqrt{\pi}}\sqrt{\frac{1}{M}\sum_{i=1}^{M}\sigma_i^2 + \text{Var}[\mu_i])} \geq \frac{1}{\sqrt{\pi}}\sqrt{\frac{1}{M}\sum_{i=1}^{M}\sigma_i^2} \geq \frac{1}{\sqrt{\pi}}\frac{1}{M}\sum_{i=1}^{M}\sigma_i = \widehat{R}^1_{\text{Bayes}}.$$

**QS.**

$$\widehat{R}^3_{\text{Bayes}} = -\frac{1}{2\sqrt{\pi}\sigma_*} = -\frac{1}{2\sqrt{\pi}}\frac{1}{\sqrt{\frac{1}{M}\sum_{i=1}^{M}\sigma_i^2 + \text{Var}[\mu_i]}} \geq$$

$$= -\frac{1}{2\sqrt{\pi}}\frac{1}{\sqrt{\frac{1}{M}\sum_{i=1}^{M}\sigma_i^2}} \geq -\frac{1}{2\sqrt{\pi}}\frac{1}{M}\sum_{i=1}^{M}\frac{1}{\sigma_i} = \widehat{R}^1_{\text{Bayes}}.$$

Here we have applied Jensen's inequality to the function $f(x) = -\frac{1}{\sqrt{x}}$, $x_i = \sigma_i^2$.

**SE.**

$$\widehat{R}^3_{\text{Bayes}} = \sigma_*^2 = \frac{1}{M}\sum_{i=1}^{M}\sigma_i^2 + \text{Var}[\mu_i] \geq \frac{1}{M}\sum_{i=1}^{M}\sigma_i^2 = \widehat{R}^1_{\text{Bayes}}.$$

### D.1 ADDITIONAL RESULTS FOR EXCESS RISK

$$\widehat{R}^{2,1}_{\text{Exc}} = \frac{1}{M}\sum_{i=1}^{M} d(\widehat{P}_{\text{ens}}, \widehat{P}_i) = \frac{1}{M}\sum_{i=1}^{M}\left[H(\widehat{P}_{\text{ens}}) - H(\widehat{P}_i) - \langle h_{\widehat{P}_{\text{ens}}}, \widehat{P}_{\text{ens}} - \widehat{P}_i\rangle\right] \tag{84}$$

$$= H(\widehat{P}_{\text{ens}}) - \frac{1}{M}\sum_{i=1}^{M} H(\widehat{P}_i) - \langle h_{\widehat{P}_{\text{ens}}}, \widehat{P}_{\text{ens}} - \frac{1}{M}\sum_{i=1}^{M}\widehat{P}_i \overset{0}{\nearrow}\rangle = \widehat{R}^2_{\text{Bayes}} - \widehat{R}^1_{\text{Bayes}} \geq 0. \tag{85}$$

$$\widehat{R}_{\text{Exc}}^{1,1} = \frac{1}{M^2} \sum_{i=1}^{M} \sum_{j=1}^{M} d(\widehat{P}_i, \widehat{P}_j) = \frac{1}{M^2} \sum_{i=1}^{M} \sum_{j=1}^{M} \left[ H(\widehat{P}_i) - H(\widehat{P}_j) - \langle h_{\widehat{P}_i}, \widehat{P}_i - \widehat{P}_j \rangle \right]$$

$$= -\frac{1}{M^2} \sum_{i=1}^{M} \sum_{j=1}^{M} \langle h_{\widehat{P}_i}, \widehat{P}_i - \widehat{P}_j \rangle = -\frac{1}{M} \sum_{i=1}^{M} \langle h_{\widehat{P}_i}, \widehat{P}_i - \frac{1}{M} \sum_{j=1}^{M} \widehat{P}_j \rangle = \frac{1}{M} \sum_{i=1}^{M} \langle h_{\widehat{P}_i}, \widehat{P}_{\text{ens}} - \widehat{P}_i \rangle$$

$$\geq \frac{1}{M} \sum_{i=1}^{M} \left[ H(\widehat{P}_{\text{ens}}) - H(\widehat{P}_i) \right] = H(\widehat{P}_{\text{ens}}) - \frac{1}{M} \sum_{i=1}^{M} H(\widehat{P}_i) = \widehat{R}_{\text{Bayes}}^{2} - \widehat{R}_{\text{Bayes}}^{1} = \widehat{R}_{\text{Exc}}^{2,1}.$$

$$(86)$$

$$\widehat{R}_{\text{Exc}}^{1,2} = \frac{1}{M} \sum_{i=1}^{M} d(\widehat{P}_i, \widehat{P}_{\text{ens}}) = \frac{1}{M} \sum_{i=1}^{M} \left[ H(\widehat{P}_i) - H(\widehat{P}_{\text{ens}}) - \langle h_{\widehat{P}_i}, \widehat{P}_i - \widehat{P}_{\text{ens}} \rangle \right]$$

$$= \frac{1}{M} \sum_{i=1}^{M} H(\widehat{P}_i) - H(\widehat{P}_{\text{ens}}) + \frac{1}{M} \sum_{i=1}^{M} \langle h_{\widehat{P}_i}, \widehat{P}_{\text{ens}} - \widehat{P}_i \rangle = \widehat{R}_{\text{Exc}}^{1,1} - \widehat{R}_{\text{Exc}}^{2,1}$$

$$\implies \widehat{R}_{\text{Exc}}^{1,1} = \widehat{R}_{\text{Exc}}^{2,1} + \widehat{R}_{\text{Exc}}^{1,2}.$$

$$(87)$$

# E   EXPERIMENTS

The code for reproducing all the experiments in this paper is provided at TBD

## E.1   TRAINING OBJECTIVE

In our experiments, we consider neural networks predicting the mean and variance of an output distribution $\mathcal{N}(\mu(x; w), \sigma^2(x; w))$ over $y$ for a given input $x$. The standard way of optimizing a model with parameters $w$ to predict a faithful output distribution is by minimizing the log-likelihood of the output distribution on the training set $D$ (Nix & Weigend, 1994; Lakshminarayanan et al., 2017). Formulated as negative log-likelihood, the element-wise loss is given by

$$\mathcal{L}(w) = \frac{1}{2} \log \sigma^2(x; w) + \frac{(y - \mu(x; w))^2}{2\sigma^2(x; w)} + const. \tag{88}$$

However, training both mean and variance networks in conjunction is known to suffer from instability. Therefore, there have been multiple attempts to stabilize training and obtain both accurate mean predictions and calibrated variance (Skafte et al., 2019; Seitzer et al., 2022; Immer et al., 2023). We follow the approach of Immer et al. (2023) in using the natural parameterization of the Gaussian to reformulate the optimization problem and obtain more stable optimization properties. Instead of predicting mean $\mu$ and variance $\sigma^2$, in the natural parameterization $\eta_1 = \frac{\mu}{\sigma^2}$ and $\eta_2 = -\frac{1}{2\sigma^2}$ are predicted, where one needs to make sure that $\eta_2 < 0$. Then, the negative log-likelihood is given by

$$\mathcal{L}(w) = -\begin{pmatrix} \eta_1(x; w) \\ \eta_2(x; w) \end{pmatrix}^T \begin{pmatrix} y \\ y^2 \end{pmatrix} - \frac{\eta_1(x; w)^2}{4\eta_2(x; w)} - \frac{1}{2} \log(-2\eta_2(x; w)) + const. \tag{89}$$

We compare the two losses on a simple regression dataset using a three layer neural networks consisting of feed-forward layers with hidden dimension 32 and ReLU activation. Data points are drawn according to a sine function with larger noise in the right open half-plane. While we observe in Figure 4 that using Equation (89) needs more gradient steps than optimization with Equation (88), training and validation losses are much more stable. Furthermore, the resulting behavior is arguable more favorable for the natural parameterization (see Figure 5). For the standard parameterization, the predicted variance collapses away from data, while it stays at least relatively constant for the natural parameterization. Additionally, for the standard parameterization the predicted mean follows a linear trend, while for the natural parameterization it resorts to a constant value. As the natural parameterization lead to much more stable results throughout all our experiments, we exclusively consider the natural parameterization as training objective in the reported results.

## E.2   SYNTHETIC EXPERIMENT

The ground-truth conditional is a heteroscedastic two-component mixture,

$$p^*(y \mid x) = \pi(x)P_1(y \mid x) + \big(1 - \pi(x)\big)P_2(y \mid x),$$

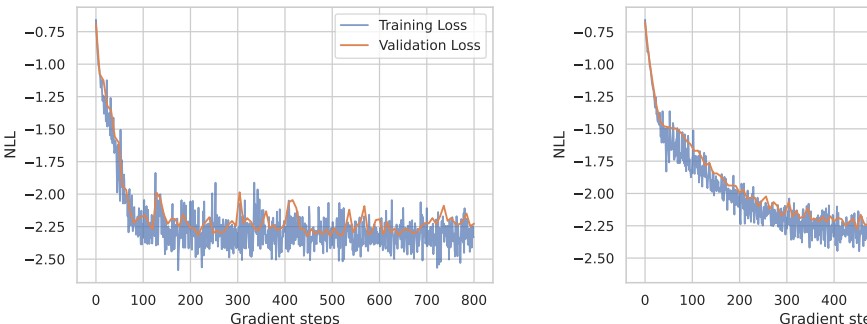

Figure 4: Training and validation loss when minimizing the Gaussian NLL on a synthetic regression example. **Left:** Training with the standard parameterization (equation (88)). **Right:** Training with the natural parameterization (equation (89)).

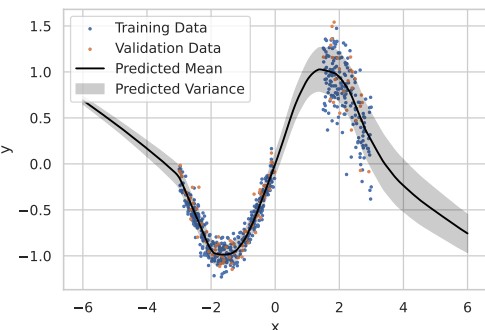 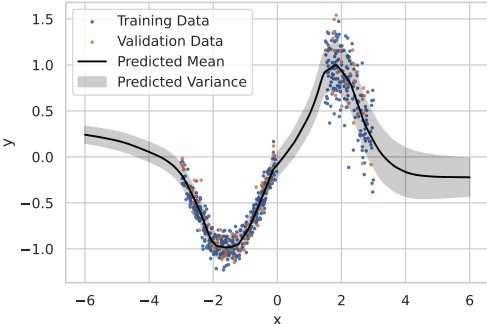

Figure 5: Best model when minimizing the Gaussian NLL on a synthetic regression example. **Left:** Predicting with the standard parameterization (equation (88)). **Right:** Predicting with the natural parameterization (equation (89)).

with mixing weight and component means

$$\pi(x) = \frac{1}{1 + \exp(1.2x)},$$
$$\mu_1(x) = \tfrac{x}{3} + 1.2\sin(0.8x),$$
$$\mu_2(x) = \tfrac{x}{3} - 1.2\cos(0.8x),$$

and input-dependent noise scale

$$\sigma(x) = 0.12 + 0.28\big(0.5 + 0.5\sin(0.7x)\big)^2.$$

We take the components to be Gaussian with shared heteroscedastic variance,

$$P_1(y \mid x) = \mathcal{N}\big(y; \mu_1(x), \sigma^2(x)\big), \qquad P_2(y \mid x) = \mathcal{N}\big(y; \mu_2(x), \sigma^2(x)\big),$$

equivalently $y = \mu_k(x) + \epsilon\sigma(x)$ with $\epsilon \sim \mathcal{N}(0, 1)$ and $k \in \{1, 2\}$ drawn according to $\pi(x)$. Figure 6 shows the ground-truth conditional and samples.

We draw $n = 1200$ training pairs from $p^*(y \mid x)$ and fit a regression network. The backbone is a fully connected MLP with two hidden layers of width 8 and SiLU (Hendrycks & Gimpel, 2016). On top of the backbone, we use two output heads that parameterize a Gaussian predictive distribution via its natural parameters (Immer et al., 2023), and we train by maximizing the corresponding Gaussian log-likelihood. We form an ensemble of 10 independently initialized models, each trained for 100 epochs with Adam (Kingma & Ba, 2014).

Given the trained ensemble, we compute our uncertainty scores. Figure 2 in the main part of the paper reports one instance of our framework using the logarithmic score: each dot shows the ensemble-averaged predictive mean, and the color intensity encodes the corresponding uncertainty (see color bar).

Below we extend the main-text visualization to additional scoring rules (CRPS, SE, and Quadratic). Across all rules, Bayes risk is highest in regions where the mapping is least determined, i.e., where the two generating curves are sampled with comparable probability, while excess risk rises for inputs outside the training data support, reflecting increased epistemic uncertainty. Although absolute magnitudes vary across rules (a score-specific effect), the spatial patterns are consistent. Extended plots are shown in Figures 7-8.

### E.3 SELECTIVE PREDICTION

**Datasets.** We evaluate on seven datasets. First, we introduce `dots`, where inputs are $32 \times 32$ grayscale images containing non-overlapping dots and targets are the corresponding dot counts. Second, we introduce `arrow`, where inputs are $32 \times 32$ grayscale images of left-pointing arrows and targets are the arrow angles. Third, we derive five datasets from Cityscapes (Cordts et al.,

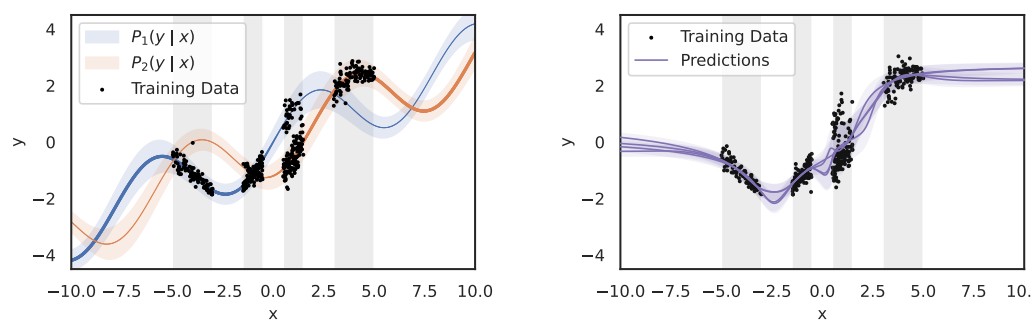

Figure 6: Synthetic regression setup. **Left:** Mixture components $P_1$ and $P_2$ of the ground-truth conditional $p^*(y \mid x)$; curve thickness encodes the mixing weights $\pi(x)$ and $1 - \pi(x)$, respectively. **Right:** Training samples drawn from $p^*$ and predictive distributions of individual ensemble members.

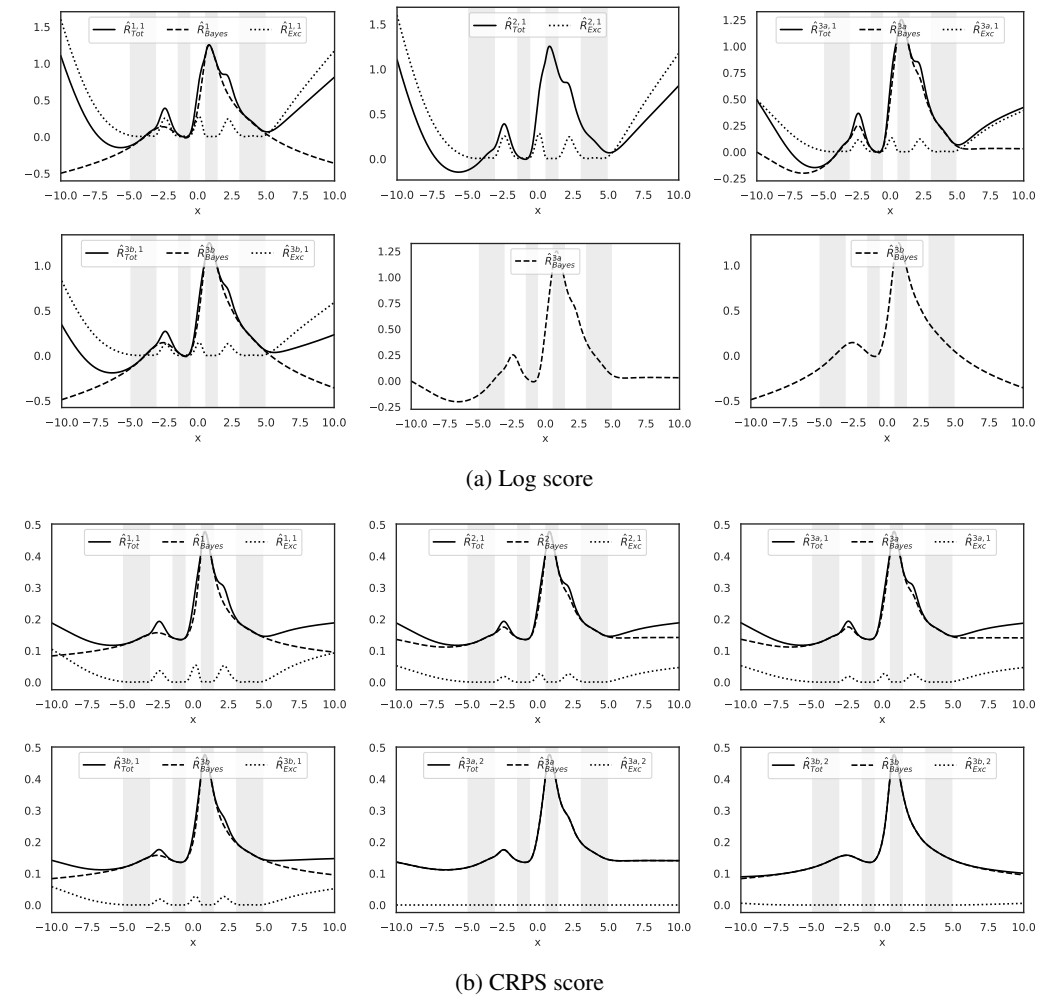

Figure 7: Synthetic 1D experiment (part 1): Log and CRPS score.

2016) by tiling each image into $224 \times 224$ RGB crops and using the provided segmentation maps to define regression targets: the number of pixels belonging to a given class. The classes are `street`, `building`, `sky`, `car`, and `vegetation`.

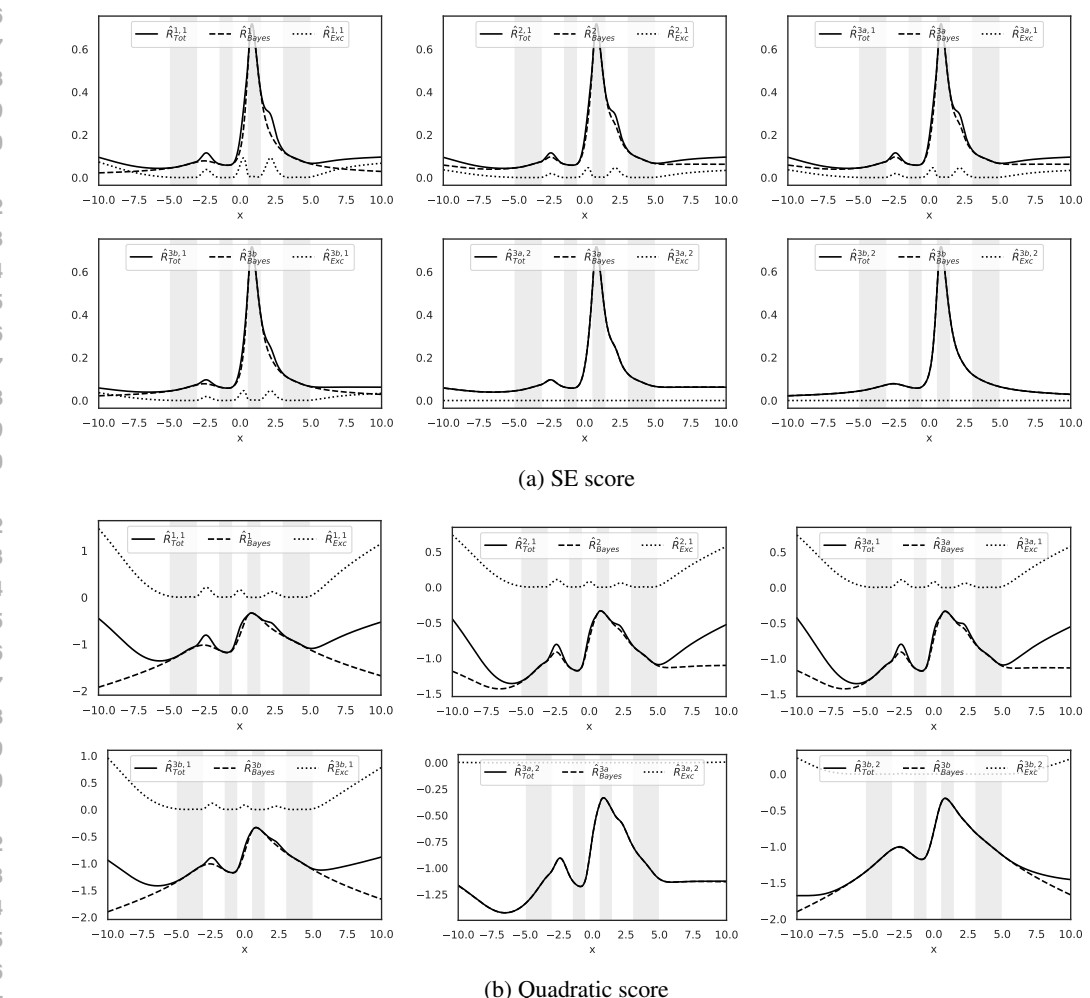

(a) SE score

(b) Quadratic score

Figure 8: Synthetic 1D experiment (part 2): SE and Quadratic score.

For train–test splits, we generate new samples for `dots` and `arrow`, and use the official Cityscapes test split (Cordts et al., 2016) for the remaining datasets.

**Training details.** For `dots` and `arrow`, we use a small CNN with three convolutional layers followed by two fully connected layers and ReLU activations. Models are trained with Adam (Kingma & Ba, 2014), learning rate $10^{-3}$, and batch size 64. For the Cityscapes-derived datasets, we train ResNet-18 (He et al., 2016) with the same optimizer settings. Unless stated otherwise, we report results from ensembles of 10 independently trained models per dataset.

**Extended results.** Selective prediction is quantified using the prediction–reject ratio (PRR; Malinin & Gales, 2021). Although introduced for classification, PRR extends directly to regression by fixing the performance metric used to compute areas; we use mean squared error (MSE). We consider retention rates (i.e., $1-$rejection) from 0.5 to 1. Example retention curves with PRR are shown in Figure 9 (insets), and full per-dataset results appear in Table 9.

### E.4 OUT-OF-DISTRIBUTION DETECTION

**Datasets.** We construct inputs as $64 \times 64$ grayscale mosaics of four MNIST digits (Lecun et al., 1998) arranged (upper-left, upper-right, bottom-left, bottom-right) to form a four-digit number; the target is this number. For out-of-distribution (OOD) data, we substitute each quadrant with images

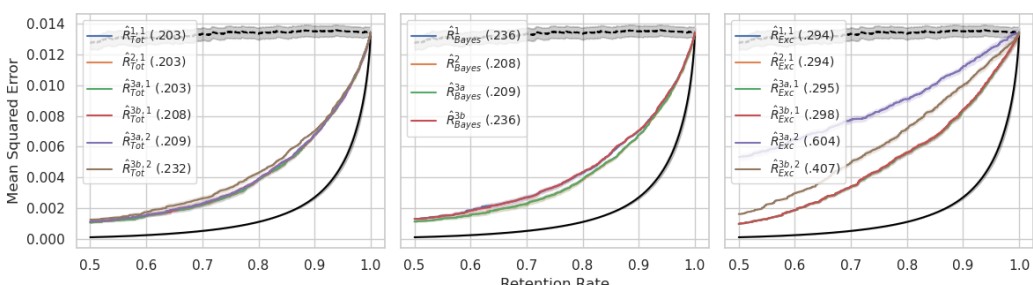

Figure 9: Exemplary retention curve for `street` dataset, PRRs reported in the inset. Black solid line is the optimal, oracle retention curved sorted by the actual MSE of each prediction. The black dashed line is the random baseline.

Table 9: Results for selective prediction (PRR↓) on different datasets $\mathcal{D}$ for different scoring rules (SR).

| $\mathcal{D}$ | SR | $\widehat{R}^{1,1}_{Tot}$ | $\widehat{R}^{2,1}_{Tot}$ | $\widehat{R}^{3a,1}_{Tot}$ | $\widehat{R}^{3b,1}_{Tot}$ | $\widehat{R}^{3a,2}_{Tot}$ | $\widehat{R}^{3b,2}_{Tot}$ | $\widehat{R}^{1}_{Bayes}$ | $\widehat{R}^{2}_{Bayes}$ | $\widehat{R}^{3a}_{Bayes}$ | $\widehat{R}^{3b}_{Bayes}$ | $\widehat{R}^{1,1}_{Exc}$ | $\widehat{R}^{2,1}_{Exc}$ | $\widehat{R}^{3a,1}_{Exc}$ | $\widehat{R}^{3b,1}_{Exc}$ | $\widehat{R}^{3a,2}_{Exc}$ | $\widehat{R}^{3b,2}_{Exc}$ |
|---|---|---|---|---|---|---|---|---|---|---|---|---|---|---|---|---|---|
| dots | CRPS | 0.631 ±.002 | 0.631 ±.002 | 0.631 ±.002 | 0.644 ±.002 | 0.644 ±.002 | 0.664 ±.002 | 0.664 ±.002 | 0.644 ±.002 | 0.644 ±.002 | 0.664 ±.002 | 0.704 ±.001 | 0.704 ±.001 | 0.704 ±.001 | 0.705 ±.001 | 0.782 ±.002 | 0.716 ±.001 |
| | Log | 0.631 ±.002 | 0.631 ±.002 | 0.631 ±.002 | 0.643 ±.002 | - | - | 0.664 ±.002 | - | 0.644 ±.002 | 0.664 ±.002 | 0.729 ±.001 | 0.729 ±.001 | 0.729 ±.001 | 0.729 ±.001 | - | - |
| | SE | 0.632 ±.002 | 0.632 ±.002 | 0.632 ±.002 | 0.644 ±.002 | 0.644 ±.002 | 0.664 ±.002 | 0.664 ±.002 | 0.644 ±.002 | 0.644 ±.002 | 0.664 ±.002 | 0.685 ±.001 | 0.685 ±.001 | 0.685 ±.001 | 0.685 ±.001 | - | - |
| | Quad. | 0.631 ±.002 | 0.631 ±.002 | 0.631 ±.002 | 0.643 ±.002 | 0.644 ±.002 | 0.663 ±.002 | 0.664 ±.002 | 0.644 ±.002 | 0.644 ±.002 | 0.664 ±.002 | 0.760 ±.002 | 0.760 ±.002 | 0.760 ±.002 | 0.759 ±.002 | 0.805 ±.002 | 0.743 ±.001 |
| arrow | CRPS | 0.776 ±.017 | 0.776 ±.017 | 0.776 ±.017 | 0.785 ±.018 | 0.786 ±.018 | 0.806 ±.018 | 0.810 ±.017 | 0.786 ±.018 | 0.786 ±.018 | 0.810 ±.017 | 0.756 ±.012 | 0.756 ±.012 | 0.756 ±.012 | 0.756 ±.012 | 0.804 ±.014 | 0.762 ±.011 |
| | Log | 0.773 ±.017 | 0.773 ±.017 | 0.775 ±.017 | 0.783 ±.018 | - | - | 0.811 ±.017 | - | 0.786 ±.018 | 0.810 ±.017 | 0.765 ±.010 | 0.765 ±.010 | 0.765 ±.010 | 0.766 ±.011 | - | - |
| | SE | 0.777 ±.017 | 0.777 ±.017 | 0.777 ±.017 | 0.786 ±.018 | 0.786 ±.018 | 0.810 ±.017 | 0.810 ±.017 | 0.786 ±.018 | 0.786 ±.018 | 0.810 ±.017 | 0.753 ±.013 | 0.753 ±.013 | 0.753 ±.013 | 0.753 ±.013 | - | - |
| | Quad. | 0.774 ±.017 | 0.774 ±.017 | 0.774 ±.017 | 0.784 ±.018 | 0.785 ±.018 | 0.801 ±.018 | 0.811 ±.017 | 0.786 ±.018 | 0.786 ±.018 | 0.810 ±.017 | 0.783 ±.010 | 0.783 ±.010 | 0.783 ±.010 | 0.782 ±.010 | 0.815 ±.014 | 0.776 ±.010 |
| street | CRPS | 0.203 ±.013 | 0.203 ±.013 | 0.204 ±.013 | 0.209 ±.014 | 0.209 ±.014 | 0.232 ±.014 | 0.236 ±.014 | 0.209 ±.014 | 0.209 ±.014 | 0.237 ±.014 | 0.295 ±.012 | 0.295 ±.012 | 0.295 ±.012 | 0.298 ±.012 | 0.605 ±.016 | 0.407 ±.012 |
| | Log | 0.215 ±.013 | 0.215 ±.013 | 0.206 ±.013 | 0.209 ±.014 | - | - | 0.237 ±.014 | - | 0.209 ±.014 | 0.237 ±.014 | 0.479 ±.014 | 0.479 ±.014 | 0.488 ±.014 | 0.488 ±.014 | - | - |
| | SE | 0.204 ±.013 | 0.204 ±.013 | 0.204 ±.013 | 0.209 ±.014 | 0.209 ±.014 | 0.237 ±.014 | 0.237 ±.014 | 0.209 ±.014 | 0.209 ±.014 | 0.237 ±.014 | 0.238 ±.011 | 0.238 ±.011 | 0.238 ±.011 | 0.238 ±.011 | - | - |
| | Quad. | 0.206 ±.013 | 0.206 ±.013 | 0.211 ±.013 | 0.208 ±.014 | 0.210 ±.014 | 0.228 ±.014 | 0.238 ±.014 | 0.208 ±.014 | 0.209 ±.014 | 0.237 ±.014 | 0.896 ±.010 | 0.896 ±.010 | 0.907 ±.011 | 0.883 ±.011 | 1.063 ±.011 | 0.787 ±.013 |
| building | CRPS | 0.227 ±.012 | 0.227 ±.012 | 0.227 ±.012 | 0.235 ±.012 | 0.236 ±.012 | 0.264 ±.012 | 0.273 ±.012 | 0.237 ±.012 | 0.236 ±.012 | 0.270 ±.012 | 0.240 ±.011 | 0.240 ±.011 | 0.242 ±.011 | 0.242 ±.011 | 0.551 ±.012 | 0.318 ±.013 |
| | Log | 0.227 ±.012 | 0.227 ±.012 | 0.227 ±.012 | 0.232 ±.012 | - | - | 0.277 ±.012 | - | 0.236 ±.012 | 0.270 ±.012 | 0.336 ±.013 | 0.336 ±.013 | 0.338 ±.013 | 0.332 ±.013 | - | - |
| | SE | 0.228 ±.012 | 0.228 ±.012 | 0.228 ±.012 | 0.236 ±.012 | 0.236 ±.012 | 0.270 ±.012 | 0.270 ±.012 | 0.236 ±.012 | 0.236 ±.012 | 0.270 ±.012 | 0.229 ±.011 | 0.229 ±.011 | 0.229 ±.011 | 0.229 ±.011 | - | - |
| | Quad. | 0.226 ±.012 | 0.226 ±.012 | 0.227 ±.012 | 0.233 ±.013 | 0.235 ±.012 | 0.257 ±.013 | 0.282 ±.011 | 0.242 ±.013 | 0.236 ±.012 | 0.270 ±.012 | 0.517 ±.012 | 0.517 ±.012 | 0.538 ±.012 | 0.523 ±.012 | 0.888 ±.011 | 0.568 ±.013 |
| sky | CRPS | 0.043 ±.003 | 0.043 ±.003 | 0.044 ±.003 | 0.045 ±.004 | 0.045 ±.004 | 0.050 ±.006 | 0.051 ±.006 | 0.044 ±.004 | 0.045 ±.004 | 0.052 ±.006 | 0.044 ±.001 | 0.044 ±.001 | 0.045 ±.001 | 0.044 ±.001 | 0.099 ±.004 | 0.050 ±.001 |
| | Log | 0.046 ±.003 | 0.046 ±.003 | 0.045 ±.003 | 0.046 ±.004 | - | - | 0.050 ±.006 | - | 0.045 ±.004 | 0.052 ±.006 | 0.054 ±.001 | 0.054 ±.001 | 0.057 ±.001 | 0.056 ±.001 | - | - |
| | SE | 0.044 ±.004 | 0.044 ±.004 | 0.044 ±.004 | 0.045 ±.004 | 0.045 ±.004 | 0.052 ±.006 | 0.052 ±.006 | 0.045 ±.004 | 0.045 ±.004 | 0.052 ±.006 | 0.043 ±.002 | 0.043 ±.002 | 0.043 ±.002 | 0.043 ±.002 | - | - |
| | Quad. | 0.041 ±.003 | 0.041 ±.003 | 0.045 ±.003 | 0.044 ±.004 | 0.046 ±.004 | 0.048 ±.006 | 0.049 ±.006 | 0.042 ±.004 | 0.045 ±.004 | 0.052 ±.006 | 0.075 ±.003 | 0.075 ±.003 | 0.079 ±.003 | 0.073 ±.003 | 0.172 ±.004 | 0.072 ±.002 |
| car | CRPS | 0.149 ±.003 | 0.149 ±.003 | 0.154 ±.003 | 0.162 ±.003 | 0.162 ±.003 | 0.189 ±.004 | 0.186 ±.005 | 0.156 ±.003 | 0.161 ±.003 | 0.196 ±.005 | 0.151 ±.006 | 0.151 ±.006 | 0.154 ±.006 | 0.154 ±.006 | 0.347 ±.008 | 0.195 ±.008 |
| | Log | 0.176 ±.004 | 0.176 ±.004 | 0.163 ±.003 | 0.172 ±.003 | - | - | 0.177 ±.004 | - | 0.161 ±.003 | 0.196 ±.005 | 0.219 ±.007 | 0.219 ±.007 | 0.226 ±.008 | 0.219 ±.009 | - | - |
| | SE | 0.153 ±.003 | 0.153 ±.003 | 0.153 ±.003 | 0.161 ±.003 | 0.161 ±.003 | 0.196 ±.005 | 0.196 ±.005 | 0.161 ±.003 | 0.161 ±.003 | 0.196 ±.005 | 0.141 ±.005 | 0.141 ±.005 | 0.141 ±.005 | 0.141 ±.005 | - | - |
| | Quad. | 0.153 ±.005 | 0.153 ±.005 | 0.174 ±.004 | 0.174 ±.003 | 0.170 ±.003 | 0.187 ±.004 | 0.170 ±.004 | 0.148 ±.004 | 0.161 ±.003 | 0.196 ±.005 | 0.352 ±.016 | 0.352 ±.016 | 0.377 ±.014 | 0.342 ±.015 | 0.617 ±.012 | 0.334 ±.013 |
| vegetation | CRPS | 0.200 ±.011 | 0.200 ±.011 | 0.199 ±.011 | 0.209 ±.011 | 0.210 ±.011 | 0.260 ±.011 | 0.274 ±.012 | 0.211 ±.011 | 0.210 ±.011 | 0.272 ±.011 | 0.187 ±.009 | 0.187 ±.009 | 0.186 ±.009 | 0.187 ±.009 | 0.337 ±.005 | 0.199 ±.008 |
| | Log | 0.195 ±.011 | 0.195 ±.011 | 0.197 ±.011 | 0.203 ±.010 | - | - | 0.276 ±.012 | - | 0.210 ±.011 | 0.272 ±.011 | 0.195 ±.008 | 0.195 ±.008 | 0.195 ±.008 | 0.196 ±.008 | - | - |
| | SE | 0.200 ±.011 | 0.200 ±.011 | 0.200 ±.011 | 0.210 ±.011 | 0.210 ±.011 | 0.272 ±.011 | 0.272 ±.011 | 0.210 ±.011 | 0.210 ±.011 | 0.272 ±.011 | 0.187 ±.011 | 0.187 ±.011 | 0.187 ±.011 | 0.187 ±.011 | - | - |
| | Quad. | 0.199 ±.012 | 0.199 ±.012 | 0.196 ±.011 | 0.204 ±.011 | 0.209 ±.011 | 0.247 ±.011 | 0.278 ±.012 | 0.214 ±.012 | 0.210 ±.011 | 0.272 ±.011 | 0.217 ±.008 | 0.217 ±.008 | 0.216 ±.008 | 0.216 ±.008 | 0.462 ±.005 | 0.241 ±.008 |

from CIFAR-10 (Krizhevsky & Hinton, 2009), SVHN (Netzer et al., 2011), or Fashion-MNIST (Xiao et al., 2017), as well as a mixture of these three sources. We also perform a fine-grained analysis of positional effects by replacing exactly one quadrant at a time with EMNIST (Cohen et al., 2017) (upper-left, upper-right, bottom-left, or bottom-right), keeping the others from MNIST.

**Training details.** Across experiments, we use the same small CNN as in the selective prediction setup, with a larger first linear layer to accommodate the increased input size. We keep hyperparameters

Table 10: Out-of-distribution detection results (AUROC↑). The ID dataset is MNIST mosaic, where four MNIST digits are tiled in a grid with each corresponding to a digit of the target value. OOD datasets use the corresponding datasets as sources for the tiles. The EMNIST variations replace just one tile with a letter from the EMNIST dataset.

| $\mathcal{D}_{ood}$ | SR | $\widehat{R}_{Tot}^{1,1}$ | $\widehat{R}_{Tot}^{2,1}$ | $\widehat{R}_{Tot}^{3a,1}$ | $\widehat{R}_{Tot}^{3b,1}$ | $\widehat{R}_{Tot}^{3a,2}$ | $\widehat{R}_{Tot}^{3b,2}$ | $\widehat{R}_{Bayes}^{1}$ | $\widehat{R}_{Bayes}^{2}$ | $\widehat{R}_{Bayes}^{3a}$ | $\widehat{R}_{Bayes}^{3b}$ | $\widehat{R}_{Exc}^{1,1}$ | $\widehat{R}_{Exc}^{2,1}$ | $\widehat{R}_{Exc}^{3a,1}$ | $\widehat{R}_{Exc}^{3b,1}$ | $\widehat{R}_{Exc}^{3a,2}$ | $\widehat{R}_{Exc}^{3b,2}$ |
|---|---|---|---|---|---|---|---|---|---|---|---|---|---|---|---|---|---|
| CIFAR-10 | CRPS | 0.969 ±.018 | 0.969 ±.018 | 0.969 ±.018 | 0.953 ±.022 | 0.949 ±.021 | 0.864 ±.006 | 0.831 ±.015 | 0.949 ±.021 | 0.949 ±.021 | 0.832 ±.015 | 0.984 ±.011 | 0.984 ±.011 | 0.984 ±.011 | 0.985 ±.011 | 0.963 ±.016 | 0.984 ±.012 |
| | Log | 0.977 ±.016 | 0.977 ±.016 | 0.972 ±.017 | 0.962 ±.022 | - | - | 0.830 ±.015 | - | 0.949 ±.021 | 0.832 ±.015 | 0.984 ±.012 | 0.984 ±.012 | 0.984 ±.013 | 0.984 ±.013 | - | - |
| | SE | 0.966 ±.018 | 0.966 ±.018 | 0.966 ±.018 | 0.949 ±.021 | 0.949 ±.021 | 0.832 ±.015 | 0.832 ±.021 | 0.949 ±.021 | 0.949 ±.021 | 0.832 ±.015 | 0.982 ±.011 | 0.982 ±.011 | 0.982 ±.011 | 0.982 ±.011 | - | - |
| | Quad. | 0.976 ±.017 | 0.976 ±.017 | 0.976 ±.017 | 0.962 ±.022 | 0.950 ±.021 | 0.893 ±.018 | 0.829 ±.016 | 0.949 ±.021 | 0.949 ±.021 | 0.832 ±.015 | 0.972 ±.022 | 0.972 ±.022 | 0.972 ±.022 | 0.974 ±.021 | 0.950 ±.019 | 0.980 ±.017 |
| SVHN | CRPS | 0.984 ±.011 | 0.984 ±.011 | 0.983 ±.011 | 0.978 ±.013 | 0.976 ±.012 | 0.938 ±.000 | 0.919 ±.012 | 0.976 ±.013 | 0.976 ±.012 | 0.920 ±.012 | 0.986 ±.012 | 0.986 ±.012 | 0.986 ±.012 | 0.986 ±.012 | 0.964 ±.022 | 0.983 ±.016 |
| | Log | 0.986 ±.011 | 0.986 ±.011 | 0.985 ±.011 | 0.981 ±.013 | - | - | 0.919 ±.013 | - | 0.976 ±.012 | 0.920 ±.012 | 0.980 ±.018 | 0.980 ±.018 | 0.980 ±.019 | 0.979 ±.019 | - | - |
| | SE | 0.982 ±.011 | 0.982 ±.011 | 0.982 ±.011 | 0.976 ±.012 | 0.976 ±.012 | 0.920 ±.012 | 0.920 ±.012 | 0.976 ±.012 | 0.976 ±.012 | 0.920 ±.012 | 0.987 ±.010 | 0.987 ±.010 | 0.987 ±.010 | 0.987 ±.010 | - | - |
| | Quad. | 0.986 ±.011 | 0.986 ±.011 | 0.986 ±.011 | 0.981 ±.014 | 0.976 ±.013 | 0.953 ±.010 | 0.918 ±.013 | 0.976 ±.013 | 0.976 ±.012 | 0.920 ±.012 | 0.950 ±.045 | 0.950 ±.045 | 0.950 ±.045 | 0.954 ±.043 | 0.943 ±.034 | 0.968 ±.031 |
| Fashion-MNIST | CRPS | 0.936 ±.014 | 0.936 ±.014 | 0.936 ±.014 | 0.911 ±.018 | 0.908 ±.018 | 0.825 ±.025 | 0.804 ±.028 | 0.908 ±.017 | 0.907 ±.018 | 0.805 ±.028 | 0.971 ±.010 | 0.971 ±.010 | 0.971 ±.010 | 0.972 ±.010 | 0.936 ±.005 | 0.973 ±.010 |
| | Log | 0.950 ±.012 | 0.950 ±.012 | 0.941 ±.013 | 0.922 ±.016 | - | - | 0.804 ±.028 | - | 0.907 ±.018 | 0.805 ±.028 | 0.974 ±.011 | 0.974 ±.011 | 0.974 ±.011 | 0.973 ±.011 | - | - |
| | SE | 0.932 ±.014 | 0.932 ±.014 | 0.932 ±.014 | 0.907 ±.018 | 0.907 ±.018 | 0.805 ±.028 | 0.805 ±.028 | 0.907 ±.018 | 0.907 ±.018 | 0.805 ±.028 | 0.966 ±.010 | 0.966 ±.010 | 0.966 ±.010 | 0.966 ±.010 | - | - |
| | Quad. | 0.946 ±.012 | 0.946 ±.012 | 0.947 ±.013 | 0.921 ±.016 | 0.908 ±.017 | 0.844 ±.023 | 0.803 ±.028 | 0.907 ±.017 | 0.907 ±.018 | 0.805 ±.028 | 0.963 ±.017 | 0.963 ±.017 | 0.963 ±.017 | 0.964 ±.017 | 0.922 ±.006 | 0.970 ±.013 |
| Mixture (C,S,F) | CRPS | 0.963 ±.013 | 0.963 ±.013 | 0.963 ±.013 | 0.947 ±.015 | 0.944 ±.015 | 0.875 ±.010 | 0.851 ±.017 | 0.944 ±.015 | 0.944 ±.015 | 0.852 ±.017 | 0.981 ±.010 | 0.981 ±.010 | 0.981 ±.010 | 0.981 ±.010 | 0.955 ±.014 | 0.980 ±.012 |
| | Log | 0.971 ±.012 | 0.971 ±.012 | 0.966 ±.013 | 0.955 ±.015 | - | - | 0.850 ±.017 | - | 0.944 ±.015 | 0.852 ±.017 | 0.979 ±.013 | 0.979 ±.013 | 0.979 ±.014 | 0.979 ±.014 | - | - |
| | SE | 0.960 ±.013 | 0.960 ±.013 | 0.960 ±.013 | 0.944 ±.015 | 0.944 ±.015 | 0.852 ±.017 | 0.852 ±.017 | 0.944 ±.015 | 0.944 ±.015 | 0.852 ±.017 | 0.979 ±.010 | 0.979 ±.010 | 0.979 ±.010 | 0.979 ±.010 | - | - |
| | Quad. | 0.969 ±.012 | 0.969 ±.012 | 0.970 ±.012 | 0.954 ±.015 | 0.945 ±.015 | 0.896 ±.013 | 0.849 ±.017 | 0.944 ±.015 | 0.944 ±.015 | 0.852 ±.017 | 0.962 ±.028 | 0.962 ±.028 | 0.962 ±.028 | 0.964 ±.027 | 0.938 ±.019 | 0.973 ±.020 |
| EMNIST (tl) | CRPS | 0.836 ±.012 | 0.836 ±.012 | 0.836 ±.012 | 0.800 ±.012 | 0.797 ±.012 | 0.718 ±.013 | 0.709 ±.014 | 0.797 ±.012 | 0.796 ±.014 | 0.709 ±.012 | 0.915 ±.012 | 0.915 ±.012 | 0.915 ±.012 | 0.915 ±.012 | 0.846 ±.020 | 0.919 ±.012 |
| | Log | 0.855 ±.012 | 0.855 ±.012 | 0.842 ±.012 | 0.811 ±.012 | - | - | 0.709 ±.014 | - | 0.796 ±.012 | 0.709 ±.014 | 0.922 ±.011 | 0.922 ±.011 | 0.922 ±.011 | 0.921 ±.011 | - | - |
| | SE | 0.831 ±.012 | 0.831 ±.012 | 0.831 ±.012 | 0.796 ±.012 | 0.796 ±.012 | 0.709 ±.014 | 0.709 ±.014 | 0.796 ±.012 | 0.796 ±.014 | 0.709 ±.012 | 0.903 ±.012 | 0.903 ±.012 | 0.903 ±.012 | 0.903 ±.012 | - | - |
| | Quad. | 0.850 ±.012 | 0.850 ±.012 | 0.850 ±.012 | 0.809 ±.012 | 0.797 ±.012 | 0.731 ±.013 | 0.708 ±.014 | 0.797 ±.012 | 0.796 ±.012 | 0.709 ±.014 | 0.910 ±.011 | 0.910 ±.011 | 0.910 ±.011 | 0.911 ±.011 | 0.827 ±.026 | 0.919 ±.012 |
| EMNIST (tr) | CRPS | 0.559 ±.010 | 0.559 ±.010 | 0.559 ±.010 | 0.550 ±.009 | 0.550 ±.009 | 0.534 ±.007 | 0.533 ±.007 | 0.550 ±.009 | 0.550 ±.009 | 0.533 ±.007 | 0.594 ±.014 | 0.594 ±.014 | 0.594 ±.014 | 0.594 ±.014 | 0.580 ±.010 | 0.595 ±.013 |
| | Log | 0.563 ±.011 | 0.563 ±.011 | 0.561 ±.010 | 0.552 ±.009 | - | - | 0.533 ±.007 | - | 0.550 ±.009 | 0.533 ±.007 | 0.600 ±.016 | 0.600 ±.016 | 0.600 ±.016 | 0.599 ±.016 | - | - |
| | SE | 0.558 ±.010 | 0.558 ±.010 | 0.558 ±.010 | 0.550 ±.009 | 0.550 ±.009 | 0.533 ±.007 | 0.533 ±.007 | 0.550 ±.009 | 0.550 ±.009 | 0.533 ±.007 | 0.589 ±.012 | 0.589 ±.012 | 0.589 ±.012 | 0.589 ±.012 | - | - |
| | Quad. | 0.562 ±.010 | 0.562 ±.010 | 0.562 ±.010 | 0.552 ±.009 | 0.550 ±.009 | 0.536 ±.007 | 0.533 ±.007 | 0.550 ±.009 | 0.550 ±.009 | 0.533 ±.007 | 0.598 ±.016 | 0.598 ±.016 | 0.599 ±.016 | 0.598 ±.016 | 0.578 ±.009 | 0.597 ±.014 |
| EMNIST (bl) | CRPS | 0.557 ±.022 | 0.557 ±.022 | 0.557 ±.022 | 0.550 ±.021 | 0.549 ±.021 | 0.535 ±.019 | 0.534 ±.018 | 0.549 ±.021 | 0.549 ±.018 | 0.534 ±.021 | 0.591 ±.022 | 0.591 ±.022 | 0.591 ±.022 | 0.591 ±.022 | 0.561 ±.013 | 0.593 ±.021 |
| | Log | 0.560 ±.022 | 0.560 ±.022 | 0.558 ±.022 | 0.551 ±.021 | - | - | 0.534 ±.018 | - | 0.549 ±.021 | 0.534 ±.018 | 0.595 ±.021 | 0.595 ±.021 | 0.594 ±.021 | 0.594 ±.021 | - | - |
| | SE | 0.556 ±.021 | 0.556 ±.021 | 0.556 ±.021 | 0.549 ±.021 | 0.549 ±.021 | 0.534 ±.018 | 0.534 ±.021 | 0.549 ±.021 | 0.549 ±.018 | 0.534 ±.021 | 0.586 ±.022 | 0.586 ±.022 | 0.586 ±.022 | 0.586 ±.022 | - | - |
| | Quad. | 0.560 ±.022 | 0.560 ±.022 | 0.560 ±.022 | 0.551 ±.021 | 0.549 ±.021 | 0.537 ±.019 | 0.534 ±.018 | 0.549 ±.021 | 0.549 ±.021 | 0.534 ±.018 | 0.593 ±.018 | 0.593 ±.018 | 0.593 ±.018 | 0.593 ±.018 | 0.557 ±.014 | 0.595 ±.019 |
| EMNIST (br) | CRPS | 0.548 ±.013 | 0.548 ±.013 | 0.548 ±.013 | 0.541 ±.014 | 0.541 ±.013 | 0.529 ±.013 | 0.528 ±.013 | 0.541 ±.013 | 0.541 ±.013 | 0.528 ±.013 | 0.577 ±.010 | 0.577 ±.010 | 0.577 ±.010 | 0.577 ±.010 | 0.553 ±.019 | 0.577 ±.009 |
| | Log | 0.551 ±.013 | 0.551 ±.013 | 0.549 ±.013 | 0.542 ±.014 | - | - | 0.528 ±.013 | - | 0.541 ±.013 | 0.528 ±.013 | 0.581 ±.009 | 0.581 ±.009 | 0.581 ±.009 | 0.581 ±.009 | - | - |
| | SE | 0.547 ±.013 | 0.547 ±.013 | 0.547 ±.013 | 0.541 ±.013 | 0.541 ±.013 | 0.528 ±.013 | 0.528 ±.013 | 0.541 ±.013 | 0.541 ±.013 | 0.528 ±.013 | 0.573 ±.011 | 0.573 ±.011 | 0.573 ±.011 | 0.573 ±.011 | - | - |
| | Quad. | 0.550 ±.013 | 0.550 ±.013 | 0.550 ±.013 | 0.542 ±.014 | 0.541 ±.013 | 0.530 ±.013 | 0.528 ±.013 | 0.541 ±.013 | 0.541 ±.013 | 0.528 ±.013 | 0.579 ±.008 | 0.579 ±.008 | 0.579 ±.008 | 0.579 ±.008 | 0.552 ±.020 | 0.578 ±.008 |

identical to the selective prediction setting, and train ensembles of 10 models on the in-distribution MNIST-mosaic task.

**Extended results.** For evaluation, we generate matched in-distribution and OOD sets of equal size and compute AUROC using uncertainty scores to discriminate OOD from in-distribution samples. Full per-dataset results are listed in Table 10. Replacing all four quadrants with another dataset yields similar detection performance across sources (first four rows), with SVHN—despite also containing digits—detected comparably well. In the positional analysis (last four rows), replacing the upper-left digit (the most significant position in the four-digit number) is detected most reliably, with performance decreasing as the replaced digit's positional significance decreases, which aligns with its influence on the target value.

### E.5 Active Learning

**Datasets.** For the active learning experiments, we considered the following six tabular datasets from the UCI repository. YearPredictionMSD (`YMSD`; Bertin-Mahieux et al., 2011), SGEMM GPU kernel

performance (`SGEMM`; Ballester-Ripoll et al., 2019), Combined Cycle Power Plant (`CCPP`; Tüfekci, 2014), Physicochemical Properties of Protein Tertiary Structure (`CASP`; Rana, 2013), Online News Popularity (`NEWS`; Fernandes et al., 2015) and BlogFeedback (`BLOG`; Buza, 2013). For all datasets, we split 20% into a test set for evaluation. The remaining set was further split into validation and pool set. We selected around 200 samples (exact sizes per dataset in code, which was selected based initial performance compared to performance on full pool set) as a training set to start the active learning experiments.

**Training details.** In each acquisition iteration, we selected a number of additional samples from the pool to augment the training set. We do not strictly select based on the score induced by a given uncertainty measure, but obtain a categorical distribution based on applying the softmax over the uncertainty score for all samples in the pool set. We then sample without replacement according to this distribution. This avoids selecting very similar samples within an acquisition batch, which is a known issue in practice (Kirsch et al., 2019). For these experiments, we utilized ensembles of standard three-layer MLPs with ReLU activations with hidden sizes of 100.

**Results.** The full results are provided in Figure 10. We observe that for most settings, using the uncertainty scores as acquisition function improves upon random sampling. Notable exceptions are total and Bayes risks for the quadratic score, which perform rather poorly.

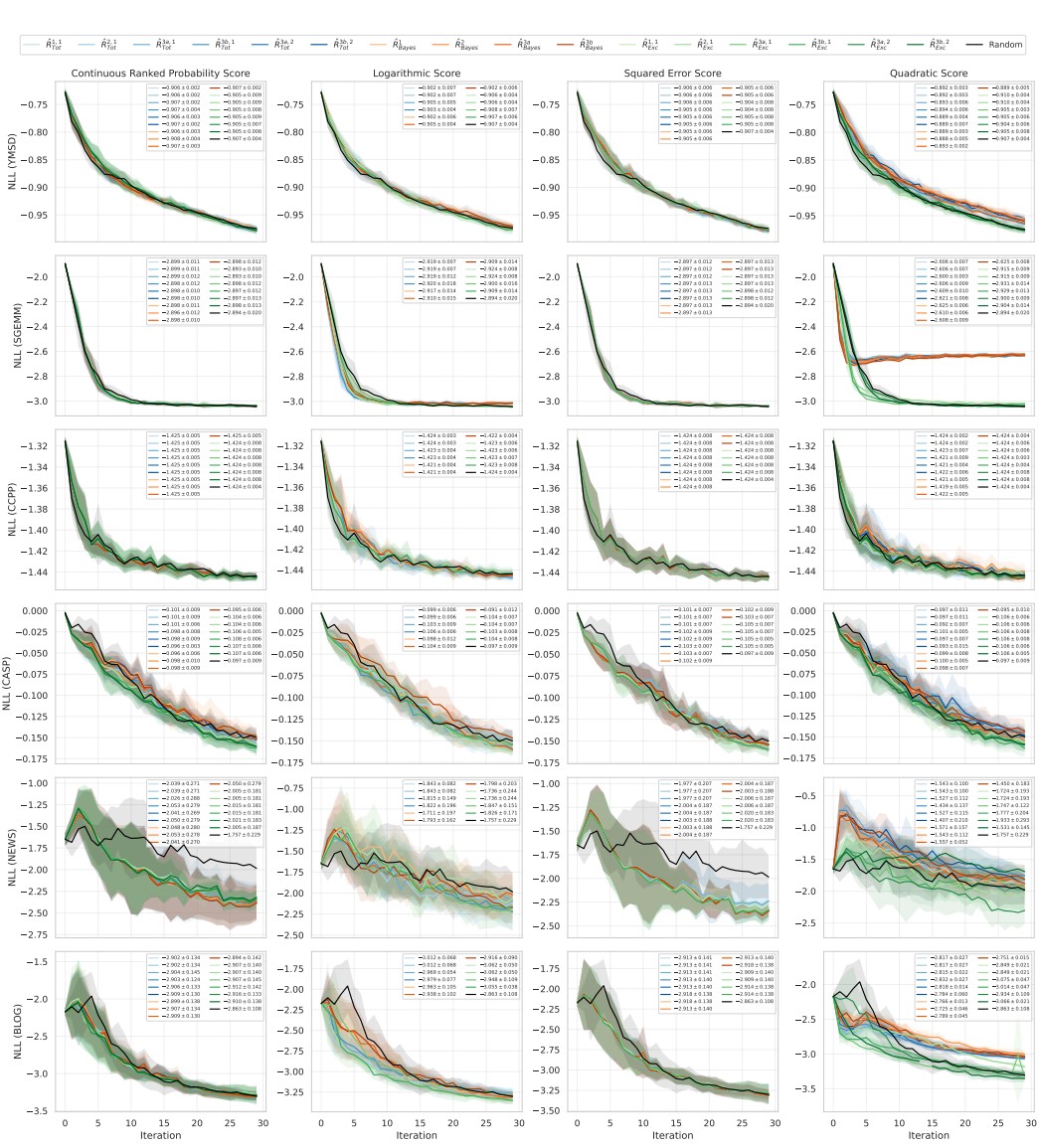

Figure 10: Detailed results on active learning. We report the negative log-likelihood (NLL) of the ensemble after training for each acquisition iteration. In the insets are the average NLLs over the iterations as summary statistic to compare different acquisition functions. Statistics are obtained over five runs.

