# OpenReview forum: "Uncertainty Quantification for Regression Using Proper Scoring Rules"
_ICLR.cc/2026/Conference — Submitted to ICLR 2026_

### Official Review · Reviewer_3MWm · 2025-10-26

**Soundness:** 3
**Presentation:** 3
**Contribution:** 3
**Rating:** 6
**Confidence:** 3

**Summary:**

The paper presents a unified framework for uncertainty quantification with focus on regression problems using proper scoring rules. The authors present theoretical derivation of their uncertainty measures and evaluate all combinations empirically on a set of tasks.

**Strengths:**

- very clear writing. Especially the motivation and background is exceptionally clear and easy to read. This lowers the barrier for the reader to understand the paper greatly.
- an exhaustive and strong generalized framework that extends existing frameworks from classification to regression
- evaluation on a rigorous set of uncertainty quantification tasks

**Weaknesses:**

- Your focus lies in parametric predictive uncertainty quantification with a mixture of Gaussians. While this is a common approach, it limits the applicability of the results to some degree.
- Within your experiments you make empirical observations which metric performs best and should be used. However, there is a lack of connection between empirical results and theory.
- In your experiments you compare performance between your metrics. However, there are no other baseline methods presented.

**Questions:**

Main questions:
- Can you approach be generalized to non-parametric uncertainty quantification or other types of parametetric functions? How do the results generalize?
- Can you connect your empirical findings to theory and therefore generate insights into why specific metrics should be preffered for specific tasks? Or more general, can you provide recommendations which metrics should be used for uncertainty quantificiation in regression in general?
- How do you results compare to other baseline methods?
- What are the limitations of your approach?


Here are some more detailled comments and questions in the order of appearance of the paper (not in order of importance). Some might overlap slightly with weaknesses from above.
- Table 1: For CRPS the symbol $F_\hat{P}$ is not introduced. Also it would be good to give small examples or intuition for each proper scoring rule; possibly in the appendix.
- line 185: with "ensemble of Gaussians", do you mean a mixture of Gaussians or is there a difference?
- line 188 following / estimation of risk: Can you justify why you use these three risk? Meaning what makes them useful and theoretically grounded?
- line 194: The posterior approximation $\bar{P}$ is not used in the written risk estimation $R(\hat R, \hat R_*)$.
- line 209: The "earlier work" under a new set of axioms. Can you explain its significance to show why your generalization is of interest?
- line 217: twice in one sentence "thus".
- line 268: Can you rigorously define the different shifts to the posterior for clarity?
- figure 1: column middle and right:
  - font size too small to be readable.
  - Possibly try to colorcode or otherwise highlight the different risks on the x-axis to simplify readability.
  - What does the gray background color for the Bayes risks indicate?
  - For row 1 and 4 you do not explain what the results show in contrast to row 2/3.
- line 272: What do you mean specifically with "None of the measures considered changes under this shift"?
- section 5.1: I am missing an interpretation of the results and to put it into context with the developed theory. Why are the results the way they are and what does this imply for the application of the specific measures?
- Figure 2:
  - missing description of what the red and blue lines mean and why the dotted risks are not continuous. This also makes the interpreation of the text hard since I do not know what I am looking at.
  - Font too small again.
- Section 5.2: The example with the figure and the text are quite hard to understand and follow. I do not really get what you are doing, what you are showing, and what the results mean. Can you please clarify this whole section?
- Section 5.3: you suggesto to use $\hat R_{Tot}^{1,1}$ for selective prediction. Can you connect that to theory why this specific risk is useful in this task? Are there any insights besides the purely empirical numbers? Similar questions hold for section 5.4

---

> ### Author Response · Authors · 2025-11-22
>
> We would like to thank the reviewer for valuable feedback on our work. Please find the answers to the questions and comment below.
>
> ---
>
> > Your focus lies in parametric predictive uncertainty quantification with a mixture of Gaussians. While this is a common approach, it limits the applicability of the results to some degree.
>
> We agree with the reviewer that we consider a specific parametric predictive distribution in our paper. However, we need to emphasize two important features of the regression.
> First, in contrast to classification, regression offers an additional layer of flexibility. Namely, the choice of the predictive distribution. In classification, the only natural choice is the categorical distribution. However, in the regression, one has more options. Besides the Gaussian, other simple analytical forms of the predictive distribution may be considered, such as log-normal, truncated normal, Laplace, and extreme value distributions -- these are used in meteorology (Baran et al., 2015) and other natural sciences. Another possibility highlighted by reviewers is the use of non-parametric models. Although the general framework remains applicable, we are skeptical that this approach can yield closed-form expressions and/or computationally efficient methods. Sampling-based methods also suffer from computational complexity issues of approximating the involved integrals.
> Second, the Gaussian predictive distribution for a single model is arguably the most popular default choice in practice. Therefore, our results, although derived for the specific parametric case, are applicable to the most popular practical choice.
>
> S. Baran and S. Lerch, Log-normal distribution based Ensemble Model Output Statistics models
> for probabilistic wind-speed forecasting. Quarterly Journal of the Royal Meteorological Society
> 141:2289–2299, 2015.
>
> ---
>
> > Within your experiments you make empirical observations which metric performs best and should be used. However, there is a lack of connection between empirical results and theory.
> > Can you connect your empirical findings to theory and therefore generate insights into why specific metrics should be preffered for specific tasks? Or more general, can you provide recommendations which metrics should be used for uncertainty quantificiation in regression in general?
>
> Thank you for the interesting question. Indeed, we did not provide a theoretical justification for the single best choice of the proper scoring rule and approximation. However, there is probably no theoretical answer to which instantiation will be the best.
> Proper scoring rules generate all instances of our framework. These rules are all equivalent in the sense that minimizing them will place a minimum in the right place. However, the peculiarities of the data and the model in use might provide different benefits to the various instantiations. And therefore lead to a different best performing measure in a particular case.
>
> Based on the feedback from the review stage, we have added a Discussion section to the main text, where we summarize our observed results and provide recommendations to practitioners for selecting the appropriate measure.
>
> Additionally, we want to emphasize that we consider the main contribution of the paper to be methodological, while the experiments are primarily intended to demonstrate that the measures derived from the framework are reasonable. The experiments we conducted indicate that it enables the generation of strong-performing uncertainty measures.
>
> ---
>
> > In your experiments you compare performance between your metrics. However, there are no other baseline methods presented.
> > How do you results compare to other baseline methods?
>
> Thank you for the interesting comment.
> Indeed, we do not have explicit competitors (except a random baseline in the active learning experiment). However, implicitly, we have many of them, as they are instances of our framework.
> Nevertheless, following your suggestion, we plan to consider additional competitors. Specifically, we will add a density-based uncertainty measure for the out-of-distribution detection experiment.
> We do not have the results yet, but will provide them as soon as they are ready.

---

> ### Author Response · Authors · 2025-11-22
>
> > Can you approach be generalized to non-parametric uncertainty quantification or other types of parametetric functions? How do the results generalize?
>
> Yes, the results presented in the paper for the case of the Gaussian predictive distribution can be generalized to other predictive distributions. This includes other parametric predictive distribution options (e.g., Laplace), and even non-parametric (e.g., Nadaraya-Watson regression).
>
> However, we are unsure if closed-form equations can still be derived. And this could be especially problematic in the case of regression, as here, in contrast to classification, where one should compute sums over the terms of the number of classes, here in regression, one should compute integrals. And if closed-form equations are missing, then we have to approximate these integrals, which adds an extra level of stochasticity in addition to the Bayesian integrals (which we approximate with ensembles).
>
> So yes, these generalizations are possible. However, they come at a price of additional integral approximation.
>
> ---
>
> > What are the limitations of your approach?
>
> One limitation of the approach is that it relies on approximating the risk function, which in turn depends on the unknown target distribution. We decided to consider a Bayesian strategy to approximate the function of this unknown quantity. Therefore, a Bayesian model is necessary. This requires either computational resources to fit a posterior distribution or several training cycles to train an ensemble of models.
>
> Another limitation of the particular instantiation of our framework we presented is the assumption that the predictive distribution of a single model is Gaussian. However, this may not be severe considering popular practical choices. Furthermore, we would like to note that one eventually needs to choose a predictive distribution and, therefore, induce some assumptions. The Gaussian assumption is one among a few that allows to derive closed-form equations.
>
> ---
>
> > Table 1: For CRPS the symbol A
> is not introduced. Also it would be good to give small examples or intuition for each proper scoring rule; possibly in the appendix.
>
> We have added the necessary notation to the main text. We will also add the motiovation/intuition of each scoring rule to the appendix, thanks for suggesting this.
>
> > line 185: with "ensemble of Gaussians", do you mean a mixture of Gaussians or is there a difference?
>
> Thanks, we have revised the discription.
>
> ---
>
> > line 188 following / estimation of risk: Can you justify why you use these three risk? Meaning what makes them useful and theoretically grounded?
>
> Our goal is to approximate a function that depends on the unknown distribution p(y|x), which represents the true data-generating process. We do not have access to this distribution, but we can estimate it. Probably the simplest estimate one can make (which is a special case of what we suggest here) is the plug-in method: simply fit a single model and place it within this function.
> However, this single estimate can be arbitrarily bad. Therefore, we consider a Bayesian approach, and assume that we have a posterior distribution over the model's parameters (in our approximation, more specifically, we need only samples from this posterior, i.e., an ensemble of models).
>
> However, it is unclear how to incorporate this posterior distribution for estimating the function of a random variable whose distribution we know. In literature, there are three common ways of doing it. The first one is to simply average plug-in estimates with their corresponding probabilities (which leads to the Bayesian averaging of the risk). The second is what is called the Posterior predictive distribution (or Bayesian Model Averaging when only samples from the posterior are available). Once this estimate is computed, it can be used as a plug-in estimate. These two strategies are generic and have been considered in prior works on classification.
> The third one is specific to the regression setting and was considered in the regression ensemble approximation literature.
>
> ---
>
> > line 194: The posterior approximation $\bar{P}$ is not used in the written risk estimation $R(\hat{P}, \hat{P}\_*)$.
>
> Thanks, we have revised this part.

---

> ### Author Response · Authors · 2025-11-22
>
> > line 209: The "earlier work" under a new set of axioms. Can you explain its significance to show why your generalization is of interest?
>
> Thank you for the great question!
>
> The introduction of axioms is important for the formalization of the field. This helps to develop a unified set of requirements for the uncertainty measures. However, introducing a set of axioms that everyone agrees on is not an easy problem. The specific choice in the paper by (Bulte et. al) is not without questions. For instance, a natural measure of uncertainty, based on the entropy of the predicted distribution, is not valid according to the proposed axioms, as it potentially allows for negative values.
> There is also a line of work where attempts have been made to induce axioms for predictive uncertainty measures in classification. However, the axioms slightly change from paper to paper, which indicates that this is a hard problem to introduce axioms that all agree upon.
>
> For this reason, we did not compare our results with the specific set of axioms proposed in that paper. However, considering axioms and comparing them to the whole class (e.g., uncertainty measures based on risks and proper scoring rules) could be an interesting direction for further research.
>
> Our generalization, introduced in this paper, is beneficial, as it enables the generation of new uncertainty measures. In particular, our paper provides closed-form results for the CRPS score, a crucial loss function in meteorological and weather forecasting problems.
>
> ---
>
> > line 217: twice in one sentence "thus".
>
> Thanks, we have fixed the typo.
>
> ---
>
> > line 268: Can you rigorously define the different shifts to the posterior for clarity?
>
> We will update the manuscript and discuss precisely what we mean. But essentially, all of the transformations to the posterior distribution are affine shifts or expansions (see Figure 1).
> For the location shift of predicted means, we make a shear transform of the mean components and shift them by a constant.
> For the location shift of predicted variances, we do the same but for variance components.
> Finally, for mean and variance scale shifts, we contract the corresponding dimensions by a factor greater than 1, while keeping the other one unchanged.
>
> ---
>
> > figure 1: column middle and right: font size too small to be readable. Possibly try to colorcode or otherwise highlight the different risks on the x-axis to simplify readability.
> > What does the gray background color for the Bayes risks indicate?
> > For row 1 and 4 you do not explain what the results show in contrast to row 2/3.
>
> Thank you!
> Based on the feedback from you and other reviewers, we reconsidered this image. We plotted an alternative one, where the same data and uncertainty measures are represented differently.
> Please see updated Figure 2 in the manuscript. In this Figure, the left panel represents the original data-generating process and samples. The panel in the middle demonstrates predictions of networks trained on the training data. Finally, the right panel displays the uncertainty measures (of all types) evaluated under log score instantiations. All the discussion we had before is still valid for this updated image. It is the same data, but visualized differently, that, based on the reviewers' comments, better addresses the questions and clarifies the experiment and observations.
>
> ---
>
> > line 272: What do you mean specifically with "None of the measures considered changes under this shift"?
>
> By this sentence, we imply that all the measures are insensitive to changes in the additive constant of predicted means. Therefore, if we apply a shear additive transform to the posterior over means, our measures will not capture the shift. This is because Bayesian uncertainties are rooted in disagreement (which arises from the spread of the posterior distribution). But if one shifts the posterior, without changing its "variance", the measures will not be affected.

---

> ### Author Response · Authors · 2025-11-22
>
> > section 5.1: I am missing an interpretation of the results and to put it into context with the developed theory. Why are the results the way they are and what does this imply for the application of the specific measures?
>
> Thank you for your question!
>
> The results presented in this section align well with the proposed methodology. We will clarify this more precisely in the revised manuscript, and we have added an explicit discussion on it.
>
> Specifically, our framework allows not only for a plain generation of uncertainty quantification measures, but also for distinguishing different types of uncertainty. In this subsection, we analyze the behavior of the proposed disentangled uncertainties in terms of their sensitivity to different controllable kinds of transformations of the posterior distribution. We discuss the intuition of what we expect the uncertainty measure to do when a specific transform of the posterior distribution occurs and what we actually observe.
>
> Again, we have made a more precise comment on the connection of this section to the whole methodology, and emphasized it in the Discussion section.
>
> ---
>
> > Figure 2: missing description of what the red and blue lines mean and why the dotted risks are not continuous. This also makes the interpreation of the text hard since I do not know what I am looking at. Font too small again.
>
> Thank you for the comment!
>
> We have reconsidered the Figure and have updated it. Please see the new version.
>
> Specifically, to your question, the color of the dots represents the amplitude of the uncertainty measure. The brighter it is, the higher the uncertainty is evaluated at the point.
>
> The dots on the plot represented actual test points, where we provided the prediction (the location of the dot) and evaluated the uncertainty (the color of the dot).
>
> We thank the reviewer for emphasizing the issues with the image and invite you to see the revised version in the updated manuscript.
>
> ---
>
> > Section 5.2: The example with the figure and the text are quite hard to understand and follow. I do not really get what you are doing, what you are showing, and what the results mean. Can you please clarify this whole section?
>
> We have updated the section in the revised manuscript, where we describe the experiment more precisely. Together with the updated figure, it is clearer now what the problem statement is and the results. Thank you for pointing this out!
>
> ---
>
> > Section 5.3: you suggest to use $\widehat{R}\_{\mathrm{Tot}}^{1,1}$ for selective prediction. Can you connect that to theory why this specific risk is useful in this task? Are there any insights besides the purely empirical numbers? Similar questions hold for section 5.4
>
> Thank you for the question!
>
> Unfortunately, it is impossible to provide any theoretical guarantee that a particular approximation (including $\widehat{R}\_{\mathrm{Tot}}^{1,1}$) will always be the best in a specific downstream problem.
> Therefore, the observed success is an empirical observation that holds true across several different datasets.
>
> One interesting observation can be made, however. Specifically, $\widehat{R}\_{\mathrm{Tot}}^{1,1}$ = $\widehat{R}\_{\mathrm{Bayes}}^{1}$ + $\widehat{R}\_{\mathrm{Exc}}^{1,1}$. As shown in Kotelevskii et al. (2025, eq. 6), $\widehat{R}\_{\mathrm{Bayes}}^{1}$ will provide a tighter approximation of the actual Bayes risk than $\widehat{R}\_{\mathrm{Bayes}}^{2}$ (on average). However, this does not indicate how well $\widehat{R}\_{\mathrm{Tot}}^{1,1}$ and $\widehat{R}\_{\mathrm{Exc}}^{1,1}$ are approximated.

---

> > ### Comment · Reviewer_3MWm · 2025-11-27
> >
> > Thank you for the detailed rebuttal and clarifications. Many points have been clarified. However, several points remain insufficiently addressed in the manuscript:
> > - Limitations of the approach as you listed them, and including the Gaussian parametric assumption and computatonal cost of Bayesian approximations
> > - comparison to baselines is still missing and should be added
> > - a discussion on the connection of theory to empricial results
> > - clarifications such as the definitions of posterior shifts
> >
> > Therefore, until these points are resolved or argued why those points should not appear within the manuscript, I will remain with my score.

---

> > > ### Author Response · Authors · 2025-12-02
> > >
> > > We thank the reviewer for their thoughtful feedback and are glad that many points were clarified by our rebuttal. Regarding the remaining points that should be addressed in the manuscript, we took the following actions. The respective changes in light of this reply are highlighted in blue in the updated paper.
> > >
> > > > Limitations of the approach as you listed them, and including the Gaussian parametric assumption and computational cost of Bayesian approximations
> > >
> > > We added a new discussion to the conclusion section (in blue), highlighting the limitations outlined by the reviewer. Still, we want to stress that the proposed framework itself is not restricted to the Gaussian parametric assumption, it may also be useful under other parametric assumptions like Laplace that allow to derive closed form solutions for particular proper scoring rules. Furthermore, we only explored Deep Ensembles as a method for Bayesian approximation and it would be an exciting avenue for future work to explore the impact of using other, computationally cheaper methods such as MC dropout or Laplace approximations. Our choice was motivated by the fact that Deep Ensembles are widely used in practice and have emerged as a gold standard for Bayesian approximation for Deep Neural Networks, see e.g. Wilson & Izmailov (2020).
> > >
> > > Wilson, Izmailov (2020) Bayesian Deep Learning and a Probabilistic Perspective of Generalization, NeurIPS
> > >
> > > > comparison to baselines is still missing and should be added
> > >
> > > Many of the standard baselines for OOD detection, such as maximum softmax entropy, outlier exposure, etc. are not directly transferable as they have been proposed for the classification setting. However, we are currently working on providing the widely used “Mahalanobis” baseline, operating on the features of intermediate representations. Due to limited computational resources during the rebuttal period, we are likely not able to include this baseline before the deadline tomorrow.
> > > Furthermore, we want to emphasize that the purpose of our experiments was not to find a new state of the art method for selective prediction, OOD detection or active learning, but to understand the empirical implications of our proposed framework: What risks are useful on what tasks, how different is the behavior of measures derived under different proper scoring rules, what Bayesian approximation strategies are more useful in practice, etc. We believe that our empirical results feature many such insights and can inform future advances on Bayesian approximation methods, where the actual goal is to devise a method that performs state of the art.
> > >
> > > > a discussion on the connection of theory to empirical results
> > >
> > > We interpret this point as requesting a clearer link between theory and results. The theoretical derivation in Sections 2 and 3 showcase how to derive concrete uncertainty measures under some parametric assumptions on the predictive distribution. We look into widely used proper scoring rules and different Bayesian approximation strategies, but none of those have any a priori theoretical advantage over any of the others, they are the available options from which one can choose. We find that empirically, the particular choice of proper scoring rule is not a determining factor, but some Bayesian approximation strategies work better than others. There is also a good discussion on the inability to select the best approximation strategy (other than empirically per task) in Appendix F in Kotelevskii et al. (2025).
> > >
> > > Kotelevskii, Kondratyev, Takac, Moulines, Panov (2025) From Risk to Uncertainty: Generating Predictive Uncertainty Measures via Bayesian Approximation, ICLR
> > >
> > > > clarifications such as the definitions of posterior shifts
> > >
> > > We thought about how to more intuitively define the posterior shifts, but could not think of a better way than we currently do it. We both verbally describe each shift and formally state how the distribution of the posterior distribution parameters change under this shift for the synthetic example we consider, e.g. “Location shift of predicted means: The distribution of predicted means shifts to $\mu \sim \mathcal{U}(1,3)$, thus the location of the posterior shifts, but not its scale.” Furthermore, we visualize each of the shifts in the left column of Figure 1, showing the original and the shifted distribution, which we believe very intuitively showcases the nature of each shift.
> > >
> > > We thank the reviewer again for their constructive feedback and hope our revisions and clarifications adequately address the remaining concerns.

---

### Official Review · Reviewer_XQtn · 2025-10-31

**Soundness:** 3
**Presentation:** 3
**Contribution:** 2
**Rating:** 4
**Confidence:** 4

**Summary:**

This paper puts forward a framework for uncertainty quantification in regression problems. The framework is based on proper scoring rules, which allow for the quantification of the total risk, which is, in this paper, equated to total uncertainty. Using a known decompositions of proper scoring rules into an entropy function and a divergence function, which represent Bayes risk and excess risk or aleatoric and epistemic uncertainty, respectively. The paper suggests three methods to estimate these risks and considers four loss functions as explicit instantiations. The framework is evaluated by looking at a characterization of the uncertainty measures, synthetic experiments, and common uncertainty tasks.

**Strengths:**

The paper uses a valuable framework that allows for a unified consideration of different known uncertainty measures.

The work includes a wide spectrum of empirical evaluations across different uncertainty applications.

In general, the work is very complete, providing a good background on the proper scoring rule literature, derivations, and many additional results.

**Weaknesses:**

The work is very similar to (Kotelevskii et al., 2025), but with different loss functions. It is, therefore, an overstatement to say that this work introduces a framework for uncertainty in regression. The framework already exists, the authors cite many previous works using the same framework, but it is used in a regression context. Thus, the novelty is limited.

The characterization of uncertainty measures section should reference the work by Bülte et al. (2025). Many of the “shifts” in the current work were introduced in (Bülte et al., 2025).

The characterization is not connected to the rest of the empirical findings. Specifically, it would be valuable to connect the performance of specific loss-approximation combinations on tasks such as selective prediction to specific properties that the measures satisfy.

Additionally, the paper mostly reports how measures perform, but there is no discussion about *why* certain measures and approximations perform well.

**Questions:**

For the three tasks (selective prediction, out-of-distribution detection, and active learning), can you give reasons for the good performance of the best performing instantiations. Specifically, could you explain why using these particular losses and approximations work well for the respective tasks?

---

> ### Author Response · Authors · 2025-11-22
>
> We thank the reviewer for helpful feedback and comments. Please find the answers below.
>
> > The work is very similar to (Kotelevskii et al., 2025), but with different loss functions. It is, therefore, an overstatement to say that this work introduces a framework for uncertainty in regression. The framework already exists, the authors cite many previous works using the same framework, but it is used in a regression context. Thus, the novelty is limited.
>
> While we agree with the reviewer that some steps in our paper follow those in (Kotelevskii et al., 2025), there are important distinctions that should be noted.
>
> First, in the case of regression, in order to obtain closed form expressions one must choose a predictive distribution. In our case, we have considered a Gaussian one, as it is a popular practical choice. However, one can consider other options.
> In contrast, in the case of classification, one can hardly consider anything else apart from the categorical distribution.
>
> Next, the providing the explicit, computable formulas for all the considered scoring rules and risk approximations, even for the case of a Gaussian predictive distribution, is not a trivial problem, and to the best of our knowledge, was not fully done in the previous works.
>
> ---
>
> > The characterization of uncertainty measures section should reference the work by Bülte et al. (2025). Many of the “shifts” in the current work were introduced in (Bülte et al., 2025).
>
> Thank you for this remark. The work by (Bülte et al., 2025) [which we cite] is indeed one of those that utilize the idea of shifts to demonstrate the behaviour of uncertainty measures. We will emphasize this explicitly in the revision.
>
> ---
>
> > The characterization is not connected to the rest of the empirical findings. Specifically, it would be valuable to connect the performance of specific loss-approximation combinations on tasks such as selective prediction to specific properties that the measures satisfy.
> > Additionally, the paper mostly reports how measures perform, but there is no discussion about why certain measures and approximations perform well.
> > For the three tasks (selective prediction, out-of-distribution detection, and active learning), can you give reasons for the good performance of the best performing instantiations. Specifically, could you explain why using these particular losses and approximations work well for the respective tasks?
>
> We kindly disagree with the reviewer's evaluation.
>
> The idea behind this experiment is to analyze the behavior of the proposed uncertainty measures under controlled changes to the posterior distribution. This allows us to study how different changes in the posterior are reflected in changes to the uncertainty measures.
>
> While this Section does not explicitly discuss the choice of approximation, this experiment allows us to capture failure modes linked to the choice of the proper scoring rule and the choice of risk. The following experiments also discuss the choice of the Bayesian approximation. Therefore, Section 5.1 should be considered not in isolation, but as part of the entire experimental evaluation.
>
> To make the conclusions and recommendations more explicit, we have dedicated a separate section, "Discussion", where we list our observations and recommendations.

---

### Official Review · Reviewer_myXV · 2025-11-01

**Soundness:** 2
**Presentation:** 2
**Contribution:** 2
**Rating:** 2
**Confidence:** 3

**Summary:**

The paper aims to extend the theoretical framework of proper scoring rules (which previously achieved more attention for classification) to the regression setting for uncertainty quantification (UQ). The authors give a systematic overview of different uncertainty measures derived from different scoring rules. These uncertainty measures can quantify the total risk and allow for decomposing the total risk into aleatoric and epistemic components. The paper also derives closed-form approximations under Gaussian assumptions, and provides empirical evaluations on synthetic and real-world datasets.

While the topic is relevant and potentially impactful, the novelty is limited. The paper largely reformulates existing ideas (such as CRPS, log score, Bayes risk, and related decompositions) under restrictive Gaussian assumptions. The empirical analysis reproduces expected behaviors but does not generate novel insights or practical guidance.

**Strengths:**

The paper tackles a relevant and timely topic. This is an area of active interest in both machine learning and statistics, and the authors’ attempt to build a framework is well-motivated. Uncertainty quantification and disentangling uncertainty into aleatoric and epistemic uncertainty are important for many applications. (However, the choice of the scoring rule does not seem to be the most pressing issue in this context.)

The empirical section is broad in scope, covering multiple datasets and several downstream applications such as selective prediction, OOD detection, and active learning. The results are consistent with theoretical expectations.

The paper nicely demonstrates that disentangling uncertainty into epistemic and aleatoric uncertainty does have a significant impact on downstream tasks for regression. This provides a useful counterargument against the framing by [Mucsányi et al] (https://proceedings.neurips.cc/paper_files/paper/2024/hash/5afa9cb1e917b898ad418216dc726fbd-Abstract-Datasets_and_Benchmarks_Track.html ). [Mucsányi et al] are sometimes interpreted as disentangling uncertainty into epistemic and aleatoric uncertainty doesn't work in practice (for classification). But the experiments by [Mucsányi et al] are limited and focus on classification only. Therefore the regression results by this paper help to put the results by [Mucsányi et al] into perspective.

**Weaknesses:**

Questionable novelty of the theoretical contribution. The theoretical sections largely restate known results (CRPS, log score, Bayes risk, total/excess risk) and rederive them under restrictive Gaussian assumptions. These notions are all well-established. It is unclear what genuinely new theoretical result the paper contributes beyond repackaging existing ideas and recomputing them explicitly for the Gaussian case.  Furthermore, the results in Tables 2–3 are mostly trivial as far as I understand, or am I wrong? I agree that these tables are a nice way to quickly provide an overview of the main intuition behind these different uncertainty measures; however, I would not count this as a major contribution. There is no Theorem that is novel, and I do not see any new insight for practitioners. I would summarize the main takeaway for practitioners as: the typically used decompositions by (Depeweg et al., 2018) work well (sometimes CRPS1,1 can be slightly better). No significant improvement over them was found, and in practice, it does not really matter which of them one should use. There are also some other possibilities to define it that behave almost identically. There are some other strange exotic ways to define it that nobody would ever use in practice, that actually don't work that well in practice. However, distinguishing between epistemic, aleatoric, and total uncertainty can have a significant difference on downstream tasks. Do you agree that this is the main summary of the results, or is there more to it?

When reading the paper for the first time, I thought: <<Strong and limiting assumptions. The framework relies entirely on Gaussian assumptions. While this simplifies derivations, it severely limits generality. Moreover, this procedure effectively measures the discrepancy between $\hat{P}$ and the Gaussian approximation, which is questionable if the Gaussian model is itself misspecified. You boldly claim that “it is necessary to make specific parametric assumptions.” Why should that be the case? There exist many nonparametric approaches to estimate P(Y|X) without imposing such restrictive assumptions.>>. Now reading it again, I think this was maybe a misunderstanding. Now, I think Tables 2 and 3 are rather an intuitive example of a more general framework. Maybe the paper could be clearer about what the main contributions are.

Issue with main contributions 3: You wrote ``A broad empirical evaluation on synthetic and real-world datasets, culminating in practical recommendations for selecting uncertainty measures (Section 5).’’ The claimed “recommendations” are missing. They are neither summarized in the conclusion nor presented in any subsection. If they exist somewhere in Section 5 or the appendix, they are too well hidden to be useful.

Empirical evaluation offers little new insight. Although the experiments are extensive, they mainly confirm well-known qualitative behaviors (“as theory suggests”) rather than providing new understanding or guidelines. The “practical recommendations” promised in the abstract and contributions are not actually presented in the paper or are too buried to be actionable.

Presentation:

* Figure 2 is very unclear. The meaning of the colors (red, blue, and multicolored dots) is not explained; at least not in the main text. If one needs to consult the appendix to understand the figure, the figure itself probably belongs in the appendix. The colors of the dots are explained, but what do the x and y coordinates of the colorful dots correspond to. What are the red and blue lines? What was the training data? Section 5.2 is absolutely not understandable without the abstract.
* Figures 2 and 3 are also too small. Typical guidelines state that the text and axes labels should be readable in print without zooming in. As it stands, they are so insanely small that I can not even zoom in enough on my tablet. The quality of the figure is good, though.
* Tables 2 and 3 include unexplained symbols (e.g. $A$). These should be defined in the captions or main text. I suppose they can be found in the appendix, but since they are used in the 'main contributions section', it should be at least explained why there are undefined symbols.
* Sections 2 and 3.1 could be merged. Much of the material repeats: the definitions of total risk (Equation 1) and Bayes risk (Definition 1) overlap conceptually, but occupy substantial space. Just an idea. I leave the choice up to you.
* In my opinion, the goal of such a survey paper should be to clarify things also for people who are new to this field. I'm a bit worried that some of it creates more confusion than clarification. For example, $\hat{R}^2_{\text{Bayes}}$ for SE, is something which would usually be referred to as total uncertainty rather than aleatoric uncertainty. However, the Bayes risk is supposed to estimate the aleatoric uncertainty. This is confusing. I understand that in your hyper-general framework, when you try to write all possible combinations of ideas, then you end up with strange situations like this. But for someone who is new to the field, this could be very confusing. I think it would be good to add a discussion warning readers about these extremely counterintuitive uncertainty measures. It seems like this plug-and-play framework that you introduce does not always make sense for every plug-and-play combination.I think, especially for the readers who are new to these topics and are looking for a nice overview, there should be a clear message that the current state of the art is to simply use the well-established decompositions by (Depeweg et al., 2018) instead of putting too much thought into this hyper-general plug-and-play system. Also, you talk about 9 combinations, but (2,2), for example, obviously does not make any sense, and then you actually don't show (2,2) in your experiments, but you never explain this or write about this in the paper. Overall, it seems as if this framework should not be sold as a general plug-and-play system, but rather there are some combinations that make sense (the ones by  (Depeweg et al., 2018) and a couple more) and others that don't make sense.
* Line 195: Not really understandable without the appendix. When you write 3a and 3b, does this correspond to $\hat{P}\_\star$ and $\bar{P}_*$?
* Line 405: You write "substantially worse", but I would say "slightly worse", as the difference is not significant and really tiny. (For vegetation, arrow, and dots Log is even the best, not worse.)
* I think Table 4 is a bit misleading as it averages over multiple datasets with results on slightly different scales from Table 9. In Table 9, one can better see how inconsistent the rankings are across datasets.
* Line 128: There is a “with with” typo.

**Questions:**

1. Can you say very specifically what the paper’s novel theoretical contribution (Sections 1-4) is? Is it anything beyond Tables 2 and 3? I am just not really seeing which definition/computation / … can't be found in existing literature (note that I did not dig deep into the Appendix)
2. Why is Gaussianity treated as necessary? Couldn't the framework be extended to nonparametric predictive distributions?
3. Could you summarize the practical recommendations your experiments yield? What should the reader actually take away from your results?
4. In paragraph "Scale shift of predicted means," it sounds as if you want aleatoric uncertainty to increase for scale shifts of the predicted mean. Intuitively, I would say that the **aleatoric** uncertainty should not be affected by a Scale shift of predicted means. Please clarify which uncertainty component is expected to increase under a scale shift of the predicted means.
5. How do Eq. (3) and (4) fit into your framework as the logarithmic and SE special cases? As superscript 1,2? Why not state this more explicitly?

It was hard for me to decide whether to give a 2 or a 4. So, if you improve the presentation and soften the tone (i.e., don't claim that something is substantially worse, when it is actually only worse by ~1.5% on average, which is not even statistically significant)

---

> ### Author Response · Authors · 2025-11-22
>
> We are grateful to the reviewer for their thorough feedback and comments. Please find our answers below.
>
> ---
>
> > ... Do you agree that this is the main summary of the results, or is there more to it?
>
> We do not agree with this summary of our results by the reviewer.
>
> "*Theoretical sections...*": Theoretical sections aim to provide practitioners with the toolbox to compute epistemic and aleatoric uncertainty estimates for the important practical case of an ensemble of Gaussian predictions. Such an ensemble approach is probably the most standard and common way to do probabilistic prediction for regression. While some of the results (for log- and SE losses) are well-known in the literature, the results for other losses are much less known to the community and are partially new.
>
> "*results in Tables 2-3 are mostly trivial*": We respectfully disagree with the reviewer. Our work proposes a clear methodology for practitioners and a concrete construction recipe that allows for the derivation of new uncertainty measures in a regression setting. We also provide closed-form, ready-to-implement uncertainty measures based on multiple proper scoring rules, which are useful to practitioners. Although some closed-form expressions for proper scoring rules were available before, our work is the first (to our knowledge) that derives all these uncertainty measures (presented in the appendix) for a popular practical choice of Gaussian ensembles and presents them in one place, which is not trivial.
>
> "*typically used decompositions*" and Depeweg et al., 2018: We agree that the mentioned work proposed two uncertainty measures that we also derived. However, the context and assumptions are different. Depeweg et al. considered a specific Bayesian latent variable model and a data model with additive Gaussian noise, for which they proceeded with the derivations of variance and entropy estimates of the predictive distribution. In contrast, we provide a general construction based on proper scoring rules that goes beyond entropy and variance without strong assumptions on the data and the inner structure of the model.
>
> While some ideas and results may have been scattered around previous works, our paper is the first, to the best of our knowledge, to connect them to a broader proper scoring rules framework and provide a general construction. It gives more choices as well as a simple and actionable recipe. Thus, we do not agree that it is a 'repackaging'. Instead, some of the known measures/decompositions naturally emerge as special cases of our approach. We also believe that particularly new CRPS score-based measures can be useful, especially for the models trained with CRPS loss (which becomes more and more popular, especially for weather forecasting).
>
> Future work may consider other predictive distributions than a Gaussian, and can exercise the steps of our framework to arrive at new uncertainty measures.
>
> Regarding practical recommendations in the new section "Discussion", we have identified failure modes associated with some approximations. We detail it in there.
>
> ---
>
> > Now, I think Tables 2 and 3 are rather an intuitive example of a more general framework. Maybe the paper could be clearer about what the main contributions are.
>
> Yes indeed it was meant as intuitive example of a general framework, motivated by a widely used practical choice of Gaussians. We will emphasize that in the text. Other approximations are definitely possible, but in our opinion are less likely to provide closed-form solutions, which was one of our goals from the start.
>
> ---
>
> > The claimed “recommendations” are missing...
>
> Thanks for spotting this. We added a "Discussion" section in the updated version of the paper, where we do this more explicitly.
>
> ---
>
> > Empirical evaluation offers little new insight. They mainly confirm well-known qualitative behaviors.
>
> We respectfully disagree with the reviewer's assessment of the lack of new insights. First, we identified some failure modes of the approximations (summarized in the new Discussion section).
> Second, while we agree that our experiments confirm the qualitative behaviour, we do not see this as a weakness, but rather a reassuring strong point. In fact, our measures, derived under popular and practical considerations (ensembles of Gaussian predictors), lead to the behaviour that resonates with common and well-established intuition.
>
> ---
>
> > Figure 2 is very unclear...
>
> Thank you for your feedback on this figure. We updated it and added a more detailed description to the main paper to make it understandable from the main text alone.
>
> > Figures 2 and 3 are also too small.
>
> We updated the figure accordingly.
>
> > Tables 2 and 3 include unexplained symbols.
>
> We made the necessary updates.

---

> ### Author Response · Authors · 2025-11-22
>
> > In my opinion, the goal of such a survey paper...
>
> We kindly disagree with the reviewer's evaluation that our paper is a survey paper. A survey paper is about summarizing the results and introducing a hierarchy or dichotomy to better structure the existing literature.
> In our paper, however, we adapt the framework introduced previously only for classification to regression. We also derive closed-form equations for different popular choices of regression proper scoring rules under the widely used Gaussian assumption for the predictive distribution.
> One of the interesting results for practitioners introduced in this paper could be the closed-form practical equations for the CRPS score. The CRPS score is of great interest in applications related to meteorology and weather forecasting [1,2]. The measures introduced in this paper can be further applied to downstream uncertainty quantification problems in the weather forecasting domain.
>
> By no means our paper contradicts or challenges (Depeweg et al., 2018). We did not intend this, and if the reviewer feels otherwise, we can assure them that this is not the case and are open to adjust any particular phrasing.
>
> Additionally, we neither consider nor position our paper as a "good overview for those who are new to the field". For these purposes, indeed, a good survey paper can be written.
> Our paper introduces a comprehensive toolbox of ready-to-use and practical closed-form equations that can be utilized in the downstream regression uncertainty quantification problem.
>
> Regarding the reviewer's point on "very confusing and counterintuitive measures" of uncertainty. Indeed, for a specific combination, $\widehat{R}\_{\mathrm{Exc}}^{2,2}$  will be 0, and $\widehat{R}\_{\mathrm{Bayes}}^2$ will coincide with the total uncertainty. This does not indicate the issue with the framework itself. This only highlights the nature of these Bayesian uncertainty measures. Specifically, they are based on disagreement. And when only one model is provided (effectively the case of the $(2,2)$ choice), then the disagreement will be zero, and the effective epistemic uncertainty is 0. We agree, though, that this is worth explicitly emphasizing, and we will add it in the revision.
>
> [1] Alet, Ferran, et al. "Skillful joint probabilistic weather forecasting from marginals." arXiv preprint arXiv:2506.10772 (2025).
> [2] Lang, Simon, Martin Leutbecher, and Pedro Maciel. "A multi-scale loss formulation for learning a probabilistic model with proper score optimisation." arXiv preprint arXiv:2506.10868 (2025).
>
> ---
>
> > Line 195: Not really understandable without the appendix. When you write 3a and 3b, does this correspond to $\hat{P}\_*$ and $\bar{P}\_*$ ?
>
> Yes exactly, we update the main paper to make this more clear.
>
> ---
>
> > Line 405: You write "substantially worse", but I would say "slightly worse", as the difference is not significant
>
> Thank you for spotting this. We have revised the wording in accordance with the suggestion from reviewer.
>
> ---
>
> > I think Table 4 is a bit misleading as it averages over multiple datasets with results on slightly different scales from Table 9. In Table 9, one can better see how inconsistent the rankings are across datasets.
>
> While the absolute values of individual datasets are indeed on different scales, they all follow roughly the trend that the bold ones of Table 4 are best and that Bayes risks perform rather bad in comparison.
>
> ---
>
> > Line 128: There is a “with with” typo.
>
> Thank you, we fixed this typos in the updated version of our work.
>
> ---
>
> > Can you say very specifically what the paper’s novel theoretical contribution (Sections 1-4) is? Is it anything beyond Tables 2 and 3? I am just not really seeing which definition/computation / … can't be found in existing literature (note that I did not dig deep into the Appendix)
>
> As we wrote in the Introduction section, we consider our contribution as follows:
>
> 1) A theoretical formulation of regression UQ based on proper scoring rules;
> 2) Closed-form, practically computable approximations for these components;
> 3) A broad empirical evaluation on synthetic and real-world datasets.
>
> "*Is it anything beyond ...*": There are no papers, up to the moment of submission, that introduce a general uncertainty measure generative framework with proper scoring rules for regression. We are not aware of any other papers that provide closed-form equations for the uncertainty measures we considered. Our work provides Tables 2 and 3 together with results in the Appendix, forming a single toolbox for practitioners, which is a valuable contribution.
>
> "*I am just not really seeing which definition/computation / … can't be found in existing literature*":
> To the best of our knowledge, there are no ready-to-use uncertainty measures of this type that have been derived in closed form for CRPS. If the reviewer disagrees with this, we invite them to provide a specific reference to the paper.

---

> ### Author Response · Authors · 2025-11-22
>
> > Why is Gaussianity treated as necessary? Couldn't the framework be extended to nonparametric predictive distributions?
>
> We kindly disagree with the reviewer's wording. We did not use this statement nor implied it. The Gaussianity itself is never treated as a necessity.
> What we wrote was that "*it is necessary to make specific parametric assumptions about the distribution*".
> Gaussianity is a popular practical choice. Another advantage of the Gaussian assumption on the predictive distribution is that it enables the closed-form computation of the integrals.
> Obviously, one can consider other parametric (or even non-parametric) choices (e.g., predictive Laplace distribution). However, we are unsure if the closed-form equations will appear in those cases. It is not necessary, by all means, to require closed-form equations. However, otherwise, one needs to approximate corresponding integrals.
> We already approximate integrals over the posterior distribution using samples from the parameter space (treating members within the ensemble as those). Approximating integrals over predictive distributions introduces an additional layer of stochasticity.
> Nevertheless, it is possible, but it might challenge practical application.
> Taking reviewers concerns into account, we have revised our wording in the corresponding section to make this argument more clear.
>
> ---
>
> > Could you summarize the practical recommendations your experiments yield? What should the reader actually take away from your results?
>
> We have added a new section, Discussion, to the main text, where we summarize the results of the experiments and provide practical recommendations.
>
> ---
>
> > In paragraph "Scale shift of predicted means," it sounds as if you want aleatoric uncertainty to increase for scale shifts of the predicted mean. Intuitively, I would say that the aleatoric uncertainty should not be affected by a Scale shift of predicted means. Please clarify which uncertainty component is expected to increase under a scale shift of the predicted means.
>
> Thank you for the interesting question! First, to clarify, in this Section, we are only talking about the change in the **estimate** of AU. The ground-truth AU is model-agnostic and will not change, indeed.
> Regarding the expected change in the estimates for the AU, it is indeed less intuitive and requires explanation. Some of our AU estimates consider **only** predicted variances, and therefore, they exhibit no change under this shift, which is in agreement with the reviewer's intuition.
> Others, however, take predicted means into account. And they do it in a nuanced way, considering the "variance of means" around the average prediction (see Table 2, eq for SE, and eq 5 in the Appendix). In this case, the estimate of AU is expected to increase.
> The question is to what type of uncertainty we should attribute this variance in means. There are two equivalent approximations of a total uncertainty: $\widehat{R}\_{\mathrm{Exc}}^{1,1} + \widehat{R}\_{\mathrm{Bayes}}^{1} = \widehat{R}\_{\mathrm{Exc}}^{2,1} + \widehat{R}\_{\mathrm{Bayes}}^{2}$ (see Kotelevskii et. al, 2025 [Appendix F] and Section A in the Appendix in our paper). Moreover, $\widehat{R}\_{\mathrm{Exc}}^{1,1} = \widehat{R}\_{\mathrm{Exc}}^{1,2} + \widehat{R}\_{\mathrm{Exc}}^{2,1}$. From these equations, it is evident that a portion of uncertainty can be attributed to epistemic uncertainty, or to aleatoric uncertainty. The choice of what it is assigned to determines the result (either AU is increased or remains unchanged). We agree that this particular moment requires nuanced discussion, and we will update the writing in the revision.
>
> ---
>
> > How do Eq. (3) and (4) fit into your framework as the logarithmic and SE special cases? As superscript 1,2? Why not state this more explicitly?
>
> We agree that making this statement more explicit will benefit the exposition and have included the corresponding changes.

---

### Official Review · Reviewer_wTuu · 2025-11-02

**Soundness:** 2
**Presentation:** 3
**Contribution:** 2
**Rating:** 4
**Confidence:** 4

**Summary:**

This paper introduces measures of uncertainty for regression based on proper scoring rules. The derived measures naturally decompose (additively) into aleatoric and epistemic parts. The proper scoring framework recovers popular measures (variance- and entropy-based ones). The authors empirically evaluate on synthetic and real-world regression datasets.

**Strengths:**

The research direction itself seems to be very promising and of considerable importance for current developments in the (reliable) machine learning community. Given that loss based approaches to uncertainty quantification for classification purposes turned out to be quite successful in the recent past, a regression lens seems to be the logical next step.

In my opinion, the paper’s main contribution is a fairly original framing of uncertainty quantification through proper scoring rules and Bayes risk, coupled with a systematic comparison of three relevant (in particular for practical use cases) aggregation schemes (per-member averaging, mixture, and single-Gaussian fit). This indeed unifies several ad-hoc practices under one lens and connects them to familiar variance/entropy based uncertainty measures, which is useful for both theory and practice (for the latter see my later critical remarks).

Moreover, the presentation is generally speaking quite clear. The paper summarizes relevant formulas in tables, and explains computational trade-offs of the three aggregation schemes so practitioners can anticipate cost implications.

Empirically, the evaluation covers synthetic and real-world settings and shows that the proposed aggregations are broadly comparable in practice, which is a valuable takeaway by itself.

**Weaknesses:**

First, let me elaborate on some more technical things I noticed. For brevity, let $\hat p_{\text{e}}=\frac{1}{M}\sum_{i=1}^M \hat p_i$. Then
\begin{align}
\hat{R}^{2}
= -\int \hat p_{\text{e}} \log \hat p_{\text{e}}
= -\int \frac{1}{M}\Big(\sum_i \hat p_i\Big)\log \Big(\frac{1}{M}\sum_j \hat p_j\Big)
= -\frac{1}{M}\int \Big(\sum_i \hat p_i\Big)\log \Big(\sum_j \hat p_j\Big) dy  +  \log M.
\end{align}

Thus, there seems to be a minus sign missing in Table 2, and Appendix 2.2. omits an additional $+ \log M$ term. Further, Appendix A.2.2 claims that "the logarithmic score is a proper scoring rule for discrete distributions". This is somehow misleading, since the log score is proper for both discrete and continuous cases.

In L. 216 the authors write that "CRPS is a non-negative metric". CRPS is a strictly proper scoring rule. The metric is the squared L^2 distance between CDFs that arises from expected CRPS, not the pointwise score itself.


Is the third bullet point (in the Bayes risk derivations for the quadratic score) a typo? Should be $H(P_{\star})$, right?

An other weakness (at least from my perspective) is the following. While I agree that such a proper scoring rule framework in the regression setting is indeed suitable, the motivation for me is not immediate; by motivation I mean, for each such loss we get uncertainty measure, but  what do we get by such collection of measures? By a quick literature search, I found some related work (see e..g [1], or [2]), which seems to argue why such different instantiations might be useful (although in the classification case). Could the authors elaborate on this and if possible relate to their theory/experiments?

On the empirical side:

Across selective prediction and OOD tables, many entries across approximation strategies are numerically very close (frequently within one standard deviation), so any rank-ordering should be done very cautiously. It may help to add a short sentence stating that the strategies are broadly comparable empirically, with no consistent winner across datasets/scores.

Several choices make it hard to draw comparative conclusions. More precisely, (i) PRR averages with overlapping SDs lead to near-ties among many methods, (ii) the OOD benchmark is so easy (by construction) that AUROCs saturate, and (iii) active-learning gains are modest and sometimes inconsistent. The setups as implemented provide limited discriminative power to justify those prescriptions beyond qualitative trends. I agree that this is not an easy task at all, but as executed, I am not sure I understand the actual take-away message from the empirics..

Minor things:

L. 128, duplicate "with with".

L. 663, typo "initilizations" instead of "initializations".

[1] Hofman, P., Sale, Y., & Hüllermeier, E. (2025). Uncertainty Quantification with Proper Scoring Rules: Adjusting Measures to Prediction Tasks. arXiv preprint arXiv:2505.22538.

[2] Mucsányi, B., Kirchhof, M., & Oh, S. J. (2024). Benchmarking uncertainty disentanglement: Specialized uncertainties for specialized tasks. Advances in neural information processing systems, 37, 50972-51038.

**Questions:**

I already formulated (implicitly) some questions in the earlier questions, but will list some more that I had during reading the manuscript.

Could you distill decision rules for when to prefer per-member averaging, the mixture, or a single-Gaussian fit (e.g., multi-modal targets, heavy tails, low ensemble diversity)?

Members are trained under a Gaussian likelihood, evaluation spans CRPS, log, quadratic, and squared-error criteria. How sensitive are rankings to loss–evaluation mismatch?

Do your takeaways extend to vector-valued regression?

---

> ### Author Response · Authors · 2025-11-22
>
> We would like to thank the reviewer for their insightful feedback and comments. Please find the answers to your comments and questions below.
>
> > Mistake in  Table 2, and Appendix 2.2
>
> Thank you for spotting this. We have fixed this in the updated manuscript.
>
> > Appendix A.2.2 claims that...
>
> Thank you, the condition "for discrete distributions" is indeed not necessary and we removed it.
>
> > In L. 216 the authors write that "CRPS is a non-negative metric".
>
> Thank you for spotting this imprecise statement, which we corrected in the updated version of the paper.
>
> ---
>
> > While I agree that such a proper scoring rule framework in the regression setting is indeed suitable, the motivation for me is not immediate; by motivation I mean, for each such loss we get uncertainty measure, but what do we get by such collection of measures? By a quick literature search, I found some related work (see e..g [1], or [2]), which seems to argue why such different instantiations might be useful (although in the classification case). Could the authors elaborate on this and if possible relate to their theory/experiments?
>
> Thank you for the interesting question. Indeed, the methodology presented in this paper enables the generation of uncertainty measures based on proper scoring rules. And there are indeed multiple possible combinations.
>
> We see the following motivation for this.
>
> First, this explains the nature of all these measures. We believe this is a crucial point to consider in terms of motivation. If a proper scoring rule is devised that we have not considered in the paper, one can follow the described methodology to derive new uncertainty measures. Moreover, having such a collection of measures, one can analyze their theoretical properties for the whole class (as we know that they are different instances of it), and not separately for each specific instance. This is what distinguishes our paper from the ones the reviewer has mentioned. There, these uncertainty measures are considered as independently existing. In our paper, we show that they have the same root.
>
> Second, regarding the practical usefulness of having multiple measures -- all of them essentially are approximations of an unknown quantity. These approximations may vary in their degree of precision. Therefore, for each of the problems at hand, the practitioner has greater flexibility in choosing the most suitable measure. This might be especially useful if there is expert/prior knowledge on which proper scoring rule induces the error, thereby expressing the mistake in the downstream task in the most effective way. For example, the CRPS score is well-known and widely used by the meteorological community[1,2], and the uncertainty measures based on this score can lead to more nuanced uncertainty measures.
>
> [1] Alet, Ferran, et al. "Skillful joint probabilistic weather forecasting from marginals." arXiv preprint arXiv:2506.10772 (2025).
> [2] Lang, Simon, Martin Leutbecher, and Pedro Maciel. "A multi-scale loss formulation for learning a probabilistic model with proper score optimisation." arXiv preprint arXiv:2506.10868 (2025).
>
> ---
>
> > Across selective prediction and OOD tables, many entries across approximation strategies are numerically very close (frequently within one standard deviation), so any rank-ordering should be done very cautiously. It may help to add a short sentence stating that the strategies are broadly comparable empirically, with no consistent winner across datasets/scores.
>
> Yes we agree that results for different scores are rather close numerically, we added a statement to make this more clear in the text. The major takeaway, especially w.r.t. to Mucsányi et al. the reviewer was referring to, is that the decomposition into aleatoric and epistemic uncertainty makes a difference in contrast to some earlier works that questioned its usefulness.

---

> > ### Author Response · Authors · 2025-11-22
> >
> > > Several choices make it hard to draw comparative conclusions. More precisely, (i) PRR averages with overlapping SDs lead to near-ties among many methods, (ii) the OOD benchmark is so easy (by construction) that AUROCs saturate, and (iii) active-learning gains are modest and sometimes inconsistent. The setups as implemented provide limited discriminative power to justify those prescriptions beyond qualitative trends. I agree that this is not an easy task at all, but as executed, I am not sure I understand the actual take-away message from the empirics.
> >
> > We do not think that the empirical results that we presented are inconclusive. Within the framework we proposed, we did not expect to have a winner or a loser in terms of the proposed measures. Instead, we demonstrate that this methodology allows us to generate a lot of reasonable uncertainty measures. Some of the measures we use are known in the literature, and it is reassuring that the quality they provide is sufficient for addressing the considered downstream problems.
> >
> > We agree with the reviewer that the takeaway messages from the empirical results could be summarized and presented more effectively. Therefore, we introduce an additional section, "Discussion", where we reiterate the conclusions and provide practical recommendations.
> >
> > Some of the specific combinations derived from the proposed framework fail in particular scenarios, which we highlight in the new section. Others, that are performing well, are indeed close. But this is not a problem of the framework. This is reassuring, as the framework allows for the generation of measures that provide good results in practice.
> >
> > We want to emphasize that we consider the main contribution of the paper to be methodological, while the experiments are primarily intended to demonstrate that the measures derived from the framework are reasonable.
> >
> > ---
> >
> > > L. 128, duplicate "with with".
> > > L. 663, typo "initilizations" instead of "initializations".
> >
> > Thank you, we fixed these typos in the updated version of our work.
> >
> > ---
> >
> > > Could you distill decision rules for when to prefer per-member averaging, the mixture, or a single-Gaussian fit (e.g., multi-modal targets, heavy tails, low ensemble diversity)?
> >
> > Thank you for the interesting question!
> >
> > We have not considered the particular experimental question you have raised (to relate the choice to different shapes of the posterior distribution), but we agree it is an interesting question. We will conduct such experiments and report the results as soon as they are ready.
> >
> > We expect that for a very concentrated posterior, there will be almost no difference for different choices of Bayesian approximations. For multi-modal targets and heavy-tailed posteriors, it is less obvious and probably depends on the specific form of it. We will perform additional synthetic experiments and add them to the Paper/Appendix.
> >
> > Here we emphasize important conclusions. We observe that for all of the proper scoring rule instantiations, the excess risk measures, based on the constant predictions (i.e., simultaneously something among 2, 3a, or 3b), are typically worse than measures that take ""diversity"" into account (the ones with 1, therefore with outer expectations). The only exception is the combination of 3b, 2.
> >
> > This highlights the general importance of ensemble diversity. Effectively, any combination of 1 and something among (2, 3a, or 3b) can be considered as a variation around a centroid, and here the ""variance"" around this centroid is essential. If only constant estimators are considered, then information about variation is not taken into account, which typically leads to worse empirical results.
> >
> > For Bayes risk, we observe that $\widehat{R}\_\mathrm{Bayes}^1$ is typically worse in terms of our metrics in downstream tasks. In prior work by Kotelevskii et al. (2025), it was shown that $\widehat{R}\_\mathrm{Bayes}^1$ is a tighter approximation of the pure Bayes risk than $\widehat{R}\_{\mathrm{Bayes}}^2$. Therefore, $\widehat{R}\_{\mathrm{Bayes}}^1$ captures more clean aleatoric uncertainty, while $\widehat{R}\_{\mathrm{Bayes}}^2$ also contains a portion of epistemic uncertainty. This is how a slightly improved performance can be explained.

---

> ### Author Response · Authors · 2025-11-22
>
> > Members are trained under a Gaussian likelihood, evaluation spans CRPS, log, quadratic, and squared-error criteria. How sensitive are rankings to loss–evaluation mismatch?
>
> Indeed, we have considered only one scenario (Gaussian likelihood), which is a popular and important practical choice.
> The Gaussian likelihood objective we used is, in essence, the log-score. Our experiments show that, at least in the case of the log-loss, using the matching score works well. However, we see no noticeable drawback of loss-eval mismatch for the other considered scoring rules, except for the quadratic score.
>
> In principle, we can consider other losses for training the models. For calculating the loss using CRPS and quadratic score, the resulting functional forms for optimizing a single model are very reminiscent of those under the log-score and would result in similar model predictions empirically, thus we would not expect any drastic changes in the results.
>
> For the SE score, there is no explicit term for the variance, as only the mean is regressed. Thus, it would not be suitable to learn heteroskedastic noise, and we could only estimate homoskedastic noise on a validation set. Here, we would expect stark differences under loss-evaluation mismatch.
>
> We did not perform training under all the different losses due to the relatively high computational requirements to perform the full set of experiments for each loss.
>
> ---
>
> > Do your takeaways extend to vector-valued regression?
>
> Yes, the results extend to vector-valued regression: Log score, Quadratic score and SE score are directly applicable to multivariate distributions. For CRPS, however, there may be multiple ways to generalize it to multiple dimensions, one popular choice is the Energy Score. We expect formulas to change but the general construction still applies.

---

### Meta-Review · Area_Chair_nuM5 · 2026-01-07

**Summary:**

The paper proposes a unified framework for uncertainty quantification (UQ) in regression grounded in proper scoring rules. It extends ideas that have been influential in classification to regression by showing how total predictive risk induced by proper scoring rules can be decomposed into aleatoric (Bayes risk) and epistemic (excess risk) components. The framework covers several common scoring rules (logarithmic, CRPS, quadratic, squared error) and multiple approximation strategies for ensembles (e.g., per-member averaging, posterior predictive mixtures, single-Gaussian fits).

Under a Gaussian predictive assumption, the authors derive closed-form expressions (or Monte Carlo approximations) for the resulting uncertainty measures, showing that widely used variance- and entropy-based uncertainties arise as special cases. A large empirical evaluation on synthetic and real-world regression datasets examines behavior under controlled posterior shifts and performance on downstream tasks such as selective prediction, OOD detection, and active learning.

**Reviewer Concerns:**

Reviewers were divided on the work, leaning negative. The main points of concern were as follows.

- Novelty (raised by myXV, XQtn, wTuu): A recurring concern is that the paper largely repackages existing concepts (proper scoring rules, Bayes risk decompositions, Gaussian ensembles) rather than introducing fundamentally new theory. Several reviewers argue that most results are either already known or become trivial once Gaussianity is assumed, with Tables 2-3 seen as summaries rather than new findings. The authors argue that the novelty lies in (i) explicitly extending the proper-scoring-rule framework to regression, and (ii) deriving close-form expressions, especially for CRPS-based measures, in a single unified treatment. Both the reviewers and I remain unconvinced that unification and collation alone justify the strength of the novelty claims.

- Gaussian assumptions (raised by myXV, 3MWm, XQtn): Reviewers note that the framework’s practical instantiation depends heavily on Gaussian predictive distributions, greatly limiting generality and potentially conflating approximation error with uncertainty estimation. The authors clarify that Gaussianity is not conceptually required, but chosen to enable closed-form results, and that extensions to other parametric or nonparametric distributions are possible with additional approximation. Nevertheless, the concerns remain, and further efforts could have been taken to consider “near-Gaussian” families.

- Few practical takeaways (raised by myXV, XQtn, 3MWm, wTuu): Reviewers criticize the empirical section for being extensive but inconclusive, with many near-ties, saturated benchmarks, and modest gains. The promised practical recommendations were initially missing or buried, making it hard to extract actionable guidance. In response, the authors added a discussion section summarizing observations, failure modes, and high-level guidance (importance of ensemble diversity, caution with certain approximations). This is certainly an improvement, although the recommendations remain weakly discriminative.

- Theory-practice gap (raised by XQtn, 3MWm): Reviewers asked why certain scoring-rule / approximation combinations perform better in specific tasks and how the empirical findings relate to the theoretical properties of the measures. The authors push back on this criticism, arguing that the goal is to demonstrate reasonableness rather than explain optimality, and that no single theoretically best instantiation exists. The concern remains, but I can appreciate the authors’ position.

- Presentation (raised by wTuu, myXV, 3MWm): Numerous issues were raised with the presentation, including typos, imprecise statements (e.g. CRPS as a metric), unclear figures, undefined symbols, and confusing notation around different risk combinations. These concerns were largely addressed in the rebuttal.

- Baseline comparisons (raised by 3MWm, myXV): Some reviewers note the absence of strong external baselines and argue that comparisons are mostly internal (within the same framework), limiting the significance of empirical claims. The authors acknowledge this and promise further baselines in future revisions. This is a common criticism raised in these venues, and is of less urgency to me than the other concerns about novelty.

**Reviewer Scores:**

Of the reviewers, only Reviewer 3MWm engaged in more substantial discussion with the authors, but remained steadfast in their impression that several important criticisms remained unresolved. I agree with this assessment and doubt that the concerns would be easily addressed in a discussion period.

Gaussian process theory can be difficult to publish in modern ML venues due to the saturation of the field. My impression here is similar to many other similar papers in this field that fail to inspire reviewers: the authors should strongly consider advancing their theoretical contributions to produce more substantial and novel results, rather than relying on the convenience of explicit expressions. This may be in the form of new bounds or imposing limits that reveal the Gaussian assumption, rather than imposing it directly.

---

### Decision · Program_Chairs · 2026-01-26

Reject